# AMORTIZED IMPLICIT DIFFERENTIATION FOR STOCHASTIC BILEVEL OPTIMIZATION

**Michael Arbel & Julien Mairal**
Univ. Grenoble Alpes, Inria, CNRS, Grenoble INP, LJK, 38000 Grenoble, France.

## ABSTRACT

We study a class of algorithms for solving bilevel optimization problems in both *stochastic* and *deterministic* settings when the inner-level objective is strongly convex. Specifically, we consider algorithms based on inexact implicit differentiation and we exploit a *warm-start strategy* to amortize the estimation of the exact gradient. We then introduce a unified theoretical framework inspired by the study of *singularly perturbed systems* (Habets, 1974) to analyze such *amortized* algorithms. By using this framework, our analysis shows these algorithms to match the computational complexity of oracle methods that have access to an unbiased estimate of the gradient, thus outperforming many existing results for bilevel optimization. We illustrate these findings on synthetic experiments and demonstrate the efficiency of these algorithms on hyper-parameter optimization experiments involving several thousands of variables.

## 1 INTRODUCTION

Bilevel optimization refers to a class of algorithms for solving problems with a hierarchical structure involving two levels: an *inner* and an *outer* level. The *inner*-level problem seeks a solution $y^\star(x)$ minimizing a cost $g(x, y)$ over a set $\mathcal{Y}$ given a fixed outer variable $x$ in a set $\mathcal{X}$. The *outer*-level problem minimizes an objective of the form $\mathcal{L}(x) = f(x, y^\star(x))$ over $\mathcal{X}$ for some *upper*-level cost $f$. When the solution $y^\star(x)$ is unique, the bilevel optimization problem takes the following form:

$$\min_{x \in \mathcal{X}} \mathcal{L}(x) := f(x, y^\star(x)), \qquad \text{such that } y^\star(x) = \arg\min_{y \in \mathcal{Y}} g(x, y). \tag{1}$$

First introduced in the field of economic game theory by Stackelberg (1934) and long studied in optimization (Ye and Zhu, 1995; Ye and Ye, 1997; Ye et al., 1997), this problem has recently received increasing attention in the machine learning community (Domke, 2012; Gould et al., 2016; Liao et al., 2018; Blondel et al., 2021; Liu et al., 2021; Shaban et al., 2019; Ablin et al., 2020). Indeed, many machine learning applications can be reduced to (1) including hyper-parameter optimization (Feurer and Hutter, 2019), meta-learning (Bertinetto et al., 2018), reinforcement learning (Hong et al., 2020b; Liu et al., 2021) or dictionary learning (Mairal et al., 2011; Lecouat et al., 2020a;b).

The hierarchical nature of (1) introduces additional challenges compared to standard optimization problems, such as finding a suitable trade-off between the computational budget for approximating the inner and outer level problems (Ghadimi and Wang, 2018; Dempe and Zemkoho, 2020). These considerations are exacerbated in machine learning applications, where the costs $f$ and $g$ often come as an average of functions over a *large* or *infinite* number of data points (Franceschi et al., 2018). All these challenges highlight the need for methods that are able to control the computational costs inherent to (1) while dealing with the large-scale setting encountered in machine learning.

Gradient-based bilevel optimization methods appear to be viable approaches for solving (1) in large-scale settings (Lorraine et al., 2020). They can be divided into two categories: Iterative differentiation (ITD) and Approximate implicit differentiation (AID). *ITD approaches* approximate the map $y^\star(x)$ by a differentiable optimization algorithm $\mathcal{A}(x)$ viewed as a function of $x$. The resulting surrogate loss $\tilde{\mathcal{L}}(x) = f(x, \mathcal{A}(x))$ is optimized instead of $\mathcal{L}(x)$ using reverse-mode automatic differentiation (see Baydin et al., 2018). *AID approaches* (Pedregosa, 2016) rely on an expression of the gradient $\nabla \mathcal{L}$ resulting from the *implicit function theorem* (Lang, 2012, Theorem 5.9). Unlike ITD, AID avoids differentiating the algorithm approximating $y^\star(x)$ and, instead, approximately

solves a linear system using only Hessian and Jacobian-vector products to estimate the gradient $\nabla\mathcal{L}$ (Rajeswaran et al., 2019). These methods can also rely on stochastic approximation to increase scalability (Franceschi et al., 2018; Grazzi et al., 2020; 2021).

In the context of machine-learning, Ghadimi and Wang (2018) provided one of the first comprehensive studies of the computational complexity for a class of bilevel algorithms based on AID approaches. Subsequently, Hong et al. (2020b); Ji et al. (2021); Ji and Liang (2021); Yang et al. (2021) proposed different algorithms for solving (1) and obtained improved overall complexity by achieving a better trade-off between the cost of the inner and outer level problems. Still, the question of whether these complexities can be improved by better exploiting the structure of (1) through heuristics such as *warm-start* remains open (Grazzi et al., 2020). Moreover, these studies proposed separate analysis of their algorithms depending on the convexity of the loss $\mathcal{L}$ and whether a *stochastic* or *deterministic* setting is considered. This points out to a lack of unified and systematic theoretical framework for analyzing bilevel problems, which is what the present work addresses.

We consider the **Am**ortized **I**mplicit **G**radient **O**ptimization (AmIGO) algorithm, a bilevel optimization algorithm based on Approximate Implicit Differentiation (AID) approaches that exploits a warm-start strategy when estimating the gradient of $\mathcal{L}$. We then propose a unified theoretical framework for analyzing the convergence of AmIGO when the inner-level problem is strongly convex in both stochastic and deterministic settings. The proposed framework is inspired from the early work of Habets (1974) on *singularly perturbed systems* and analyzes the effect of warm start by viewing the iterates of AmIGO algorithm as a dynamical system. The evolution of such system is described by a total energy function which allows to recover the convergence rates of *unbiased oracle methods* which have access to an unbiased estimate of $\nabla\mathcal{L}$ (c.f. Table 1). To the best of our knowledge, this is the first time a bilevel optimization algorithm based on a warm-start strategy provably recovers the rates of *unbiased oracle methods* across a wide range of settings including the stochastic ones.

## 2 Related work

**Singularly perturbed systems (SPS)** are continuous-time deterministic dynamical systems of coupled variables $(x(t), y(t))$ with two time-scales where $y(t)$ evolves much faster than $x(t)$. As such, they exhibit a hierarchical structure similar to (1). The early work of Habets (1974); Saberi and Khalil (1984) provided convergence rates for SPS towards equilibria by studying the evolution of a single scalar energy function summarizing these systems. The present work takes inspiration from these works to analyze the convergence of AmIGO which involves three time-scales.

**Two time-scale Stochastic Approximation (TTSA)** can be viewed as a discrete-time stochastic version of SPS. (Kaledin et al., 2020) showed that TTSA achieves a finite-time complexity of $O(\epsilon^{-1})$ for linear systems while Doan (2020) obtained a complexity of $O(\epsilon^{-3/2})$ for general non-linear systems by extending the analysis for SPS. Hong et al. (2020b) further adapted the non-linear TTSA for solving (1). In the present work, we obtain faster rates by taking into account the dynamics of a third variable $z_k$ appearing in AmIGO, thus resulting in a three time-scale dynamics.

**Warm-start in bilevel optimization.** Ji et al. (2021); Ji and Liang (2021) used a warm-start for the inner-level algorithm to obtain an improved computational complexity over algorithms without warm-start. In the deterministic non-convex setting, Ji et al. (2021) used a warm-start strategy when solving the linear system appearing in AID approaches to obtain improved convergence rates. However, it remained open whether using a warm-start when solving both inner-level problem and linear system arising in AID approaches can yield faster algorithms in the more challenging stochastic setting (Grazzi et al., 2020). In the present work, we provide a positive answer to this question.

## 3 Amortized Implicit Gradient Optimization

### 3.1 General setting and main assumptions

**Notations.** In all what follows, $\mathcal{X}$ and $\mathcal{Y}$ are Euclidean spaces. For a differentiable function $h(x, y) : \mathcal{X} \times \mathcal{Y} \to \mathbb{R}$, we denote by $\nabla h$ its gradient w.r.t. $(x, y)$, by $\partial_x h$ and $\partial_y h$ its partial derivatives w.r.t. $x$ and $y$ and by $\partial_{xy} h$ and $\partial_{yy} h$ the partial derivatives of $\partial_y h$ w.r.t $x$ and $y$, respectively.

| Geometries | Setting | Algorithms | Complexity |
|---|---|---|---|
| (SC) | (D) | BA (Ghadimi and Wang, 2018) | $O(\kappa_{\mathcal{L}}^2 \vee \kappa_g^2 \log^2 \epsilon^{-1})$ |
| | | AccBio (Ji and Liang, 2021) | $O(\kappa_{\mathcal{L}}^{1/2} \kappa_g^{1/2} \log^2 \epsilon^{-1})$ |
| | | AmIGO (Corollary 1) | $O(\kappa_{\mathcal{L}} \kappa_g \log \epsilon^{-1})$ |
| | (S) | BSA (Ghadimi and Wang, 2018) | $O(\kappa_{\mathcal{L}}^4 \epsilon^{-2})$ |
| | | TTSA (Hong et al., 2020b) | $O(\kappa_{\mathcal{L}}^{0.5} (\kappa_g^{8.5} + \kappa_{\mathcal{L}}^3) \epsilon^{-3/2} \log \epsilon^{-1})$ |
| | | AmIGO (Corollary 2) | $O(\kappa_{\mathcal{L}}^2 \kappa_g^3 \epsilon^{-1} \log \epsilon^{-1})$ |
| (NC) | (D) | BA (Ghadimi and Wang, 2018) | $O(\kappa_g^5 \epsilon^{-5/4})$ |
| | | AID-BiO (Ji et al., 2021) | $O(\kappa_g^4 \epsilon^{-1})$ |
| | | AmIGO (Corollary 3) | $O(\kappa_g^4 \epsilon^{-1})$ |
| | (S) | BSA (Ghadimi and Wang, 2018) | $O(\kappa_g^9 \epsilon^{-3} + \kappa_g^6 \epsilon^{-2})$ |
| | | TTSA (Hong et al., 2020b) | $O(\kappa_g^{16} \epsilon^{-5/2} \log \epsilon^{-1})$ |
| | | stocBiO (Ji et al., 2021) | $O(\kappa_g^9 \epsilon^{-2} + \kappa_g^6 \epsilon^{-2} \log \epsilon^{-1})$ |
| | | MRBO/VRBO$^\star$ (Yang et al., 2021) | $O(\text{poly}(\kappa_g) \epsilon^{-3/2} \log \epsilon^{-1})$ |
| | | AmIGO (Corollary 4) | $O(\kappa_g^9 \epsilon^{-2})$ |

Table 1: Cost of finding an $\epsilon$-accurate solution as measured by $\mathbb{E}[\mathcal{L}(x_k) - \mathcal{L}^\star] \wedge 2^{-1} \mu \mathbb{E}\left[\|x_k - x^\star\|^2\right]$ when $\mathcal{L}$ is $\mu$-strongly-convex (**SC**) and $\frac{1}{k} \sum_{i=1}^{k} \mathbb{E}\left[\|\nabla \mathcal{L}(x_i)\|^2\right]$ when $\mathcal{L}$ is non-convex (**NC**). The settings **(D)** and **(S)** stand for the *deterministic* and *stochastic* settings. The cost corresponds to the total number of gradients, Jacobian and Hessian-vector products used by the algorithm. $\kappa_{\mathcal{L}}$ and $\kappa_g$ are the conditioning numbers of $\mathcal{L}$ and $g$ whenever applicable. The dependence on $\kappa_L$ and $\kappa_g$ for TTSA and AccBio are derived in Proposition 11 of Appendix A.4. The rate of MRBO/VRBO is obtained under the additional *mean-squared smoothness* assumption (Arjevani et al., 2019).

To ensure that (1) is well-defined, we consider the setting where the inner-level problem is strongly convex so that the solution $y^\star(x)$ is unique as stated by the following assumption:

**Assumption 1.** *For any $x \in \mathcal{X}$, the function $y \mapsto g(x, y)$ is $L_g$-smooth and $\mu_g$-strongly convex.*

Assumption 1 holds in the context of hyper-parameter selection when the inner-level is a kernel regression problem (Franceschi et al., 2018), or when the variable $y$ represents the last *linear* layer of a neural network as in many meta-learning tasks (Ji et al., 2021). Under Assumption 1 and additional smoothness assumptions on $f$ and $g$, the next proposition shows that $\mathcal{L}$ is differentiable:

**Proposition 1.** *Let $g$ be a twice differentiable function satisfying Assumption 1. Assume that $f$ is differentiable and consider the quadratic problem:*

$$\min_{z \in \mathbb{R}^{d_y}} Q(x, y, z) := \frac{1}{2} z^\top (\partial_{yy} g(x, y)) z + z^\top \partial_y f(x, y). \tag{2}$$

*Then, (2) admits a unique minimizer $z^\star(x, y)$ for any $(x, y)$ in $\mathcal{X} \times \mathcal{Y}$. Moreover, $y^\star(x)$ is unique and well-defined for any $x$ in $\mathcal{X}$ and $\mathcal{L}$ is differentiable with gradient given by:*

$$\nabla \mathcal{L}(x) = \partial_x f(x, y^\star(x)) + \partial_{xy} g(x, y^\star(x)) z^\star(x, y^\star(x)). \tag{3}$$

Proposition 1 follows by application of the *implicit function theorem* (Lang, 2012, Theorem 5.9) and provides an expression for $\nabla \mathcal{L}$ solely in terms of partial derivatives of $f$ and $g$ evaluated at $(x, y^\star(x))$. Following Ghadimi and Wang (2018), we further make two smoothness assumptions on $f$ and $g$:

**Assumption 2.** *There exist positive constants $L_f$ and $B$ such that for all $x, x' \in \mathcal{X}$ and $y, y' \in \mathcal{Y}$:*

$$\|\nabla f(x, y) - \nabla f(x', y')\| \leq L_f \|(x, y) - (x', y')\|, \qquad \|\partial_y f(x, y)\| \leq B.$$

**Assumption 3.** *There exit positive constants $L_g'$, $M_g$ such that for any $x, x' \in \mathcal{X}$ and $y, y' \in \mathcal{Y}$:*

$$\max \{\|\partial_{xy} g(x, y) - \partial_{xy} g(x', y')\|, \|\partial_{yy} g(x, y) - \partial_{yy} g(x', y')\|\} \leq M_g \|(x, y) - (x', y')\|$$
$$\|\partial_y g(x, y) - \partial_y g(x', y)\| \leq L_g' \|x - x'\|.$$

Assumptions 1 to 3 allow a control of the variations of $y^\star$ and $z^\star$ and ensure $\mathcal{L}$ is $L$-smooth for some positive constant $L$ as shown in Proposition 6 of Appendix B.2. As an $L$-smooth function, $\mathcal{L}$ is necessarily weakly convex (Davis et al., 2018), meaning that $\mathcal{L}$ satisfies the inequality $\mathcal{L}(x) - \mathcal{L}(y) \leq \nabla \mathcal{L}(x)^\top (x - y) - \frac{\mu}{2}\|x - y\|^2$ for some fixed $\mu \in \mathbb{R}$ with $|\mu| \leq L$. In particular, $\mathcal{L}$ is convex when $\mu \geq 0$, strongly convex when $\mu > 0$ and generally non-convex when $\mu < 0$. We thus consider two cases for $\mathcal{L}$, the *strongly convex* case ($\mu > 0$) and the *non-convex* case ($\mu < 0$). When $\mathcal{L}$ is convex, we denote by $\mathcal{L}^\star$ its minimum value achieved at a point $x^\star$ and define $\kappa_\mathcal{L} = L/\mu$ when $\mu > 0$.

**Stochastic/deterministic settings.** We consider the general setting where $f(x, y)$ and $g(x, y)$ are expressed as an expectation of stochastic functions $\hat{f}(x, y, \xi)$ and $\hat{g}(x, y, \xi)$ over a noise variable $\xi$. We recover the deterministic setting as a particular case when the variable $\xi$ has zero variance, thus allowing us to treat both *stochastic* (**S**) and *deterministic* (**D**) settings in a unified framework. As often in machine-learning, we assume we can always draw a new batch $\mathcal{D}$ of i.i.d. samples of the noise variable $\xi$ with size $|\mathcal{D}| \geq 1$ and use it to compute stochastic approximations of $f$ and $g$ defined by abuse of notation as $\hat{f}(x, y, \mathcal{D}) := \frac{1}{|\mathcal{D}|} \sum_{\xi \in \mathcal{D}} \hat{f}(x, y, \xi)$ and $\hat{g}(x, y, \mathcal{D}) := \frac{1}{|\mathcal{D}|} \sum_{\xi \in \mathcal{D}} \hat{g}(x, y, \xi)$. We make the following noise assumptions which are implied by those in Ghadimi and Wang (2018):

**Assumption 4.** *For any batch $\mathcal{D}$, $\nabla \hat{f}(x, y, \mathcal{D})$ and $\partial_y \hat{g}(x, y, \mathcal{D})$ are unbiased estimator of $\nabla f(x, y)$ and $\partial_y g(x, y)$ with a uniformly bounded variance, i.e. for all $x, y \in \mathcal{X} \times \mathcal{Y}$:*

$$\mathbb{E}\left[\left\|\nabla \hat{f}(x, y, \mathcal{D}) - \nabla f(x, y)\right\|^2\right] \leq \tilde{\sigma}_f^2 |\mathcal{D}|^{-1}, \qquad \mathbb{E}\left[\left\|\partial_y \hat{g}(x, y, \mathcal{D}) - \partial_y g(x, y)\right\|^2\right] \leq \tilde{\sigma}_g^2 |\mathcal{D}|^{-1}.$$

**Assumption 5.** *For any batch $\mathcal{D}$, the matrices $F_1(x, y, \mathcal{D}) := \partial_{xy}\hat{g}(x, y, \mathcal{D}) - \partial_{xy}g(x, y)$ and $F_2(x, y, \mathcal{D}) := \partial_{yy}\hat{g}(x, y, \mathcal{D}) - \partial_{yy}g(x, y)$ have zero mean and satisfy for all $x, y \in \mathcal{X} \times \mathcal{Y}$:*

$$\left\|\mathbb{E}\left[F_1(x, y, \mathcal{D})^\top F_1(x, y, \mathcal{D})\right]\right\|_{op} \leq \tilde{\sigma}_{g_{xy}}^2 |\mathcal{D}|^{-1}, \quad \left\|\mathbb{E}\left[F_2(x, y, \mathcal{D})^\top F_2(x, y, \mathcal{D})\right]\right\|_{op} \leq \tilde{\sigma}_{g_{yy}}^2 |\mathcal{D}|^{-1}.$$

For conciseness, we will use the notations $\sigma_f^2 := \tilde{\sigma}_f^2 |\mathcal{D}|^{-1}$, $\sigma_g^2 := \tilde{\sigma}_g^2 |\mathcal{D}|^{-1}$, $\sigma_{g_{xy}}^2 := \tilde{\sigma}_{g_{xy}}^2 |\mathcal{D}|^{-1}$ and $\sigma_{g_{yy}}^2 := \tilde{\sigma}_{g_{yy}}^2 |\mathcal{D}|^{-1}$, without explicit reference to the batch $\mathcal{D}$. Next, we describe the algorithm.

## 3.2 ALGORITHMS

Amortized Implicit Gradient Optimization (AmIGO) is an iterative algorithm for solving (1). It constructs iterates $x_k$, $y_k$ and $z_k$ such that $x_k$ approaches a stationary point of $\mathcal{L}$ while $y_k$ and $z_k$ track the quantities $y^\star(x_k)$ and $z^\star(x_k, y_k)$. AmIGO computes the iterate $x_{k+1}$ using an update equation $x_{k+1} = x_k - \gamma_k \hat{\psi}_k$ for some given step-size $\gamma_k$ and a stochastic estimate $\hat{\psi}_k$ of $\nabla \mathcal{L}(x_k)$ based on (3) and defined according to (4) below for some new batches of samples $\mathcal{D}_f$ and $\mathcal{D}_{g_{xy}}$.

$$\hat{\psi}_k := \partial_x \hat{f}(x_k, y_k, \mathcal{D}_f) + \partial_{x,y}\hat{g}(x_k, y_k, \mathcal{D}_{g_{xy}})^\top z_k. \tag{4}$$

**Algorithm 1** AmIGO

1: Inputs: $x_0, y_{-1}, z_{-1}$.
2: Parameters: $\gamma_k, K$.
3: **for** $k \in \{0, ..., K\}$ **do**
4: $\quad y_k \leftarrow \mathcal{A}_k(x_k, y_{k-1})$
5: $\quad$ Sample batches $\mathcal{D}_f, \mathcal{D}_g$.
6: $\quad (u_k, v_k) \leftarrow \nabla \hat{f}(x_k, y_k, \mathcal{D}_f)$.
7: $\quad z_k \leftarrow \mathcal{B}_k(x_k, y_k, v_k, z_{k-1})$
8: $\quad w_k \leftarrow \partial_{xy}\hat{g}(x_k, y_k, \mathcal{D}_{g_{xy}})z_k$
9: $\quad \hat{\psi}_{k-1} \leftarrow u_k + w_k$
10: $\quad x_k \leftarrow x_{k-1} - \gamma_k \hat{\psi}_{k-1}$
11: **end for**
12: Return $x_K$.

AmIGO computes $\hat{\psi}_k$ in 4 steps given iterates $x_k$, $y_{k-1}$ and $z_{k-1}$. A first step computes an approximation $y_k$ to $y^\star(x_k)$ using a stochastic algorithm $\mathcal{A}_k$ *initialized* at $y_{k-1}$. A second step computes unbiased estimates $u_k = \partial_x \hat{f}(x_k, y_k, \mathcal{D}_f)$ and $v_k = \partial_y \hat{f}(x_k, y_k, \mathcal{D}_f)$ of the partial derivatives of $f$ w.r.t. $x$ and $y$. A third step computes an approximation $z_k$ to $z^\star(x_k, y_k)$ using a second stochastic algorithm $\mathcal{B}_k$ for solving (2) *initialized* at $z_{k-1}$. To increase efficiency, algorithm $\mathcal{B}_k$ uses the pre-computed vector $v_k$ for approximating the partial derivative $\partial_y f$ in (2). Finally, the stochastic estimate $\hat{\psi}_k$ is computed using (4) by summing the pre-computed vector $u_k$ with the jacobian-vector product $w_k = \partial_{xy}\hat{g}(x_k, y_k, \mathcal{D}_{g_{xy}})z_k$. AmIGO is summarized in Algorithm 1.

**Algorithms $\mathcal{A}_k$ and $\mathcal{B}_k$.** While various choices for $\mathcal{A}_k$ and $\mathcal{B}_k$ are possible, such as adaptive algorithms (Kingma and Ba, 2015), or accelerated stochastic algorithms (Ghadimi and Lan, 2012), we

focus on simple stochastic gradient descent algorithms with a pre-defined number of iterations $T$ and $N$. These algorithms compute intermediate iterates $y^t$ and $z^n$ optimizing the functions $y \mapsto g(x_k, y)$ and $z \mapsto Q(x_k, y_k, z)$ starting from some initial values $y^0$ and $z^0$ and returning the last iterates $y^T$ and $z^N$ as described in Algorithms 2 and 3. Algorithm $\mathcal{A}_k$ updates the current iterate $y^{t-1}$ using a stochastic gradient $\partial_y \hat{g}(x_k, y^{t-1}, \mathcal{D}_g)$ for some new batch of samples $\mathcal{D}_g$ and a fixed step-size $\alpha_k$. Algorithm $\mathcal{B}_k$ updates the current iterate $z^{t-1}$ using a stochastic estimate of $\partial_z Q(x_k, y_k, z^{t-1})$ with step-size $\beta_k$. The stochastic gradient is computed by evaluating the Hessian-vector product $\partial_{yy} \hat{g}(x_k, y_k, \mathcal{D}_{g_{yy}}) z^{t-1}$ for some new batch of samples $\mathcal{D}_{g_{yy}}$ and summing it with a vector $v_k$ approximating $\partial_y f(x_k, y_k)$ provided as input to algorithm $\mathcal{B}_k$.

**Warm-start for $y^0$ and $z^0$.** Following the intuition that $y^\star(x_k)$ remains close to $y^\star(x_{k-1})$ when $x_k \simeq x_{k-1}$, and assuming that $y_{k-1}$ is an accurate approximation to $y^\star(x_{k-1})$, it is natural to initialize $\mathcal{A}_k$ with the iterate $y_{k-1}$. The same intuition applies when initializing $\mathcal{B}_k$ with $z_{k-1}$. Next, we introduce a framework for analyzing the effect of warm-start on the convergence speed of AmIGO.

| **Algorithm 2** $\mathcal{A}_k(x, y^0)$ | **Algorithm 3** $\mathcal{B}_k(x, y, v, z^0)$ |
|---|---|
| 1: Parameters: $\alpha_k, T$ | 1: Parameters: $\beta_k, N$. |
| 2: **for** $t \in \{1, ..., T\}$ **do** | 2: **for** $n \in \{1, ..., N\}$ **do** |
| 3:     Sample batch $\mathcal{D}_{t,k}^g$. | 3:     Sample batch $\mathcal{D}_{n,k}^{g_{yy}}$. |
| 4:     $y^t \leftarrow y^{t-1} - \alpha_k \partial_y \hat{g}\left(x, y^{t-1}, \mathcal{D}_{t,k}^g\right)$. | 4:     $z^n \leftarrow z^{n-1} - \beta_k\left(\partial_{yy}\hat{g}\left(x, y, \mathcal{D}_{n,k}^{g_{yy}}\right)z^{n-1} + v\right)$. |
| 5: **end for** | 5: **end for** |
| 6: Return $y^T$. | 6: Return $z^N$. |

## 4 ANALYSIS OF AMORTIZED IMPLICIT GRADIENT OPTIMIZATION

### 4.1 GENERAL APPROACH AND MAIN RESULT

The proposed approach consists in three main steps: (1) Analysis of the outer-level problem , (2) Analysis of the inner-level problem and (3) Analysis of the joint dynamics of both levels.

**Outer-level problem.** We consider a quantity $E_k^x$ describing the evolution of $x_k$ defined as follows:

$$E_k^x := \begin{cases} \frac{\delta_k}{2\gamma_k} \mathbb{E}\left[\|x_k - x^\star\|^2\right] + (1-u)\mathbb{E}[\mathcal{L}(x_k) - \mathcal{L}^\star], & \mu \geq 0 \\ \frac{\delta_k}{2\gamma_k L^2} \mathbb{E}\left[\|\nabla\mathcal{L}(x_k)\|^2\right], & \mu < 0. \end{cases} \tag{5}$$

where $u \in \{0, 1\}$ is set to 1 in the *stochastic* setting and to 0 in the *deterministic* one and $\delta_k$ is a positive sequence that determines the convergence rate of the outer-level problem and is defined by:

$$\delta_k := \eta_k \gamma_k, \qquad \eta_{k+1} := \eta_k(1 + \gamma_{k+1}(\eta_k - \mu))\mathbb{1}_{\mu \geq 0} + L\mathbb{1}_{\mu < 0}.$$

with $\eta_0$ such that $\gamma_0^{-1} \geq \eta_0 \geq \mu$ if $\mu \geq 0$ and $\eta_0 = L$ if $\mu < 0$ and where we choose the step-size $\gamma_k$ to be a non-increasing sequence with $\gamma_0 \leq \frac{1}{L}$. With this choice for $\delta_k$ and by setting $u = 1$ in (5), $E_k^x$ recovers the quantity considered in the *stochastic estimate sequences* framework of Kulunchakov and Mairal (2020) to analyze the convergence of stochastic optimization algorithms when $\mathcal{L}$ is convex. When $\mathcal{L}$ is non-convex, $E_k^x$ recovers a standard measure of stationarity (Davis and Drusvyatskiy, 2018). In Section 4.3, we control $E_k^x$ using bias and variance error $E_{k-1}^\psi$ and $V_{k-1}^\psi$ of $\hat{\psi}_k$ given by (6) below where $\mathbb{E}_k$ denotes expectation conditioned on $(x_k, y_k, z_{k-1})$.

$$E_k^\psi := \mathbb{E}\left[\left\|\mathbb{E}_k\left[\hat{\psi}_k\right] - \nabla\mathcal{L}(x_k)\right\|^2\right], \qquad V_k^\psi := \mathbb{E}\left[\left\|\hat{\psi}_k - \mathbb{E}_k\left[\hat{\psi}_k\right]\right\|^2\right]. \tag{6}$$

**Inner-level problems.** We consider the mean-squared errors $E_k^y$ and $E_k^z$ between initializations ($y^0 = y_{k-1}$ and $z^0 = z_{k-1}$) and stationary values ($y^\star(x_k)$ and $z^\star(x_k, y_k)$) of algorithms $\mathcal{A}_k$ and $\mathcal{B}_k$:

$$E_k^y := \mathbb{E}\left[\|y_k^0 - y^\star(x_k)\|^2\right], \qquad E_k^z := \mathbb{E}\left[\|z_k^0 - z^\star(x_k, y_k)\|^2\right].$$

In Section 4.3, we show that the *warm-start strategy* allows to control $E_k^y$ and $E_k^z$ in terms of previous iterates $E_{k-1}^y$ and $E_{k-1}^z$ as well as the bias and variance errors in (6). We further prove that such bias and variance errors are, in turn, controlled by $E_k^y$ and $E_k^z$.

**Joint dynamics.** Following Habets (1974), we consider an aggregate error $E_k^{tot}$ defined as a linear combination of $E_k^x$, $E_k^y$ and $E_k^z$ with carefully selected coefficients $a_k$ and $b_k$:

$$E_k^{tot} = E_k^x + a_k E_k^y + b_k E_k^z. \tag{7}$$

As such $E_k^{tot}$ represents the dynamics of the whole system. The following theorem provides an error bound for $E_k^{tot}$ in both convex and non-convex settings for a suitable choice of the coefficients $a_k$ and $b_k$ provided that $T$ and $N$ are large enough:

**Theorem 1.** *Choose a batch-size* $\left|\mathcal{D}_{g_{yy}}\right| \geq 1 \vee \frac{\tilde{\sigma}_{g_{yy}}^2}{\mu_g L_g}$ *and the step-sizes* $\alpha_k = L_g^{-1}$, $\beta_k = (2L_g)^{-1}$, $\gamma_k = L^{-1}$. *Set the coefficients* $a_k$ *and* $b_k$ *to be* $a_k := \delta_0 (1 - \alpha_k \mu_g)^{1/2}$ *and* $b_k := \delta_0 \left(1 - \frac{1}{2} \beta_k \mu_g\right)^{1/2}$ *and set the number of iterations* $T$ *and* $N$ *of Algorithms 2 and 3 to be of order* $T = O(\kappa_g)$ *and* $N = O(\kappa_g)$ *up to a logarithmic dependence on* $\kappa_g$. *Let* $\hat{x}_k = u(1 - \delta_k)\hat{x}_{k-1} + (1 - u(1 - \delta_k))x_k$, *with* $\hat{x}_0 = x_0$. *Then, under Assumptions 1 to 5,* $E_k^{tot}$ *satisfies:*

$$
\begin{cases}
\mathbb{E}[\mathcal{L}(\hat{x}_k) - \mathcal{L}^\star] + E_k^{tot} \leq \left(1 - (2\kappa_\mathcal{L})^{-1}\right)^k \left[E_0^{tot} + \mathbb{E}[\mathcal{L}(x_0) - \mathcal{L}^\star]\right] + \frac{2\mathcal{W}^2}{L}, & \mu \geq 0 \\[2mm]
\frac{1}{k}\sum_{t=1}^{k} E_t^{tot} \leq \frac{2}{k}(\mathbb{E}[\mathcal{L}(x_0) - \mathcal{L}^\star] + E_0^y + E_0^z) + \frac{2\mathcal{W}^2}{L}, & \mu < 0,
\end{cases}
$$

*where* $\mathcal{W}^2$, *defined in (20) of Appendix A.2, is the* **effective variance** *of the problem with* $\mathcal{W}^2 = 0$ *in the deterministic setting and, in the stochastic setting,* $\mathcal{W}^2 > 0$ *is of the following order:*

$$\mathcal{W}^2 = O\left(\delta_0^{-1}\kappa_g^5 |\mathcal{D}_g|^{-1}\tilde{\sigma}_g^2 + \kappa_g^3 \left|\mathcal{D}_{g_{yy}}\right|^{-1}\tilde{\sigma}_{g_{yy}}^2 + \kappa_g^2 \left|\mathcal{D}_{g_{xy}}\right|^{-1}\tilde{\sigma}_{g_{xy}}^2 + \kappa_g^2 |\mathcal{D}_f|^{-1}\tilde{\sigma}_f^2\right),$$

We describe the strategy of the proof in Section 4.3 and provide a proof outline in Appendix A.1 with exact expressions for all variables including the expressions of $T$, $N$ and $\mathcal{W}^2$. The full proof is provided in Appendix A.2. The choice of $a_k$ and $b_k$ ensures that $E_k^y$ and $E_k^z$ contribute less to $E_k^{tot}$ as the algorithms $\mathcal{A}_k$ and $\mathcal{B}_k$ become more accurate. The *effective variance* $\mathcal{W}^2$ accounts for interactions between both levels in the presence of noise and becomes proportional to the outer-level variance $\sigma_f^2$ when the inner-level problem is solved exactly. In the deterministic setting, all variances $\tilde{\sigma}_f^2$, $\tilde{\sigma}_g^2$, $\tilde{\sigma}_{g_{xy}}^2$ and $\tilde{\sigma}_{g_{yy}}^2$ vanish so that $\mathcal{W}^2 = 0$. Hence, we characterize such setting by $\mathcal{W}^2 = 0$ and the stochastic one by $\mathcal{W}^2 > 0$. Next, we apply Theorem 1 to obtain the complexity of AmIGO.

## 4.2 COMPLEXITY ANALYSIS

We define the *complexity* $\mathcal{C}(\epsilon)$ of a bilevel algorithm to be the total number of queries to the gradients of $f$ and $g$, Jacobian/hessian-vector products needed by the algorithm to achieve an error $\epsilon$ according to some pre-defined criterion. Let the number of iterations $k$, $T$ and $N$ and sizes of the batches $|\mathcal{D}_g|$, $|\mathcal{D}_f|$, $\left|\mathcal{D}_{g_{xy}}\right|$ and $\left|\mathcal{D}_{g_{yy}}\right|$, be such that AmIGO achieves a precision $\epsilon$. Then $\mathcal{C}(\epsilon)$ is given by:

$$\mathcal{C}(\epsilon) = k\left(T|\mathcal{D}_g| + N\left|\mathcal{D}_{g_{yy}}\right| + \left|\mathcal{D}_{g_{xy}}\right| + |\mathcal{D}_f|\right), \tag{8}$$

We provide the complexity of AmIGO in the 4 settings of Table 1 in the form of Corrolaries 1 to 4 .

**Corollary 1** (Case $\mu > 0$ and $\mathcal{W}^2 = 0$). *Use batches of size* 1. *Achieving* $\mathcal{L}(x_k) - \mathcal{L}^\star + \frac{\mu}{2}\|x_k - x^\star\|^2 \leq \epsilon$ *requires* $\mathcal{C}(\epsilon) = O\left(\kappa_\mathcal{L}\kappa_g \log\left(\frac{E_0^{tot}}{\epsilon}\right)\right)$.

Corollary 1 outperforms the complexities in Table 1 in terms of the dependence on $\epsilon$. It is possible to improve the dependence on $\kappa_g$ to $\kappa_g^{1/2}$ using acceleration in $\mathcal{A}_k$ and $\mathcal{B}_k$ as discussed in Appendix A.5.1, or using generic acceleration methods such as Catalyst (Lin et al., 2018).

**Corollary 2** (Case $\mu\mathcal{W}^2 > 0$). *Choose* $|\mathcal{D}_g| = \Theta\left(\epsilon^{-1}\kappa_\mathcal{L}\kappa_g^2\tilde{\sigma}_g^2\right)$, $\left|\mathcal{D}_{g_{xy}}\right| = \Theta\left(\epsilon^{-1}\tilde{\sigma}_{g_{xy}}^2\right)$, $|\mathcal{D}_f| = \Theta\left(\frac{\tilde{\sigma}_f^2}{\epsilon}\right)$ *and* $\left|\mathcal{D}_{g_{yy}}\right| = \Theta\left(\tilde{\sigma}_{g_{yy}}^2\left(\frac{1}{\epsilon}\vee\kappa_g\right)\right)$. *Achieving* $\mathbb{E}[\mathcal{L}(\hat{x}_k) - \mathcal{L}^\star] + \frac{\mu}{2}\mathbb{E}\left[\|x_k - x^\star\|^2\right] \leq \epsilon$ *requires:*

$$\mathcal{C}(\epsilon) = O\left(\kappa_\mathcal{L}\left(\kappa_\mathcal{L}\kappa_g^3\tilde{\sigma}_g^2 + \kappa_g(1 \vee \epsilon\kappa_g)\tilde{\sigma}_{g_{yy}}^2 + \tilde{\sigma}_{g_{xy}}^2 + \tilde{\sigma}_f^2\right)\frac{1}{\epsilon}\log\left(\frac{E_0^{tot} + \mathbb{E}[\mathcal{L}(x_0) - \mathcal{L}^\star]}{\epsilon}\right)\right).$$

Corollary 2 improves over the results in Table 1 in the stochastic strongly-convex setting and recovers the dependence on $\epsilon$ of stochastic gradient descent for smooth and strongly convex functions up to a logarithmic factor.

**Corollary 3** (Case $\mu<0$ and $\mathcal{W}^2=0$). *Choose batches of size 1. Achieving $\frac{1}{k}\sum_{i=1}^{k}\|\nabla\mathcal{L}(x_i)\|^2 \leq \epsilon$ requires $\mathcal{C}(\epsilon) = \mathcal{O}\left(\frac{\kappa_g^4}{\epsilon}((\mathcal{L}(x_0) - \mathcal{L}^\star) + E_0^y + E_0^z)\right)$.*

Corollary 3 recovers the complexity of AID-BiO (Ji et al., 2021) in the deterministic non-convex setting. This is expected since AID-BiO also exploits warm-start for both $\mathcal{A}_k$ and $\mathcal{B}_k$.

**Corollary 4** (Case $\mu<0$ and $\mathcal{W}>0$). *Choose $|\mathcal{D}_{g_{xy}}|=\Theta\left(\frac{\kappa_g^2}{\epsilon}\tilde{\sigma}_{g_{xy}}^2\right)$, $|\mathcal{D}_{g_{yy}}|=\Theta\left(\frac{\kappa_g^3}{\epsilon}\tilde{\sigma}_{g_{yy}}^2\left(1\vee\epsilon\mu_g^2\right)\right)$, $|\mathcal{D}_f|=\Theta\left(\frac{\kappa_g^2\tilde{\sigma}_f^2}{\epsilon}\right)$ and $|\mathcal{D}_g|=\Theta\left(\frac{\kappa_g^5\tilde{\sigma}_g^2}{\epsilon}\right)$. Achieving an error $\frac{1}{k}\sum_{i=1}^{k}\mathbb{E}\left[\|\nabla\mathcal{L}(x_i)\|^2\right] \leq \epsilon$ requires:*

$$\mathcal{C}(\epsilon) = O\left(\frac{\kappa_g^5}{\epsilon^2}\left(\kappa_g^4\tilde{\sigma}_g^2 + \kappa_g^2\left(1\vee\epsilon\mu_g^2\right)\tilde{\sigma}_{g_{yy}}^2 + \tilde{\sigma}_{g_{xy}}^2 + \tilde{\sigma}_f^2\right)\left(\mathbb{E}[\mathcal{L}(x_k) - \mathcal{L}^\star] + E_0^y + E_0^z\right)\right)$$

Corollary 4 recovers the optimal dependence on $\epsilon$ of $O(\frac{1}{\epsilon^2})$ achieved by stochastic gradient descent in the smooth non-convex case (Arjevani et al., 2019, Theorem 1). It also improves over the results in (Ji et al., 2021) which involve an additional logarithmic factor $\log(\epsilon^{-1})$ as $N$ is required to be $O(\kappa_g \log(\epsilon^{-1}))$. In our case, $N$ remains constant since $\mathcal{B}_k$ benefits from warm-start. The faster rates of MRBO/VRBO$^\star$ (Yang et al., 2021) are obtained under the additional *mean-squared smoothness* assumption (Arjevani et al., 2019), which we do not investigate in the present work. Such assumption allows to achieve the improved complexity of $O(\epsilon^{-3/2}\log(\epsilon^{-1}))$. However, these algorithms still require $N=O(\log(\epsilon^{-1}))$, indicating that the use of warm-start in $\mathcal{B}_k$ could further reduce the complexity to $O(\epsilon^{-3/2})$ which would be an interesting direction for future work.

### 4.3 OUTLINE OF THE PROOF

The proof of Theorem 1 proceeds by deriving a *recursion* for both outer-level error $E_k^x$ and inner-level errors $E_k^y$ and $E_k^z$ and then combining those to obtain an error bound on the total error $E_k^{tot}$.

**Outer-level recursion.** To allow a unified analysis of the behavior of $E_k^x$ in both convex and non-convex settings, we define $F_k$ as follows:

$$F_k := u\delta_k \mathbb{E}[\mathcal{L}(x_k) - \mathcal{L}^\star]\mathbb{1}_{\mu>0} + \left(\mathbb{E}[\mathcal{L}(x_k) - \mathcal{L}(x_{k-1})] + E_{k-1}^x - E_k^x\right)\mathbb{1}_{\mu<0}.$$

The following proposition, with a proof in Appendix C.1, provides a recursive inequality on $E_k^x$ involving the errors in (6) due to the inexact gradient $\hat{\psi}_k$:

**Proposition 2.** *Let $\rho_k$ be a non-increasing sequence with $0<\rho_k<2$. Assumptions 1 to 3 ensure that:*

$$F_k + E_k^x \leq \left(1 - \left(1 - 2^{-1}\rho_k\right)\delta_k\right)E_{k-1}^x + \gamma_k s_k V_{k-1}^\psi + \gamma_k\left(s_k + \rho_k^{-1}\right)E_{k-1}^\psi, \qquad (10)$$

*with $s_k$ defined as $s_k := \frac{1}{2}\delta_k + \left(\frac{u}{2}\delta_k + (1-u)\right)\mathbb{1}_{\mu>0}$.*

In the ideal case where $y_k = y^\star(x_k)$ and $z_k = z^\star(x_k, y_k)$, the bias $E_k^\psi$ vanishes and (10) simplifies to (Kulunchakov and Mairal, 2019, Proposition 1) which recovers the convergence rates for stochastic gradient methods in the convex case. However, $y_k$ and $z_k$ are generally inexact solutions and introduce a positive bias $E_k^\psi$. Therefore, controlling the inner-level iterates is required to control the bias $E_k^\psi$ which, in turn, impacts the convergence of the outer-level as we discuss next.

**Controlling the inner-level iterates $y_k$ and $z_k$.** Proposition 3 below controls the expected mean squared errors between iterates $y_k$ and $z_k$ and their limiting values $y^\star(x_k)$ and $z^\star(x_k, y_k)$:

**Proposition 3.** *Let the step-sizes $\alpha_k$ and $\beta_k$ be such that $\alpha_k \leq L_g^{-1}$ and $\beta_k \leq \frac{1}{2L_g} \wedge \frac{\mu_g}{\mu_g^2 + \sigma_{g_{yy}}^2}$. Let $\Lambda_k := (1-\alpha_k\mu_g)^T$ and $\Pi_k := \left(1 - \frac{\beta_k\mu_g}{2}\right)^N$. Under Assumptions 1, 4 and 5, it holds that:*

$$\mathbb{E}\left[\|y_k - y^\star(x_k)\|^2\right] \leq \Lambda_k E_k^y + R_k^y, \qquad \mathbb{E}\left[\|z_k - z^\star(x_k,y_k)\|^2\right] \leq \Pi_k E_k^z + R_k^z, \qquad (11)$$

*where $R_k^y = O\left(\kappa_g\sigma_g^2\right)$ and $R_k^z = O\left(\kappa_g^3\sigma_{g_{yy}}^2 + \kappa_g^2\sigma_f^2\right)$ are defined in (14) of Appendix A.2.*

While Proposition 3 is specific to the choice of the algorithms $\mathcal{A}_k$ and $\mathcal{B}_k$ in Algorithms 2 and 3, our analysis directly extends to other algorithms satisfying inequalities similar to (11) such as *accelerated* or *variance reduced* algorithms discussed in Appendices A.5.1 and A.5.2. Proposition 4 below controls the bias and variance terms $V_k^\psi$ and $E_k^\psi$ in terms of the warm-start error $E_k^y$ and $E_k^z$.

**Proposition 4.** *Under Assumptions 1 to 5, the following inequalities hold:*

$$E_k^\psi \leq 2L_\psi^2(\Lambda_k E_k^y + \Pi_k E_k^z + R_k^y), \qquad V_k^\psi \leq w_x^2 + \sigma_x^2 \Pi_k E_k^z,$$

*where $w_x^2 = O\left(\kappa_g^2\left(\sigma_f^2 + \sigma_{g_{xy}}^2\right) + \kappa_g^3 \sigma_{g_{yy}}^2\right)$, $\sigma_x^2 = O\left(\sigma_{g_{xy}}^2 + \kappa_g^2 \sigma_{g_{yy}}^2\right)$ and $L_\psi = O(\kappa_g^2)$ are positive constants defined in (13) and (16) of Appendix A.2 with $L_\psi$ controlling the variations of $\mathbb{E}_k[\hat{\psi}_k]$.*

Proposition 4 highlights the dependence of $E_k^\psi$ and $V_k^\psi$ on the inner-level errors. It suggests analyzing the evolution of $E_k^y$ and $E_k^z$ to quantify how large the bias and variances can get:

**Proposition 5.** *Let $\zeta_k > 0$, a $2 \times 2$ matrix $\boldsymbol{P_k}$, two vectors $\boldsymbol{U_k}$ and $\boldsymbol{V_k}$ in $\mathbb{R}^2$ all independent of $x_k$, $y_k$ and $z_k$ be as defined in Proposition 8 of Appendix A.2. Under Assumptions 1 to 5, it holds that:*

$$\begin{pmatrix} E_k^y \\ E_k^z \end{pmatrix} \leq \boldsymbol{P_k} \begin{pmatrix} \Lambda_{k-1} E_{k-1}^y + R_{k-1}^y \\ \Pi_{k-1} E_{k-1}^z + R_{k-1}^z \end{pmatrix} + 2\gamma_k \left( E_{k-1}^\psi + V_{k-1}^\psi + \zeta_k E_{k-1}^x \right) \boldsymbol{U_k} + \boldsymbol{V_k}. \qquad (12)$$

Proposition 5 describes the evolution of the inner-level errors as the number of iterations $k$ increases. The matrix $\boldsymbol{P_k}$ and vectors $\boldsymbol{U_k}$ and $\boldsymbol{V_k}$ arise from discretization errors and depend on the step-sizes and constants of the problem. The second term of (12) represents interactions with the outer-level through $E_{k-1}^x$, $V_{k-1}^\psi$ and $E_{k-1}^\psi$. Propositions 2, 4 and 5 describe the joint dynamics of $(E_k^x, E_k^y, E_k^z)$ from which the evolution of $E_k^{tot}$ can be deduced as shown in Appendices A.1 and A.2.

## 5 EXPERIMENTS

We run three sets of experiments described in Sections 5.1 to 5.3. In all cases, we consider AmIGO with either gradient descent (AmIGO-GD) or conjugate gradient (AmIGO-CG) for algorithm $\mathcal{B}_k$. We AmIGO with AID methods without warm-start for $\mathcal{B}_k$ which we refer to as (AID-GD) and (AID-CG) and with (AID-CG-WS) which uses warm-start for $\mathcal{B}_k$ but not for $\mathcal{A}_k$. We also consider other variants using either a fixed-point algorithm (AID-FP) (Grazzi et al., 2020) or Neumann series expansion (AID-N) (Lorraine et al., 2020) for $\mathcal{B}_k$. Finally, we consider two algorithms based on iterative differentiation which we refer to as (ITD) (Grazzi et al., 2020) and (Reverse) (Franceschi et al., 2017). For all methods except (AID-CG-WS), we use warm-start in algorithm $\mathcal{A}_k$, however only AmIGO, AmIGO-CG and AID-CG-WS exploits warm-start in $\mathcal{B}_k$ the other AID based methods initializing $\mathcal{B}_k$ with $z^0 = 0$. In Sections 5.2 and 5.3, we also compare with BSA algorithm (Ghadimi and Wang, 2018), TTSA algorithm (Hong et al., 2020a) and stocBiO (Ji et al., 2021). An implementation of AmIGO is available in `https://github.com/MichaelArbel/AmIGO`.

### 5.1 SYNTHETIC PROBLEM

To study the behavior of AmIGO in a controlled setting, we consider a synthetic problem where both inner and outer level losses are quadratic functions with thousands of variables as described in details in Appendix F.1. Figure 1(a) shows the complexity $\mathcal{C}(\epsilon)$ needed to reach $10^{-6}$ relative error amongst the best choice for $T$ and $M$ over a grid as the conditioning number $\kappa_g$ increases. AmIGO-CG achieves the lowest time and is followed by AID-CG thus showing a favorable effect of warm-start for $\mathcal{B}_k$. The same conclusion holds for AmIGO-GD compared to AID-GD. Note that AID-CG is still faster than AmIGO-CG for larger values of $\kappa_g$ highlighting the advantage of using algorithms $\mathcal{B}_k$ with $O(\sqrt{\kappa_g})$ complexity such as (CG) instead non-accelerated ones with $O(\kappa_g^{-1})$ such as (GD). Figure 1(b) shows the relative error after 10s and maintains the same conclusions. For moderate values of $\kappa_g$, only AmIGO and AID-CG reach an error of $10^{-20}$ as shown in Figure 1(c). We refer to Figures 2 and 3 of Appendix F for additional results on the effect of the choice of $T$ and $M$ showing that AmIGO consistently performs well for a wide range of values of $T$ and $M$.

### 5.2 HYPER-PARAMETER OPTIMIZATION

We consider a classification task on the `20Newsgroup` dataset using a logistic loss and a linear model. Each dimension of the linear model is regularized using a different hyper-parameter. The

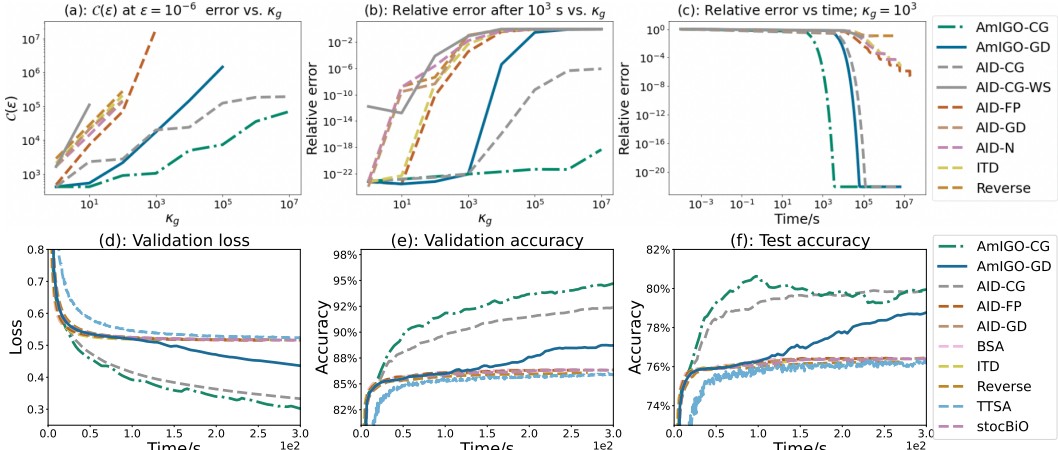

Figure 1: Top row: performance on the synthetic task. The relative error is defined as a ratio between current and initial errors $(\mathcal{L}(x_k) - \mathcal{L}^\star)/(\mathcal{L}(x_0) - \mathcal{L}^\star)$. The complexity $\mathcal{C}(\epsilon)$ as defined in (8). Bottom row: performance on the hyper-parameter optimization task.

collection of those hyper-parameters form a vector $x$ of dimension $d{=}101631$ optimized using an un-regularized regression loss over the validation set while the model is learned using the training set. We consider two evaluations settings: A *default* setting based on Grazzi et al. (2020); Ji et al. (2021) and a *grid-seach* search setting near the default values of $\beta_k$, $T$ and $N$ as detailed in Appendix F.2. We also vary the batch-size from $10^3 * \{0.1, 1, 2, 4\}$ and report the best performing choices for each method. Figure 1(d,e,f) show AmIGO-CG to be the fastest, achieving the lowest error and highest validation and test accuracies. The test accuracy of AmIGO-CG decreases after exceeding $80\%$ indicating a potential overfitting as also observed in Franceschi et al. (2018). Similarly, AmIGO-GD outperformed all other methods that uses an algorithm $\mathcal{B}_k$ with $O(\kappa_g)$ complexity. Moreover, all remaining methods achieved comparable performance matching those reported in Ji et al. (2021), thus indicating that the warm-start in $\mathcal{B}_k$ and acceleration in $\mathcal{B}_k$ were the determining factors for obtaining an improved performance. Additionally, Figure 4 of Appendix F report similar results for each choice of the batch-size indicating robustness to the choice of the batch-size.

## 5.3 DATASET DISTILLATION

Dataset distillation (Wang et al., 2018) consists in learning a synthetic dataset so that a model trained on this dataset achieves a small error on the training set. Figure 5 of Appendix F.3 shows the training loss (outer loss), the training and test accuracies of a model trained on MNIST by dataset distillation. Similarly to Figure 1, AmIGO-CG achieves the best performance followed by AID-CG. AmIGO obtains the best performance by far among methods without acceleration for $\mathcal{B}_k$ while all the remaining ones fail to improve. This finding is indicative of an ill-conditioned inner-level problem as confirmed when computing the conditioning number of the hessian $\partial_{yy}g(x,y)$ which we found to be of order $7{\times}10^4$. Indeed, when compared to the synthetic example for $\kappa_g{=}10^4$ as shown in Figure 2, we also observe that only AmIGO-CG, AmIGO and AID-CG could successfully optimize the loss. Hence, these results confirm the importance of warm-start for an improved performance.

## 6 CONCLUSION

We studied AmIGO, an algorithm for bilevel optimization based on amortized implicit differentiation and introduced a unified framework for analyzing its convergence. Our analysis showed that AmIGO achieves the same complexity as *unbiased oracle methods*, thus achieving improved rates compared to methods without warm-start in various settings. We then illustrated empirically such improved convergence in both synthetic and a hyper-optimization experiments. A future research direction consists in extending the proposed framework to non-smooth objectives and analyzing acceleration in both inner and outer level problems as well as variance reduction techniques.

## 7 ACKNOWLEDGMENTS AND FUNDING

This project was supported by the ERC grant number 714381 (SOLARIS project) and by ANR 3IA MIAI@Grenoble Alpes, (ANR19-P3IA-0003).

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

# A CONVERGENCE OF AMIGO ALGORITHM

In this section, we provide a proof of Theorem 1 as well as its corollaries Corrolaries 1 to 4. In Appendix A.1, we provide an outline of the proof Theorem 1 that states the main intermediary results needed for the proof and provide explicit expressions for the quantities needed throughout the rest of the paper. Appendices A.2 and A.3 provide the proofs of Theorem 1 and Corrolaries 1 to 4. The proofs of the intermediary results are deferred to Appendices B and C.

## A.1 PROOF OUTLINE OF THEOREM 1

The proof of Theorem 1 proceeds in 8 steps as discussed bellow.

**Step 1: Smoothness properties.** This step consists in characterizing the smoothness of $\nabla \mathcal{L}$, $y^\star$, $z^\star$ as well as the conditional expectation $\mathbb{E}[\hat{\psi}_k | x_k, y_k, z_k]$ knowing $x_k$, $y_k$ and $z_k$. For this purpose, we consider the function $\Psi : \mathcal{X} \times \mathcal{Y} \times \mathcal{Y} \to \mathcal{X}$ defined as follows:

$$\Psi(x, y, z) := \partial_x f(x, y) + \partial_{xy} g(x, y) z.$$

Hence, by definition of $\hat{\psi}_k$, it is easy to see that $\mathbb{E}[\hat{\psi}_k | x_k, y_k, z_k] = \Psi(x_k, y_k, z_k)$. The following proposition controls the smoothness of $\nabla \mathcal{L}$, $y^\star$, $z^\star$ and $\Psi$ and is adapted from (Ghadimi and Wang, 2018, Lemma 2.2). We provide a proof in Appendix B.2 for completeness.

**Proposition 6.** *Under Assumptions 1 to 3, $\mathcal{L}$, $\Psi$, $y^\star$ and $z^\star$ satisfy:*

$$\|y^\star(x) - y^\star(x')\| \leq L_y \|x - x'\|, \qquad \|z^\star(x, y) - z^\star(x', y')\| \leq L_z(\|x - x'\| + \|y - y'\|)$$
$$\|\nabla \mathcal{L}(x) - \nabla \mathcal{L}(x')\| \leq L \|x - x'\|, \quad \|\Psi(x, y, z) - \nabla \mathcal{L}(x)\| \leq L_\psi[\|y - y^\star(x)\| + \|z - z^\star(x, y)\|]$$
$$\|z^\star(x, y^\star(x))\| \leq \mu_g^{-1} B,$$

*where $L_y$, $L_z$, $L_\psi$ and $L$ are given by:*

$$L_y := \mu_g^{-1} L'_g, \qquad L_z := \mu_g^{-2} M_g B + \mu_g^{-1} L_f, \tag{13}$$
$$L_\psi := \max \left( (L_f + M_g \mu_g^{-1} B + L'_g L_z), L'_g \right)$$
$$L := \left( L_f + \mu_g^{-2} L'_g M_g B + \mu_g^{-1} (L'_g L_f + M_g B) \right) \left( 1 + \mu_g^{-1} L'_g \right)$$

The expressions of $L_y$, $L_z$, $L_\psi$ and $L$ suggests the following dependence on the conditioning $\kappa_g$ of the inner-level problem which will be useful for the complexity analysis: $L_y = O(\kappa_g)$, $L_\psi = O(\kappa_g^2)$, $L_z = O(\kappa_g^2)$ and $L = O(\kappa_g^3)$.

**Step 2: Convergence of the inner-level iterates.** In this step, we control the mean squared errors $\mathbb{E}\left[\|y_k - y^\star(x_k)\|^2\right]$ and $\mathbb{E}\left[\|z_k - z^\star(x_k, y_k)\|^2\right]$ as stated in Proposition 3. In fact we prove a slightly stronger version stated below:

**Proposition 7.** *Let $\alpha_k$ and $\beta_k$ be two positive sequences with $\alpha_k \leq L_g^{-1}$ and $\beta_k \leq \frac{1}{2L_g} \min \left( 1, \frac{2L_g}{\mu_g(1 + \mu_g^{-2} \sigma_{g_{yy}}^2)} \right)$ and define $\Lambda_k := (1 - \alpha_k \mu_g)^T$ and $\Pi_k := \left( 1 - \frac{\beta_k \mu_g}{2} \right)^N$. Denote by $\bar{z}_k$ the conditional expectation of $z_k$ knowing $x_k$, $y_k$ and $z_k^0$. Let $R_k^y$ and $R_k^z$ be defined as:*

$$R_k^y := 2\alpha_k \mu_g^{-1} \sigma_g^2, \quad R_k^z := \beta_k B^2 \mu_g^{-3} \sigma_{g_{yy}}^2 + 3\mu_g^{-2} \sigma_f^2. \tag{14}$$

*Then, under Assumptions 1, 4 and 5 the iterates $z_k$ and $\bar{z}_k$ satisfy:*

$$\mathbb{E}\left[\|y_k - y^\star(x_k)\|^2\right] \leq \Lambda_k E_k^y + R_k^y$$
$$\mathbb{E}\left[\|z_k - z^\star(x_k, y_k)\|^2\right] \leq \Pi_k \mathbb{E}\left[\|z_k^0 - z^\star(x_k, y_k)\|^2\right] + R_k^z,$$
$$\mathbb{E}\left[\|\bar{z}_k - z^\star(x_k, y_k)\|^2\right] \leq \Pi_k \mathbb{E}\left[\|z_k^0 - z^\star(x_k, y_k)\|^2\right]$$
$$\mathbb{E}\left[\|z_k - \bar{z}_k\|^2\right] \leq 4\sigma_g^2 \mu_g^{-2} \Pi_k \mathbb{E}\left[\|z_k^0 - z^\star(x_k, y_k)\|^2\right] + 2R_k^z. \tag{15a}$$

It is easy to see from the above expressions that $R_k^y = O\big(\kappa_g \sigma_g^2\big)$ while $R_k^z = O\big(\kappa_g^3 \sigma_{g_y y}^2 + \kappa_g^2 \sigma_f^2\big)$ as stated in Proposition 3. Controlling $\mathbb{E}[\|y_k - y^\star(x_k)\|]^2$ follows by standard results on SGD (Kulunchakov and Mairal, 2020, Corollary 31) since the iterates of Algorithm 2 uses i.i.d. samples. The error terms $\mathbb{E}\big[\|z_k - z^\star(x_k, y_k)\|^2\big]$ is more delicate since Algorithm 3 uses the same sample $\partial_y \hat{f}(x_k, y_k)$ for updating the iterates, therefore introducing additional correlations between these iterates. We defer the proof of Proposition 7 to Appendix B.3 which relies on a general result for stochastic linear systems with correlated noise provided in Appendix E.

**Step 3: Controlling the bias and variance errors $V_k^\psi$ and $E_k^\psi$** is achieved by Proposition 4. The bias $E_k^\psi$ is controlled simply by using the smoothness of the potential $\Psi$ near the point $(x_k, y^\star(x_k), z^\star(x_k, y^\star(x_k))$ as shown in Proposition 6. The variance term $V_k^\psi$ is more delicate to control due to the multiplicative noise resulting from the Jacobian-vector product between $\partial_{xy} \hat{g}(x_k, y_k, \mathcal{D}_{g_{xy}}) z_k$. We defer the proof to Appendix B.4 and provide below explicit expressions for the constants $\sigma_x^2$ and $w_x^2$:

$$w_x^2 := \Big(1 + 2L'_g \mu_g^{-1} + 6\big(\sigma_{g_{xy}}^2 + (L'_g)^2\big)\mu_g^{-2}\Big)\sigma_f^2 \tag{16a}$$
$$+ 2\big(\sigma_{g_{xy}}^2 + (L'_g)^2\big)B^2 L_g^{-1} \mu_g^{-3} \sigma_{g_{yy}}^2 + 2B^2 \mu_g^{-2} \sigma_{g_{xy}}^2.$$
$$\sigma_x^2 := 2\sigma_{g_{xy}}^2 + 2(L'_g)^2 \mu_g^{-2} \sigma_{g_{yy}}^2.$$

Note that $w_x^2 = O\Big(\kappa_g^2\big(\sigma_f^2 + \sigma_{g_{xy}}^2\big) + \kappa_g^3 \sigma_{g_{yy}}^2\Big)$ and $\sigma_x^2 = O\Big(\sigma_{g_{xy}}^2 + \kappa_g^2 \sigma_{g_{yy}}^2\Big)$, as stated in Proposition 4.

**Step 4: Outer-level error bound.** This step consists in obtaining the inequality in Proposition 2 which extends the result of (Kulunchakov and Mairal, 2020, Proposition 1) to biased gradients and to the non-convex case. We defer the proof of such result to Appendix C.1.

**Step 5: Inner-level error bound.** This step consists in proving Proposition 5. For clarity, we provide a second statement with explicit expressions for the quantities of interest:

**Proposition 8.** *Let $r_k$ and $\theta_k$ be two positive non-increasing sequences no greater than 1. For any $0 \leq v \leq 1$, denote by $\phi_k$ and $\tilde{R}_k$ the following non-negative scalars:*

$$\phi_k := (1-v)L_g^2 T^2 \alpha_k^2 + 2v, \qquad \tilde{R}_k^y := 2(1-v)L_g \mu_g^{-1} T^2 \sigma_g^2 \alpha_k^2 + vR_k^y$$
$$\zeta_k := 2L\big(\min\big((1-u)^{-1}, L\eta_{k-1}^{-1}\big)\mathbb{1}_{\mu \geq 0} + \mathbb{1}_{\mu < 0}\big)$$

*Finally, consider the following matrices and vectors:*

$$\boldsymbol{P_k} := \begin{pmatrix} 1+r_k, & 0 \\ 16L_z^2 \frac{\phi_k}{\theta_k}, & 1+\theta_k \end{pmatrix}, \quad \boldsymbol{U_k} := \begin{pmatrix} 2L_y^2 \frac{\gamma_k}{r_k} \\ 4L_z^2 \frac{\gamma_k}{\theta_k}\big(1 + 4L_y^2 \frac{\phi_k}{r_k}\big) \end{pmatrix}, \quad \boldsymbol{V_k} := \tilde{R}_k^y \begin{pmatrix} 0 \\ 8L_z^2 \theta_k^{-1} \end{pmatrix}. \tag{17}$$

*Then, under Assumptions 1 to 5, the following holds:*

$$\begin{pmatrix} E_k^y \\ E_k^z \end{pmatrix} \leq \boldsymbol{P_k} \begin{pmatrix} \Lambda_{k-1} E_{k-1}^y + R_{k-1}^y \\ \Pi_{k-1} E_{k-1}^z + R_{k-1}^z \end{pmatrix} + 2\gamma_k \big(E_{k-1}^\psi + V_{k-1}^\psi + \zeta_k E_{k-1}^x\big)\boldsymbol{U_k} + \boldsymbol{V_k}.$$

We defer the proof of the above result to Appendix C.2.

**Step 6: General error bound.** By combining the inequalities in Propositions 2 and 8 resulting from the analysis of both outer and inner levels, we obtain a general error bound on $E_k^{tot}$ in Proposition 9 with a proof in Appendix C.4.

**Proposition 9.** *Choose the step-sizes $\alpha_k$ and $\beta_k$ such that they are non-increasing in $k$ and choose $r_k$ and $\theta_k$ such that $\delta_k r_k^{-1}$ and $\delta_k \theta_k^{-1}$ are non-increasing sequences. Choose the coefficients $a_k$ and $b_k$ defining $E_k^{tot}$ in (7) to be of the form $a_k = \delta_k r_k^{-1} \Lambda_k^s$ and $b_k = \delta_k \theta_k^{-1} \Pi_k^s$ for some $0 < s < 1$. and fix a non-increasing sequence $0 < \rho_k < 1$. Then, under Assumptions 1 to 5 $E_k^{tot}$ satisfies:*

$$F_k + E_k^{tot} \leq \|A_k\|_\infty E_{k-1}^{tot} + V_k^{tot}.$$

*where $V_k$ and $\boldsymbol{A_k}$ are given by:*

$$A_k := \begin{pmatrix} 1 - (1 - \frac{1}{2}\rho_k)\delta_k + 2\zeta_k\lambda_k u_k^I \\ \Lambda_k^{1-s}\left(1 + r_k\left(1 + 16L_z^2\Pi_k^s\theta_k^{-2}\phi_k + 2L_\psi^2\eta_k^{-1}\left(2u_k^I + s_k + \rho_k^{-1}\right)\right)\right) \\ \Pi_k^{1-s}\left(1 + \theta_k\left(1 + \eta_k^{-1}\left(2L_\psi^2 + \sigma_x^2\right)\left(2u_k^I + s_k + \rho_k^{-1}\right)\right)\right) \end{pmatrix} \quad (18)$$

$$V_k^{tot} := \delta_k\left(\Lambda_k^s(1 + r_k^{-1}) + 16L_z^2\phi_k\theta_k^{-2}\Pi_k^s + 2L_\psi^2\eta_k^{-1}\left(2u_k^I + s_k + \rho_k^{-1}\right)\right)R_{k-1}^y$$
$$+ \delta_k\left((1 + \theta_k^{-1})\Pi_k^s R_{k-1}^z + 8L_z^2\theta_k^{-2}\Pi_k^s \tilde{R}_k^y\right) + \gamma_k\left(s_k + u_k^I\right)w_x^2$$

*where we introduced $u_k^I = a_k U_k^{(1)} + b_k U_k^{(2)}$ for conciseness with $U_k^{(1)}$ and $U_k^{(2)}$ being the components of the vector $\boldsymbol{U_k}$ defined in Proposition 8.*

Proposition 9 holds without conditions on the error made by Algorithms 2 and 3. The general form of $a_k$ and $b_k$ allows to account for potentially decreasing step-sizes $\gamma_k$, $\alpha_k$ and $\beta_k$. However, in the present work, we will restrict to the constant step-size for ease of presentation as we discuss next.

**Step 7: Controlling the precision of the inner-level algorithms.** In this step, we provide conditions on $T$ and $N$ in Proposition 10 bellow so that $\|A_k\|_\infty \leq 1 - (1 - \rho_k)\delta_k$ in the constant step-size case. These conditions are expressed in terms of the following constants:

$$C_1 := 1 + 2\log\left(6 + 24L_\psi^2\eta_0^{-1}\right) \tag{19a}$$

$$C_2 := 2\log\left(1 + 4L_y^2 L^{-2}\max\left(\eta_0, 8\zeta_0\right)\right) \tag{19b}$$

$$C_3 := \max\left(0, -2\log\left(5L_\psi^2\eta_0^{-1}\right), -2\log\left(\frac{L}{4L_y^2}\right)\right) \tag{19c}$$

$$C_1' := 1 + 2\log\left(4 + 12\eta_0^{-1}(2L_\psi^2 + \sigma_x^2)\right) \tag{19d}$$

$$C_2' := 2\log\left(1 + 2L_z^2 L_y^{-2}\left(1 + 16L_y^2\right)\right), \tag{19e}$$

$$C_3' := \max\left(0, -2\log\left(\frac{L_\psi^2}{4L_z^2\eta_0}\right), -2\log\left(\gamma\left(\sigma_{g_{xy}}^2 + (L_g')^2\right)\right)\right) \tag{19f}$$

**Proposition 10.** *Let Assumptions 1 to 5 hold. Choose $\rho_k = \frac{1}{2}$, the step-sizes to be constant: $\alpha_k = \alpha \leq \frac{1}{L_g}$, $\beta_k = \beta \leq \frac{1}{2L_g}$ and $\gamma_k = \gamma \leq \frac{1}{L}$ and choose the batch-size $|\mathcal{D}_{g_{yy}}| = \Theta\left(\frac{\tilde{\sigma}_{g_{yy}}^2}{\mu_g L_g}\right)$. Moreover, set $s = \frac{1}{2}$, $r_k = \theta_k = 1$. Finally, choose $T$ and $N$ as follows:*

$$T = \lfloor \alpha^{-1}\mu_g^{-1}\max\left(C_1, C_2, C_3\right)\rfloor + 1,$$
$$N = \lfloor 2\beta^{-1}\mu_g^{-1}(\max\left(C_1, C_2, C_3\right) + \max\left(C_1', C_2', C_3'\right))\rfloor + 1$$

*with $C_1, C_2, C_3, C_1', C_2'$ and $C_3'$ defined in (19a) to (19f). Then, $\|A_k\|_\infty \leq 1 - (1 - \rho_k)\delta_k$ and $V_k^{tot} \leq \gamma\delta_0 \mathcal{W}^2$, with $\mathcal{W}^2$ given by:*

$$\mathcal{W}^2 := \left(\delta_0^{-1}\frac{1-u}{2}\mathbb{1}_{\mu>0} + 3\right)w_x^2 + \frac{60L_\psi^2}{\delta_0\mu_g L_g}\sigma_g^2 \tag{20}$$

We provide a proof of Proposition 10 in Appendix D. It is easy to see from Proposition 10 that that $T = O(\kappa_g)$ and $N = O(\kappa_g)$ when $\alpha = \frac{1}{L_g}$ and $\beta = \frac{1}{2L_g}$, where the big-$O$ notation hides a logarithmic dependence in $\kappa_g$ coming from the constants $\{C_i, C_i' | i \in \{1, 2, 3\}\}$.

**Step 8: Proving the main inequalities.** The final step combines Propositions 9 and 10 to get the desired inequality. We provide a full proof in Appendix A.2 assuming Propositions 9 and 10 hold.

## A.2 Proof of Theorem 1

In order to prove Theorem 1 in the convex case, we need the following *averaging strategy lemma*, a generalization of (Kulunchakov and Mairal, 2020, Lemma 30):

**Lemma 1.** *Let $\mathcal{L}$ be a convex function on $\mathcal{X}$. Let $x_k$ be a (potentially stochastic) sequence of iterates in $\mathcal{X}$. Let $(E_k)_{k\geq 0}$, $(V_k)_{k\geq 0}$ and $(\delta_k)_{k\geq 0}$ be non-negative sequences such that $\delta_k \in (0,1)$. Fix some non-negative number $u \in [0,1]$ and define the following averaged iterates $\hat{x}_k$ recursively by $\hat{x}_k = u(1-\delta_k)\hat{x}_{k-1} + (1-(1-\delta_k)u)x_k$ and starting from any initial point $\hat{x}_0$. Assume the iterates $(x_k)_{k\geq 1}$ satisfy the following relation for all $k \geq 1$:*

$$(1 - u(1-\delta_k))\mathbb{E}[\mathcal{L}(x_k) - \mathcal{L}^\star] + E_k \leq (1-\delta_k)(E_{k-1} + (1-u)\mathbb{E}[\mathcal{L}(x_{k-1}) - \mathcal{L}^\star]) + V_k. \quad (21)$$

*Let $\Gamma_k := \prod_{t=1}^{k}(1-\delta_k)$. Then the averaged iterates $(\hat{x}_k)_{k\geq 1}$ satisfy the following:*

$$\mathbb{E}[\mathcal{L}(\hat{x}_k) - \mathcal{L}^\star] + E_k \leq \Gamma_k\big(E_0 + \mathbb{E}\big[\mathcal{L}(x_0) - \mathcal{L}^\star + u^k(\mathcal{L}(\hat{x}_0) - \mathcal{L}^\star)\big]\big) + \Gamma_k \sum_{1 \leq t \leq k} \Gamma_t^{-1} V_t.$$

*Proof.* For simplicity, we write $F_k = \mathbb{E}[\mathcal{L}(x_k) - \mathcal{L}^\star]$ and $\hat{F}_k = \mathbb{E}[\mathcal{L}(\hat{x}_k) - \mathcal{L}^\star]$. We first multiply (21) by $\Gamma_k^{-1}$ and sum the resulting inequalities for all $1 \leq k \leq K$ to get:

$$\sum_{k=1}^{K} \Gamma_k^{-1}(1 - u(1-\delta_k))F_k + \Gamma_k^{-1}T_k \leq \sum_{k=1}^{K}\Gamma_k^{-1}(1-\delta_k)(T_{k-1} + (1-u)F_{k-1}) + \sum_{k=1}^{K}\Gamma_k^{-1}V_k.$$

Grouping the terms in $F_k$ together and recalling that $\Gamma_k^{-1}(1-\delta_k) = \Gamma_{k-1}^{-1}$ yields:

$$\sum_{k=1}^{K}\Gamma_k^{-1}F_k - \Gamma_{k-1}^{-1}F_{k-1} + u\sum_{k=1}^{K}\Gamma_{k-1}^{-1}(F_{k-1} - F_k) \leq \sum_{k=1}^{K}\left(\Gamma_{k-1}^{-1}T_{k-1} - \Gamma_k^{-1}T_k\right) + \sum_{k=1}^{K}\Gamma_k^{-1}V_k.$$

Simplifying the telescoping sums and multiplying by $\Gamma_K$, we get:

$$F_K + u\Gamma_K \sum_{k=1}^{K}\Gamma_{k-1}^{-1}(F_{k-1} - F_k) \leq -T_K + \Gamma_K\left(F_0 + T_0 + \sum_{k=1}^{K}\Gamma_k^{-1}V_k\right). \quad (22)$$

Consider now the quantity $\hat{F}_k$. Recalling that $\mathcal{L}$ is convex and by definition of the iterates $\hat{x}_k$ we apply Jensen's inequality to write:

$$\hat{F}_k \leq u(1-\delta_k)\hat{F}_{k-1} + (1 - u(1-\delta_k))F_k.$$

By iteratively applying the above inequality, we get that:

$$\hat{F}_K \leq u^K \Gamma_K \hat{F}_0 + \Gamma_K \sum_{k=1}^{K} u^{K-k}\Gamma_k^{-1}(1 - u(1-\delta_k))F_k$$

$$= u^K \Gamma_K \hat{F}_0 + \Gamma_K \sum_{k=1}^{K} u^{K-k}\Gamma_k^{-1}F_k - \sum_{k=1}^{K} u^{K-k+1}\Gamma_{k-1}F_k$$

$$= u^K \Gamma_K \hat{F}_0 + F_K + \Gamma_K \sum_{k=1}^{K-1} u^{K-k}\Gamma_k^{-1}(F_k - F_{k+1})$$

We can therefore apply (22) to the above inequality to get the desired result. $\square$

We now proceed to prove Theorem 1.

*Proof of Theorem 1.* By application of Proposition 9 and using the choice of $T$ and $N$ given by Proposition 10, the following inequality holds:

$$F_k + E_k^{tot} \leq (1 - (1-\rho_k)\delta_k)E_{k-1}^{tot} + \gamma\delta_0\mathcal{W}^2. \quad (23)$$

with $\mathcal{W}$ defined in Proposition 10. We then distinguish two cases depending on the sign of $\mu$:

**Case $\mu \geq 0$.** Recall that $F_k$ and $E_k^x$ are given by:

$$F_k = u\delta_k\mathbb{E}[\mathcal{L}(x_k) - \mathcal{L}^\star], \qquad E_k^x = \frac{\delta_k}{2\gamma_k}\|x_k - x^\star\|^2 + (1-u)\mathbb{E}[\mathcal{L}(x_k) - \mathcal{L}^\star]$$

Since $\mu > 0$, $\mathcal{L}$ is a convex function and we can apply Lemma 1 with $V_k = \gamma\delta_0\mathcal{W}^2$ and $E_k = \frac{\delta_k}{2\gamma_k}\|x_k - x^\star\|^2 + a_k E_k^y + b_k E_k^z$. The result follows by noting that $\Gamma_k \sum_{t=1}^{k} \Gamma_t^{-1} \le \delta_0^{-1}$.

**Case $\mu < 0$.** In this case, we recall that $F_k$ and $E_k^x$ are given by:

$$F_k = \mathbb{E}[\mathcal{L}(x_k) - \mathcal{L}(x_{k-1})] + E_{k-1}^x - E_k^x, \qquad E_k^x = L^{-1}\mathbb{E}\Big[\|\nabla\mathcal{L}(x_k)\|^2\Big].$$

We then sum (23) for all iterations $0 \le t \le k$ which, by telescoping, yields:

$$\mathbb{E}[\mathcal{L}(x_k) - \mathcal{L}(x_0)] + E_0^x - E_k^x + E_k^{tot} - E_0^{tot} + \sum_{1 \le t \le k}(1 - \rho_t)\delta_t E_t^{tot} \le k\gamma\delta_0\mathcal{W}^2.$$

Using that $\mathbb{E}[\mathcal{L}(x_k) - \mathcal{L}^\star] + E_k^{tot} - E_k^x$ is non-negative since $E_k^{tot} - E_k^x = a_k E_k^y + b_k E_k^z$, we get:

$$\sum_{1 \le t \le k}(1 - \rho_t)\delta_t E_t^x \le (\mathbb{E}[\mathcal{L}(x_0) - \mathcal{L}^\star] + a_0 E_0^y + b_0 E_0^z) + k\gamma\delta_0\mathcal{W}^2.$$

Finally, since $\rho_t = \frac{1}{2}$, $\delta_t = L\gamma$, the result follows after dividing both sides by $\frac{1}{2}kL\gamma$. □

## A.3 PROOF OF CORROLARIES 1 TO 4

*Proof of Corollary 1.* Choosing $u = 0$ implies that $E_k^x = \frac{\mu}{2}\|x_k - x^\star\|^2 + \mathcal{L}(x_k) - \mathcal{L}^\star \le E_k^{tot}$. We can then apply Theorem 1 for $\mu > 0$ which yields the following:

$$\frac{\mu}{2}\|x_k - x^\star\| + \mathcal{L}(x_k) - \mathcal{L}^\star \le \left(1 - (2\kappa_\mathcal{L})^{-1}\right)^k E_0^{tot} + \frac{2\mathcal{W}^2}{L}. \tag{24}$$

In the deterministic setting, it holds all variances vanish : $\sigma_f^2 = \sigma_g^2 = \sigma_{g_{yy}}^2 = \sigma_{g_{xy}}^2 = 0$. Hence, $\mathcal{W}^2 = 0$ by definition of $\mathcal{W}^2$. Therefore, to achieve an error $\mathcal{L}(x_k) - \mathcal{L}^\star \le \epsilon$ for some $\epsilon > 0$, (24) suggests choosing $k = O\left(\kappa_\mathcal{L} \log\left(\frac{E_0^{tot}}{\epsilon}\right)\right)$. Additionally, $T = \Theta(\kappa_g)$ and $N = \Theta(\kappa_g)$ as required by Theorem 1 and since $\sigma_{g_{yy}}^2 = 0$, it holds that $N = O(\kappa_g)$. Using batches of size 1, yields the desired complexity. □

*Proof of Corollary 2.* Here we choose $u = 1$ and apply Theorem 1 for $\mu > 0$ which yields:

$$\mathbb{E}[\mathcal{L}(\hat{x}_k) - \mathcal{L}^\star] + E_k^{tot} \le \left(1 - (2\kappa_\mathcal{L})^{-1}\right)^k \left(E_0^{tot} + \mathbb{E}[\mathcal{L}(x_0) - \mathcal{L}^\star]\right) + 2L^{-1}\mathcal{W}^2$$

Hence, to achieve an error $\mathbb{E}[\mathcal{L}(\hat{x}_k) - \mathcal{L}^\star] \le \epsilon$, we need $k = O\left(\kappa_\mathcal{L} \log\left(\frac{E_0^{tot} + \mathbb{E}[\mathcal{L}(x_0) - \mathcal{L}^\star]}{\epsilon}\right)\right)$ to guarantee that the first term in the l.h.s. of the above inequality is $O(\epsilon)$. Moreover, we recall that $L^{-1} = O(\kappa_g^{-3})$ from Proposition 6 and that Theorem 1 ensure the variance $\mathcal{W}$ satisfies:

$$\mathcal{W}^2 = O\left(\kappa_g^5\sigma_g^2 + \kappa_g^3\sigma_{g_{yy}}^2 + \kappa_g^2\sigma_{g_{xy}}^2 + \kappa_g^2\sigma_f^2\right).$$

Hence, ensuring the variance term $2L^{-1}\mathcal{W}^2$ is of order $\epsilon$ is achieved by choosing the size of the batches as follows:

$$|\mathcal{D}_f| = \Theta\left(\frac{\tilde{\sigma}_f^2}{\epsilon}\right), \quad |\mathcal{D}_g| = \Theta\left(\frac{\kappa_\mathcal{L}\kappa_g^2\tilde{\sigma}_g^2}{\epsilon}\right), \quad |\mathcal{D}_{g_{xy}}| = \Theta\left(\frac{\tilde{\sigma}_{g_{xy}}^2}{\epsilon}\right),$$

$$|\mathcal{D}_{g_{yy}}| = \Theta\left(\tilde{\sigma}_{g_{yy}}^2\left(\frac{1}{\epsilon} \vee \kappa_g\right)\right)$$

Recall that $T = \Theta(\kappa_g)$ and $N = \Theta(\kappa_g)$ as required by, Theorem 1, thus yielding the desired result. □

*Proof of Corollary 3.* In the non-convex deterministic case, recall that $E_k^x = \frac{1}{L}\|\nabla\mathcal{L}(x_k)\|^2 \le E_k^{tot}$. We thus apply Theorem 1 for $\mu < 0$, multiply by $L$ to get:

$$\frac{1}{k}\sum_{t=1}^{k}\|\nabla\mathcal{L}(x_t)\|^2 \le \frac{2L}{k}(\mathcal{L}(x_0) - \mathcal{L}^\star + (E_0^y + E_0^z)) + 2\mathcal{W}^2.$$

The setting being deterministic, it holds that $\mathcal{W}^2 = 0$. Moreover, recall that $L = O(\kappa_g^3)$ from Proposition 6. Hence, to achieve an error of order $\min_{1 \le t \le k}\|\nabla\mathcal{L}(x_t)\|^2 \le \epsilon$, it suffice to choose $k = O\left(\frac{\kappa_g^3}{\epsilon}(\mathcal{L}(x_0) - \mathcal{L}^\star + (E_0^y + E_0^z))\right)$. Thus using batches of size 1 and $T$ and $N$ of order $\kappa_g$. □

*Proof of Corollary 4.* In the non-convex stochastic case, $E_k^x = \frac{1}{L}\mathbb{E}\Big[\|\nabla\mathcal{L}(x_k)\|^2\Big] \leq E_k^{tot}$. We thus apply Theorem 1 for $\mu < 0$, multiply by $L$ to get:

$$\frac{1}{k}\sum_{t=1}^{k}\mathbb{E}\Big[\|\nabla\mathcal{L}(x_t)\|^2\Big] \leq \frac{2L}{k}(\mathbb{E}[\mathcal{L}(x_0) - \mathcal{L}^\star] + (E_0^y + E_0^z)) + 2\mathcal{W}^2.$$

to achieve an error of order $\epsilon$, we need to ensure each term in the l.h.s. of the above inequality is of order $\epsilon$. For the first term, similarly to the deterministic setting Corollary 3, we simply need $k = O\Big(\frac{\kappa_g^3}{\epsilon}(\mathcal{L}(x_0) - \mathcal{L}^\star + (E_0^y + E_0^z))\Big)$. For the second term, we need to have $\mathcal{W}^2 = O(\epsilon)$, which is achieved using the following choice for the sizes of the batches:

$$|\mathcal{D}_f| = O\left(\frac{\kappa_g^2\tilde{\sigma}_f^2}{\epsilon}\right), \quad |\mathcal{D}_g| = O\left(\frac{\kappa_g^5\tilde{\sigma}_g^2}{\epsilon}\right), \quad |\mathcal{D}_{g_{xy}}| = O\left(\frac{\kappa_g^2\tilde{\sigma}_{g_{xy}}^2}{\epsilon}\right),$$

$$|\mathcal{D}_{g_{yy}}| = O\left(\frac{\kappa_g^3\tilde{\sigma}_{g_{yy}}^2}{\epsilon}\left(1 \vee \epsilon\mu_g^2\right)\right).$$

Finally, as required by Theorem 1, we set $T = \Theta(\kappa_g)$ and $N = \Theta(\kappa_g)$ thus yielding the desired complexity. $\qquad\square$

## A.4 COMPARAISONS WITH OTHER METHODS

In this subsection we derive and discuss the complexities of methods presented in Table 1.

### A.4.1 COMPARAISON WITH TTSA (HONG ET AL., 2020B)

**Proposition 11. strongly-convex case $\mu > 0$.** *The complexity of the TTSA algorithm in Hong et al. (2020b) to achieve an error $\frac{\mu}{2}\mathbb{E}\Big[\|x_k - x^\star\|^2\Big] \leq \epsilon$ is given by:*

$$\mathcal{C}(\epsilon) := O\left(\left(\kappa_g^{\frac{17}{2}}\kappa_{\mathcal{L}}^{1/2} + \kappa_{\mathcal{L}}^{7/2}\right)\frac{1}{\epsilon^{\frac{3}{2}}}\log\frac{1}{\epsilon}\right)$$

**non-convex case $\mu < 0$.** *The complexity of the TTSA algorithm in Hong et al. (2020b) to achieve an error $\frac{1}{k}\sum_{1\leq i\leq k}\mathbb{E}\Big[\|\nabla\mathcal{L}(x_i)\|^2\Big] \leq \epsilon$ is given by:*

$$\mathcal{C}(\epsilon) := k(1 + N) = O\left(\left(\kappa_g^{11} + \kappa_g^{16}\right)\frac{1}{\epsilon^{\frac{5}{2}}}\log\left(\frac{1}{\epsilon}\right)\right).$$

*Proof.* **strongly-convex case $\mu > 0$** Using the choice of step-sizes in Hong et al. (2020b), the following bound holds:

$$\mathbb{E}\Big[\|x_k - x^\star\|^2\Big] \lesssim \prod_{i=0}^{k}\left(1 - \frac{8}{3(k + k_\alpha)}\right)\left(\Delta_x^0 + L_\psi^2\mu_g^{-2}\Delta_0^y\right)$$

$$+ \frac{L_\psi^2}{\mu_g\mu^{\frac{4}{3}}}\left(\mu_g^{-1} + \frac{\mu_g}{\mu^2}L_y^2\right)\left(\frac{1}{k + k_\alpha}\right)^{\frac{2}{3}}$$

where $\Delta_x^0 = \mathbb{E}\Big[\|x_0 - x^\star\|^2\Big], \mathbb{E}\Big[\|y_1 - y^\star(x_0)\|^2\Big]$ and $k_\alpha$ given by:

$$k_\alpha = \max\left(35\left(\frac{L_g^3}{\mu_g^3}(1 + \sigma_g^2)^{\frac{3}{2}}\right), \frac{(512)^{\frac{3}{2}}L_\psi^2L_y^2}{\mu^2}\right).$$

By a simple calculation, it is easy to see that $\prod_{i=0}^{k}\left(1 - \frac{8}{3(k + k_\alpha)}\right) \leq \frac{(k_\alpha - 1)^2}{(k - 1 + k_\alpha)^2}$. Moreover, using that $L_\psi = O(\mu_g^{-2})$, $L_y = O(\mu_g^{-1})$, we get that

$$\frac{\mu}{2}\mathbb{E}\Big[\|x_k - x^\star\|^2\Big] \lesssim \frac{(k_\alpha - 1)^2}{(k + k_\alpha - 1)^2}\left(\mu\Delta_x^0 + \mu\mu_g^{-6}\Delta_0^y\right) + \mu_g^{-6}\mu^{-\frac{1}{3}}\left(1 + \mu^{-2}\right)\left(\frac{1}{k + k_\alpha}\right)^{\frac{2}{3}}$$

Using that $L = O(\mu_g^{-3})$, we get $\mu_g^{-6}\mu^{-\frac{1}{3}}(1 + \mu^{-2}) = O\left(\mu_g^{-5}\kappa_{\mathcal{L}}^{\frac{1}{3}} + \mu_g\kappa_{\mathcal{L}}^{\frac{7}{3}}\right)$. Hence, to reach an error $\epsilon$, we need to control both terms in the above inequality. This suggests the following condition on $k$ to control the second term which dominates the error:

$$k \geq \left(\mu_g^{\frac{3}{2}}\kappa_{\mathcal{L}}^{7/2} + \kappa_{\mathcal{L}}^{1/2}\mu_g^{-\frac{15}{2}}\right)\frac{1}{\epsilon^{\frac{3}{2}}}.$$

Moreover, the result in (Hong et al., 2020b, Theorem 1) requires $N = \Theta\left(\kappa_g \log\frac{1}{\epsilon}\right)$, where $N$ is the number of terms in the Neumann series used to approximate the hessian inverse $(\partial_{yy}g(x, y))^{-1}$ in the expression of the gradient $\nabla\mathcal{L}$. Hence, the total complexity is given by the following expression:

$$\mathcal{C}(\epsilon) := k(1 + N) = O\left(\left(\kappa_g^{\frac{17}{2}}\kappa_{\mathcal{L}}^{1/2} + \kappa_{\mathcal{L}}^{7/2}\right)\frac{1}{\epsilon^{\frac{3}{2}}}\log\frac{1}{\epsilon}\right)$$

**Smooth Non-convex case** $\mu < 0$. Following Hong et al. (2020b), consider the proximal map of $\mathcal{L}$ for a fixed $\rho > 0$:

$$\hat{x}(z) := \arg\min_{x \in \mathcal{X}}\left\{\mathcal{L}(x) + \frac{\rho}{2}\|x - z\|^2\right\}$$

and define the quantity $\tilde{\Delta}_x^k := \mathbb{E}\left[\|\hat{x}(x_k) - x_k\|^2\right]$, where $x_k$ are the iterates produced by the TTSA algorithm. Let $K$ be a random variable uniformly distributed on $\{0, ..., K - 1\}$ and independent from the remaining r.v. used in the TTSA algorithm. The result in (Hong et al., 2020b, Theorem 2) provide the following error bound on $\tilde{\Delta}_x^k$

$$\frac{1}{k}\sum_{1 \leq i \leq k}\tilde{\Delta}_x^i \lesssim \left(L_\psi^2\left(\Delta^0 + \frac{\sigma_g^2}{\mu_g^2}\right) + \mu_g\right)\frac{k^{-\frac{2}{5}}}{L^2}.$$

where $\rho$ is set to $2L$ and $\Delta^0 \leq \max\left(\mathbb{E}[\mathcal{L}(x_0) - \mathcal{L}^\star], \mathbb{E}\left[\|y_1 - y^\star(x_0)\|^2\right]\right)$. Now, recall that by definition of the proximal map, we have the following identity:

$$\frac{1}{k}\sum_{1 \leq i \leq k}\mathbb{E}\left[\|\nabla\mathcal{L}(x_i)\|^2\right] \leq 2(\rho^2 + L^2)\frac{1}{k}\sum_{1 \leq i \leq k}\tilde{\Delta}_x^i \lesssim L^2\frac{1}{k}\sum_{1 \leq i \leq k}\tilde{\Delta}_x^i.$$

Hence, we obtain the following error bound:

$$\frac{1}{k}\sum_{1 \leq i \leq k}\mathbb{E}\left[\|\nabla\mathcal{L}(x_i)\|^2\right] \leq \left(\kappa_g^4(\Delta^0 + \kappa_g^2\sigma_g^2) + \mu_g\right)k^{-\frac{2}{5}}$$

where we used that $L_\psi = O(\kappa_g^2)$. Therefore, to reach an error of order $\epsilon$, TTSA requires:

$$k \geq \frac{1}{\epsilon^{\frac{5}{2}}}\left(\kappa_g^{10}\Delta_0^{\frac{5}{2}} + \kappa_g^{15}\sigma_g^5\right)$$

Moreover, controlling the bias in the estimation of the gradient requires $N = O(\kappa_g \log\frac{1}{\epsilon})$ terms in the Neumann series approximating the hessian. Hence, the total complexity of the algorithm is:

$$\mathcal{C}(\epsilon) := k(1 + N) = O\left(\left(\kappa_g^{11} + \kappa_g^{16}\right)\frac{1}{\epsilon^{\frac{5}{2}}}\log\left(\frac{1}{\epsilon}\right)\right).$$

$\square$

### A.4.2 COMPARAISON WITH ACCBIO (JI AND LIANG, 2021)

**Complexity of AccBio.** The bilevel algorithm AccBio introduced in Ji and Liang (2021) uses acceleration for both the inner and outer loops. This allows to obtain the following conditions on $k, T$ and $N$ to achieve an $\epsilon$ accuracy:

$$k = O\left(\kappa_{\mathcal{L}}^{\frac{1}{2}}\log\frac{1}{\epsilon}\right), T = O\left(\kappa_g^{\frac{1}{2}}\right), N = O\left(\kappa_g^{\frac{1}{2}}\log\frac{1}{\epsilon}\right).$$

Note that, since AccBio do not use a warm-start strategy when solving the linear system, $N$ is required to grow as $\log \frac{1}{\epsilon}$ in order to achieve an $\epsilon$ accuracy. This contributes an additional logarithmic factor to the total complexity so that $\mathcal{C}(\epsilon) = O(\kappa_{\mathcal{L}}^{\frac{1}{2}} \kappa_g^{\frac{1}{2}} \left( \log \frac{1}{\epsilon} \right)^2)$. This is by contrast with AmIGO which exploits warm start when solving the linear system and thus only needs a constant number of iterations $N = O(\kappa_g)$ although the dependence on $\kappa_g$ is worse compared to AccBio. However, it is possible to improve such dependence by using acceleration in the inner-level algorithms $\mathcal{A}_k$ and $\mathcal{B}_k$ as we discuss in Appendix A.5.1.

**Complexity of AccBio as a function of $\mu$ and $\mu_g$.** The authors choose to report the complexity as a function of $\mu$ and $\mu_g$ instead of the conditioning numbers $\kappa_{\mathcal{L}}$ and $\kappa_g$. To achieve this, they observe that, under the additional assumption that the hessian $\partial_{yy} g(x, y)$ is constant w.r.t. $y$, the Lipschitz constant $L$ has an improved dependence on $\mu_g$: $L = O(\mu_g^{-2})$ instead of $L = O(\mu_g^{-3})$ in the general case where $\partial_{yy} g(x, y)$ is only Lipschitz in $y$. This allows them to express $\kappa_{\mathcal{L}} = \frac{L}{\mu} = O(\mu_g^{-2} \mu^{-1})$ and to report the following complexities in terms of $\mu$ and $\mu_g$:

$$\mathcal{C}(\epsilon) = O\left( \mu^{-\frac{1}{2}} \mu_g^{-\frac{3}{2}} \left( \log \frac{1}{\epsilon} \right)^2 \right).$$

Note that, in the general case where $L = O(\mu_g^{-3})$, the complexity as a function of $\mu$ and $\mu_g$ becomes $O\left( \mu^{-\frac{1}{2}} \mu_g^{-2} \left( \log \frac{1}{\epsilon} \right)^2 \right)$, while still maintaining the same expression in terms of $\kappa_{\mathcal{L}}$ and $\kappa_g$. Hence, using the expression in terms of conditioning allows a more general expression for the complexity that is less sensitive to the specific assumptions on $g$ and is therefore more suitable for comparaison with other results in the literature.

## A.5 CHOICE OF THE INNER-LEVEL ALGORITHMS $\mathcal{A}_k$ AND $\mathcal{B}_k$.

The choice of $\mathcal{A}_k$ and $\mathcal{B}_k$ has an impact on the total complexity of the algorithm. We discuss two choices for $\mathcal{A}_k$ and $\mathcal{B}_k$ which improve the total complexity of AmIGO: Accelerated algorithms (in Appendix A.5.1) and variance reduced algorithms (in Appendix A.5.2).

### A.5.1 ACCELERATION OF THE INNER-LEVEL FOR AMIGO

AmIGO could benefit from acceleration in the inner-loop by using standard acceleration schemes Nesterov (2003) for $\mathcal{A}_k$ and $\mathcal{B}_k$. As a consequence, and using analysis of accelerated algorithms (Nesterov, 2003) in the deterministic setting, the error of the inner-level iterates would satisfy:

$$\mathbb{E}\left[ \|y_k - y^\star(x_k)\|^2 \right] \leq \tilde{\Lambda}_k E_k^y, \qquad \mathbb{E}\left[ \|y_k - y^\star(x_k)\|^2 \right] \leq \tilde{\Pi}_k E_k^z$$

where $\tilde{\Lambda}_k$ and $\tilde{\Pi}_k$ are accelerated rates of the form $\tilde{\Lambda}_k = O((1-\sqrt{\kappa_g})^T)$ and $\tilde{\Pi}_k = O((1-\sqrt{\kappa_g})^N)$. The rest of the proofs are similar provided that $\Lambda_k$ and $\Pi_k$ are replaced by their accelerated rates $\tilde{\Lambda}_k$ and $\tilde{\Pi}_k$. This implies that $T$ and $N$ need to be only of order $T = O(\sqrt{\kappa_g})$ and $N = O(\sqrt{\kappa_g})$ so that the final complexity becomes:

$$\mathcal{C}(\epsilon) := O\left( \kappa_{\mathcal{L}} \kappa_g^{1/2} \log \frac{1}{\epsilon} \right).$$

Note that using conjugate gradient for $\mathcal{B}_k$ also enjoys an accelerated convergence rate Shewchuk et al. (1994). This is confirmed in our experiments of Figure 1 where AmIGO-CG enjoys the fastest convergence.

In order to further improve the dependence on $\kappa_{\mathcal{L}}$ to $\kappa_{\mathcal{L}}^{1/2}$, one would need to use an accelerated scheme when updating the iterates $x_k$. The analysis of such scheme along with warm-start would be an interesting direction for future work.

### A.5.2 VARIANCE REDUCED ALGORITHMS FOR $\mathcal{A}_k$ AND $\mathcal{B}_k$

When the inner-level cost function $g$ is a finite average of functions $g(x, y) = \frac{1}{n} \sum_{1 \leq i \leq n} g_i(x, y)$ empirical average, it is possible to use variance reduced algorithms such as SAG (Schmidt et al.,

2017). If every function $g_i$ is $L_g$-smooth, then by (Schmidt et al., 2017, Proposition 1), the inner level error becomes:

$$\mathbb{E}\Big[\|y_k - y^\star(x_k)\|^2\Big] \lesssim \tilde{\Lambda}_k(3E_k^y + \frac{9}{4}L_g^{-2}\sigma_g^2),$$

with $\tilde{\Lambda}_k = \left(1 - \frac{\kappa_g}{8n}\right)^T$. This has the advantage that the error due to the variance decays exponentially with the number of iterations $T$. As a consequence, the dependence of the effective variance $\mathcal{W}^2$ on the conditioning numbers $\kappa_\mathcal{L}$ and $\kappa_g$ can be improved to:

$$\mathcal{W}^2 = O\Big(|\mathcal{D}_g|^{-1}\tilde{\sigma}_g^2 + \kappa_g^3|\mathcal{D}_{g_{yy}}|^{-1}\tilde{\sigma}_{g_{yy}}^2 + \kappa_g^2|\mathcal{D}_{g_{xy}}|^{-1}\tilde{\sigma}_{g_{xy}}^2 + \kappa_g^2|\mathcal{D}_f|^{-1}\tilde{\sigma}_f^2\Big).$$

This can be achieved by taking $T = O(n\kappa_g)$ up to a logarithmic dependence on the condition numbers. As a consequence, the complexity in the strongly convex stochastic setting becomes:

$$\mathcal{C}(\epsilon) = O\left(\kappa_\mathcal{L}\Big(n\tilde{\sigma}_g^2 + \kappa_g(1 \vee \epsilon\kappa_g)\tilde{\sigma}_{g_{yy}}^2 + \tilde{\sigma}_{g_{xy}}^2 + \tilde{\sigma}_f^2\Big)\frac{1}{\epsilon}\log\left(\frac{E_0^{tot} + \mathbb{E}[\mathcal{L}(x_0) - \mathcal{L}^\star]}{\epsilon}\right)\right).$$

In the non-convex setting, the complexity becomes:

$$\mathcal{C}(\epsilon) = O\left(\frac{\kappa_g^4}{\epsilon^2}\Big(n\tilde{\sigma}_g^2 + \kappa_g^3\big(1 \vee \epsilon\mu_g^2\big)\tilde{\sigma}_{g_{yy}}^2 + \kappa_g\tilde{\sigma}_{g_{xy}}^2 + \kappa_g\tilde{\sigma}_f^2\Big)\big(\mathbb{E}[\mathcal{L}(x_k) - \mathcal{L}^\star] + E_0^y + E_0^z\big)\right).$$

The downside of this approach is the dependence on the number $n$ of functions $g_i$ in the total complexity.

## B    PRELIMINARY RESULTS

### B.1    EXPRESSION OF THE GRADIENT

We provide a proof of Proposition 1 which shows that $\mathcal{L}$ is differentiable and provides an expression of its gradient.

*Proof.* Assumption 1 ensures that $y \mapsto g(x, y)$ admits a unique minimizer $y^\star(x)$ defined as the unique solution to the implicit equation $\partial_y g(x, y^\star(x)) = 0$. Moreover, since $g$ is twice continuously differentiable and strongly convex, it follows that $\partial_{yy} g(x, y^\star(x))$ is invertible for any $x \in \mathcal{X}$. Therefore the implicit function theorem (Lang, 2012, Theorem 5.9), ensures that $x \mapsto y^\star(x)$ is continuously differentiable with Jacobian given by $\nabla y^\star(x) = -\partial_{xy} g(x, y^\star(x))\partial_{yy} g(x, y^\star(x))^{-1}$. Hence, by composition of differentiable functions, $\mathcal{L}$ is also differentiable with gradient given by:

$$\nabla \mathcal{L}(x) = \partial_x f(x, y^\star(x)) - \partial_{xy} g(x, y^\star(x))\partial_{yy} g(x, y^\star(x))^{-1}\partial_y f(x, y^\star(x)).$$

We can thus define $z^\star(x, y) = -\partial_{yy} g(x, y)^{-1}\partial_y f(x, y)$ to get the desired expression for $\nabla \mathcal{L}(x)$ and note that $z^\star$ is the solution to (2). $\qquad\square$

### B.2    SMOOTHNESS PROPERTIES OF $\mathcal{L}$, $\mathcal{Y}^\star$, $z^\star$ AND $\Psi$

*Proof of Proposition 6.* **Lipschitz continuity of $x \mapsto y^\star(x)$.** By Assumptions 1 and 3, the implicit function theorem (Lang, 2012, Theorem 5.9) ensures $y^\star(x)$ is differentiable with Jacobian given by:

$$\nabla y^\star(x) = -\partial_{xy} g(x, y^\star(x))(\partial_{yy} g(x, y^\star(x)))^{-1}.$$

Moreover, by Assumption 3, we know that $\partial_y g(x, y)$ is $L_g'$-Lipchitz in $x$ for any $y \in \mathcal{Y}$, hence, $\|\partial_{xy} g(x, y^\star(x))\|_{op}$ is upper-bounded by $L_g'$. Moreover, by Assumption 1, $g$ is $\mu_g$-strongly convex in $y$ uniformly on $\mathcal{X}$. Therefore, it holds that $\left\|\partial_{yy} g(x, y^\star(x))^{-1}\right\|_{op} \leq \mu_g^{-1}$. This allows to deduce that $\|\nabla y^\star(x)\|_{op} \leq \mu_g^{-1}L_g'$, and by application of the fundamental theorem of calculus that:

$$\|y^\star(x) - y^\star(x')\| \leq \mu_g^{-1}L_g'\|x - x'\|.$$

This shows that $y^\star$ is $L_y$-Lipschitz continuous with $L_y := \mu_g^{-1}L_g'$.

**Lipchitz continuity of** $x \mapsto z^\star(x, y)$**.** Let $(x, y)$ and $(x', y')$ be two points in $\mathcal{X} \times \mathcal{Y}$. Recalling the definition of $z^\star(x, y)$ in Proposition 1, it is easy to see that $z^\star(x, y)$ admits the following expression:

$$z^\star(x, y) = -\partial_{yy} g(x, y)^{-1} \partial_y f(x, y). \tag{25}$$

Recalling the expression of $z^\star(x, y)$, the following holds:

$$
\begin{aligned}
z^\star(x, y) - z^\star(x', y') =& \partial_{yy} g(x', y')^{-1} \partial_y f(x', y') - \partial_{yy} g(x, y)^{-1} \partial_y f(x, y) \\
=& \left( \partial_{yy} g(x', y')^{-1} - \partial_{yy} g(x, y)^{-1} \right) \partial_y f(x', y') \\
&+ \partial_{yy} g(x, y)^{-1} (\partial_y f(x', y') - \partial_y f(x, y)) \\
=& \partial_{yy} g(x', y')^{-1} (\partial_{yy} g(x, y) - \partial_{yy} g(x', y')) \partial_{yy} g(x, y)^{-1} \partial_y f(x', y') \\
&+ \partial_{yy} g(x, y)^{-1} (\partial_y f(x', y') - \partial_y f(x, y))
\end{aligned}
$$

Hence, by taking the norm of the above quantity a triangular inequality followed by operator inequalities, it follows that:

$$
\begin{aligned}
\| z^\star(x, y) - z^\star(x', y') \| \leq & \left\| H_2^{-1} \right\|_{op} \| H_1 - H_2 \|_{op} \left\| H_1^{-1} \right\|_{op} \| \partial_y f(x', y') \| \\
&+ \left\| H_1^{-1} \right\| \| \partial_y f(x', y') - \partial_y f(x, y) \|.
\end{aligned}
$$

where we introduced $H_1 = \partial_{yy} g(x, y)$ and $H_2 = \partial_{yy} g(x', y')$ for conciseness. Using Assumption 1, we can upper-bound $\left\| H_1^{-1} \right\|_{op}$ and $\left\| H_2^{-1} \right\|_{op}$ by $\mu_g^{-1}$. By Assumption 3, we know that $\| H_1 - H_2 \|_{op} \leq M_g \| (x, y) - (x', y') \|$. Finally by Assumption 2, we also have that $\| \partial_y f(x', y') - \partial_y f(x, y) \| \leq L_f \| (x, y) - (x', y') \|$ and that $\| \partial_y f(x', y') \| \leq B$ ensuring that:

$$\| z^\star(x, y) - z^\star(x', y') \| \leq \left( \mu_g^{-2} M_g B + \mu_g^{-1} L_f \right) \| (x, y) - (x', y') \|.$$

Hence, we conclude that $z^\star$ is $L_z$-Lipchitz continuous with $L_z$ defined as in (13).

**boundedness of** $z^\star(x, y)$ Recalling the expression of $z^\star$ in (25), it is easy to see that $\| z^\star(x, y) \|$ is upper-bounded by $\mu_g^{-1} B$ since $\partial_{yy} g(x, y)$ is $\mu_g$-strongly convex in $y$ by Assumption 1 and $\partial_y f(x, y)$ is bounded by $B$ by Assumption 2.

**Regularity of** $\Psi$**.**

$$
\begin{aligned}
\Psi(x, y, z) - \Psi(x', y', z') =& \partial_x f(x, y) - \partial_x f(x', y') + \partial_{xy} g(x, y) z - \partial_{xy} g(x', y') z' \\
=& \partial_x f(x, y) - \partial_x f(x', y') + \partial_{xy} g(x, y)(z - z') \\
&+ (\partial_{xy} g(x, y) - \partial_{xy} g(x', y')) z'.
\end{aligned}
$$

By taking the norm of the above expression and applying a triangular inequality followed by operator inequalities, it follows that:

$$
\begin{aligned}
\| \Psi(x, y, z) - \Psi(x', y', z') \| \leq & \| \partial_x f(x, y) - \partial_x f(x', y') \| + \| \partial_{xy} g(x, y) \|_{op} \| z - z' \| \tag{26} \\
&+ \| \partial_{xy} g(x, y) - \partial_{xy} g(x', y') \|_{op} \| z' \| \\
\leq & L_f (\| x - x' \| + \| y - y' \|) + L_g' \| z - z' \| \\
&+ M_g \| z' \| (\| x - x' \| + \| y - y' \|).
\end{aligned}
$$

To get the first term of the last inequality above, we used that $\partial_x f$ is $L_f$-Lipschitz by Assumption 2. To get the second term, we used that $\partial_{xy} g(x, y)$ is bounded since $\partial_y g(x, y)$ is $L_g'$-Lipschitz by Assumption 3. Finally, for the last term, we used that $\partial_{xy} g(x, y)$ is $M_g$-Lipschitz by Assumption 3.

By choosing $x' = x$, $y' = y^\star(x)$ and $z' = z^\star(x, y^\star(x))$, it is easy to see from Proposition 1 that $\Psi(x, y^\star(x), z^\star(x, y^\star(x))) = \nabla \mathcal{L}(x)$. Hence, applying the above inequality yields:

$$
\begin{aligned}
\| \Psi(x, y, z) - \nabla \mathcal{L}(x) \| \leq & (L_f + M_g \| z^\star(x, y^\star(x)) \|) \| y - y^\star(x) \| + L_g' \| z - z^\star(x, y^\star(x)) \| \\
\leq & (L_f + M_g \| z^\star(x, y^\star(x)) \|) \| y - y^\star(x) \| + L_g' \| z - z^\star(x, y) \| \\
&+ L_g' \| z^\star(x, y) - z^\star(x, y^\star(x)) \|
\end{aligned}
$$

As shown earlier, $\| z^\star(x, y^\star(x)) \|$ is upper-bounded by $\mu_g^{-1} B$, while $\| z^\star(x, y) - z^\star(x, y^\star(x)) \|$ is bounded by $L_z \| y - y^\star(x) \|$. This allows to conclude that $\| \Psi(x, y, z) - \nabla \mathcal{L}(x) \| \leq L_\psi$ with $L_\psi$ defined in (13).

**Lipschitz continuity of $x \mapsto \nabla\mathcal{L}(x)$.** We apply (26) with $(y, z) = (y^\star(x), z^\star(x, y^\star(x)))$ and $(y', z') = (y^\star(x'), z^\star(x, y^\star(x')))$ which yields:

$$
\begin{aligned}
\|\nabla\mathcal{L}(x) - \nabla\mathcal{L}(x')\| \leq & (L_f + M_g \|z^\star(x', y^\star(x'))\|)(\|x - x'\| + \|y^\star(x) - y^\star(x')\|) \\
& + L_g' \|z^\star(x, y^\star(x)) - z^\star(x', y^\star(x'))\| \\
\leq & (L_f + M_g \mu_g^{-1} B + L_g' L_z)(1 + L_y)\|x - x'\|,
\end{aligned}
$$

where we used that $\|z^\star(x', y^\star(x'))\|$ is upper-bounded by $\mu_g^{-1} B$, $z^\star$ is $L_z$-Lipschitz and $y^\star$ is $L_y$-Lipschitz. Hence, $\nabla\mathcal{L}$ is $L$-Lipschitz continuous, with $L$ as given by (13). $\qquad\square$

## B.3 Convergence of the iterates of algorithms $\mathcal{A}_k$ and $\mathcal{B}_k$

*Proof.* **Controlling the iterates $y^t$ of $\mathcal{A}_k$.**

Consider a new batch $\mathcal{D}_g$ of samples $\xi$. We have by definition of the update equation of $y^t$ that:

$$
\begin{aligned}
\|y^t - y^\star(x_k)\|^2 = & \|y^{t-1} - y^\star(x_k)\|^2 + \alpha_k^2 \|\partial_y \hat{g}(x_k, y^{t-1}, \mathcal{D}_g)\|^2 \\
& - 2\alpha_k \partial_y \hat{g}(x_k, y^{t-1}, \mathcal{D}_g)^\top (y^{t-1} - y^\star(x_k))
\end{aligned}
$$

Taking the expectation conditionally on $x_k$ and $y^{t-1}$, we get:

$$
\begin{aligned}
\mathbb{E}\Big[\|y^t - y^\star(x_k)\|^2 \Big| x_k, y^{t-1}\Big] = & (1 - \alpha_k \mu_g)\|y^{t-1} - y^\star(x_k)\|^2 \\
& + \alpha_k^2 \mathbb{E}\Big[\|\partial_y \hat{g}(x_k, y^{t-1}, \mathcal{D}_g) - \partial_y g(x_k, y^{t-1})\|^2 \Big| x_k, y^{t-1}\Big] \\
& - 2\alpha_k \partial_y g(x_k, y^{t-1})^\top \Big(y^{t-1} - y^\star(x_k) - \frac{\alpha_k}{2}\partial_y g(x_k, y^{t-1})\Big) \\
\leq & (1 - \alpha_k \mu_g)\|y^{t-1} - y^\star(x_k)\|^2 + 2\alpha_k^2 \sigma_g^2
\end{aligned}
$$

The first line uses that $\partial_y \hat{g}(x_k, y^{t-1}, \mathcal{D}_g)$ is an unbiased estimator of $\partial_y g(x_k, y^{t-1})$. For the second line, we use Assumption 4 which allows to upper-bound the variance of $\partial_y \hat{g}$ by $\sigma_g^2$. Moreover, since $g$ is convex and $L_g$-smooth and since $\alpha_k \leq L_g^{-1}$, it follows that the last term in the above inequality is non-positive and can thus be upper-bounded by $0$. By unrolling the resulting inequality recursively for $1 < t \leq k$, we obtain the desired result.

**Controlling the iterates $z^n$ of $\mathcal{B}_k$.** The poof follows by direct application of Proposition 15 with $\beta = \beta_k$ and the following choices for $A_n$, $A$, $\hat{b}$, $b$:

$$
\begin{aligned}
A_n & = \partial_{yy} \hat{g}(x_k, y_k, \mathcal{D}_{g_{yy}}), & \hat{b} & = \partial_y \hat{f}(x_k, y_k, \mathcal{D}_f) \\
A & = \partial_{yy} g(x_k, y_k) & b & = \partial_y f(x_k, y_k).
\end{aligned}
$$

This directly yields the following inequalities:

$$
\begin{aligned}
\mathbb{E}\Big[\|z_k - z^\star(x_k, y_k)\|^2\Big] & \leq \tilde{\Pi}_k \mathbb{E}\Big[\|z_k^0 - z^\star(x_k, y_k)\|^2\Big] + \tilde{R}_k^z, \\
\mathbb{E}\Big[\|\bar{z}_k - z^\star(x_k, y_k)\|^2\Big] & \leq \tilde{\Pi}_k \mathbb{E}\Big[\|z_k^0 - z^\star(x_k, y_k)\|^2\Big].
\end{aligned}
$$

where $\tilde{\Pi}_k$ and $\tilde{R}_k^z$ are given by:

$$
\tilde{\Pi}_k := (1 - \beta_k \mu_g)^N, \qquad \tilde{R}_k^z := \beta_k^2 \Big(\sigma_g^2 \mathbb{E}\Big[\|z^\star(x_k, y_k)\|^2\Big] + 3\Big(N \wedge \frac{1}{\beta_k \mu_g}\Big)\sigma_f^2\Big)\Big(N \wedge \frac{1}{\beta_k \mu_g}\Big).
$$

First we have that $\tilde{\Pi}_k \leq \Pi_k$. Moreover, Proposition 6, we have that $\|z^\star(x_k, y_k)\| \leq B\mu_g^{-1}$ hence, $\tilde{R}_k^z \leq R_k^z$ thus yielding the desired inequalities. Finally (15a) also follows similarly using (45) from Proposition 15. $\qquad\square$

### B.4 CONTROLLING THE BIAS AND VARIANCE $E_k^\psi$ AND $V_k^\psi$

*Proof of Proposition 4* . Recall that the expressions of $E_k^\psi$ and $V_k^\psi$ in (6) involves the conditional expectation $\mathbb{E}_k\left[\hat\psi_k\right]$ knowing $x_k, y_k$ and $z_{k-1}$. This can be also expressed using $\Psi$ as follows:

$$\begin{aligned}
\mathbb{E}_k\left[\hat\psi_k\right] &= \mathbb{E}\left[\mathbb{E}\left[\hat\psi_k | x_k, y_k, z_k\right] | x_k, y_k, z_{k-1}\right] \\
&= \mathbb{E}[\Psi(x_k, y_k, z_k) | x_k, y_k, z_{k-1}] \\
&= \mathbb{E}[\Psi(x_k, y_k, \mathbb{E}[z_k || x_k, y_k, z_{k-1}])]
\end{aligned}$$

where we used the tower property for conditional expectations in the first line, then the fact that the expectation of $\psi_k$ conditionally on $x_k, y_k$ and $z_k$ is simply $\Psi(x_k, y_k, z_k)$. Finally, for the last line, we use the independence of the noise and the linearity of $\Psi$ w.r.t. the last variable. In all what follows, we write $\bar z_k = \mathbb{E}[z_k | x_k, y_k, z_{k-1}]$ which is the same object as defined in Proposition 7. We then treat $E_k^\psi$ and $V_k^\psi$ separately.

**Bounding $E_k^\psi$.** Using Propositions 6 and 7 we directly get the desired inequality:

$$\begin{aligned}
E_k^\psi &\leq 2L_\psi^2\left(\mathbb{E}\left[\|y_k - y^\star(x_k)\|^2\right] + \mathbb{E}\left[\|\bar z_k - z^\star(x_k, y_k)\|^2\right]\right) \\
&\leq 2L_\psi^2(\Lambda_k E_k^y + \Pi_k E_k^z + R_k^y)
\end{aligned}$$

**Bound on $V_k^\psi$.** We decompose $V_k^\psi$ into a sum of three terms $W_k, W_k'$ and $W_k''$ given by:

$$W_k := \mathbb{E}\left[\left\|\partial_x \hat f(x_k, y_k, \mathcal{D}_f) - \partial_x f(x_k, y_k)\right\|^2\right],$$

$$W_k' := \mathbb{E}\left[\left\|\partial_{xy}\hat g\left(x_k, y_k, \tilde\xi_{N+1,k}\right)z_k - \partial_{xy}g(x_k, y_k)\bar z_k\right\|^2\right]$$

$$W_k'' := \mathbb{E}\left[\left(\partial_x \hat f(x_k, y_k, \mathcal{D}_f) - \partial_x f(x_k, y_k)\right)^\top \partial_{xy}g(x_k, y_k)(z_k - \bar z_k)\right].$$

where we used that $\tilde\xi_{N+1,k}$ is independent from $z_k$ and $\mathcal{D}_f$ to get the last term. Hence, using Assumption 4 to bound the first term of the above relation, we get $V_k^\psi \leq \sigma_f^2 + W_k' + 2W_k''$. Thus, it remains to control each of $W_k'$ and $W_k''$.

**Bound on $W_k''$.** Using that $\mathcal{D}_f$ is independent from $\tilde\xi_{n,k}$, we can apply Proposition 14 to write:

$$W_k'' = \beta_k \mathbb{E}\left[\partial_x(\hat f - f)(x_k, y_k, \mathcal{D}_f)^\top \partial_{xy}g(x_k, y_k)\left(\sum_{t=1}^N (I - \beta_k A)^{N-t}\right)\partial_y(\hat f - f)(x_k, y_k, \mathcal{D}_f)\right]$$

where we used the simplifying notion $(\hat f - f)(x_k, y_k, \mathcal{D}_f) = \hat f(x_k, y_k, \mathcal{D}_f) - f(x_k, y_k)$. Using Assumption 3 to bound $\|\partial_{xy}g(x_k, y_k)\|_{op}$ by $L_g'$, Assumption 1 to upper-bound $\left\|\left(\sum_{t=1}^N (I - \beta_k A)^{N-t}\right)\right\|_{op}$ by $\left(\sum_{t=1}^N (1 - \beta_k \mu_g)^{N-t}\right)$ we get

$$\begin{aligned}
|W_k''| &\leq \beta_k L_g' \sum_{t=0}^{N-1}(1 - \beta_k\mu_g)^t \mathbb{E}\left[\left|\partial_x(\hat f - f)(x_k, y_k, \mathcal{D}_f)^\top \partial_y(\hat f - f)(x_k, y_k, \mathcal{D}_f)\right|\right] \\
&\leq L_g'\mu_g^{-1}\mathbb{E}\left[\left|\partial_x(\hat f - f)(x_k, y_k, \mathcal{D}_f)^\top \partial_y(\hat f - f)(x_k, y_k, \mathcal{D}_f)\right|\right] \\
&\leq L_g'\mu_g^{-1}\mathbb{E}\left[\left\|\partial_x(\hat f - f)(x_k, y_k, \mathcal{D}_f)^2\right\|\right]^{\frac{1}{2}}\mathbb{E}\left[\left\|\partial_y(\hat f - f)(x_k, y_k, \mathcal{D}_f)\right\|^2\right]^{\frac{1}{2}} \leq L_g'\mu_g^{-1}\sigma_f^2
\end{aligned}$$

where we used that $\sum_{t=0}^{N-1}(1 - \beta_k\mu_g) \leq \frac{1}{\beta_k\mu_g}$ for the second line, Cauchy-Schwarz inequality to get the third line and Assumption 4 to get the last line.

**Bound on $W_k'$** Using that $\tilde{\xi}_{N+1,k}$ is independent from $z_k$, we write:

$$
\begin{aligned}
W_k' =& \mathbb{E}\left[\left\|\partial_{xy}(\hat{g}-g)\left(x_k,y_k,\tilde{\xi}_{N+1,k}\right)z_k\right\|^2\right] + \mathbb{E}\left[\left\|\partial_{xy}g(x_k,y_k)(z_k-\bar{z}_k)\right\|^2\right] \\
\overset{(i)}{\leq}& \sigma_{g_{xy}}^2\mathbb{E}\left[\|z_k\|^2\right] + (L_g')^2\mathbb{E}\left[\|z_k-\bar{z}_k\|^2\right] \\
\overset{(ii)}{\leq}& 2\sigma_{g_{xy}}^2\left(\mathbb{E}\left[\|z_k-z^\star(x_k,y_k)\|^2\right] + \mathbb{E}\left[\|z^\star(x_k,y_k)\|^2\right]\right) + (L_g')^2\mathbb{E}\left[\|z_k-\bar{z}_k\|^2\right] \\
\overset{(iii)}{\leq}& 2\sigma_{g_{xy}}^2\mathbb{E}\left[\|z_k-z^\star(x_k,y_k)\|^2\right] + (L_g')^2\mathbb{E}\left[\|z_k-\bar{z}_k\|^2\right] + 2\sigma_{g_{xy}}^2 B^2\mu_g^{-2} \\
\overset{(iv)}{\leq}& 2\sigma_{g_{xy}}^2\left(\Pi_k\mathbb{E}\left[\|z_k^0-z^\star(x_k,y_k)\|^2 + R_k^z\right]\right) \\
& + (L_g')^2\left(4\sigma_{g_{yy}}^2\mu_g^{-2}\Pi_k\mathbb{E}\left[\|z_k^0-z^\star(x_k,y_k)\|^2\right] + 2R_k^z\right) + 2\sigma_{g_{xy}}^2 B^2\mu_g^{-2} \\
\leq& 2\left(\sigma_{g_{xy}}^2 + 2(L_g')^2\mu_g^{-2}\sigma_{g_{yy}}^2\right)\Pi_k\mathbb{E}\left[\|z_k^0-z^\star(x_k,y_k)\|^2\right] \\
& + 2\left(\sigma_{g_{xy}}^2 + (L_g')^2\right)R_k^z + 2\sigma_{g_{xy}}^2 B^2\mu_g^{-2}
\end{aligned}
$$

(i) follows from Assumptions 3 and 5, (ii) uses that $\|z_k\|^2 \leq 2\left(\|z_k-z\|^2 + \|z\|^2\right)$, (iii) uses that $\|z^\star(x_k,y_k)\| \leq B\mu_g^{-1}$ by Proposition 6. Finally (iv) follows by application of Proposition 7. We further have by definition of $R_k^z$ that:

$$
R_k^z \leq B^2 L_g^{-1}\mu_g^{-3}\sigma_{g_{yy}}^2 + 3\mu_g^{-2}\sigma_f^2 \tag{29}
$$

Combining the inequalities on $W_k'$, $W_k''$ and (29), we get that $V_k^\psi \leq w_x^2 + \sigma_x^2\Pi_k E_k^z$, with $w_x^2$ and $\sigma_x^2$ given by (16). $\qquad\square$

## C  GENERAL ANALYSIS OF AMIGO

### C.1  ANALYSIS OF THE OUTER-LOOP

*Proof of Proposition 2.* We treat both cases $\mu \geq 0$ and $\mu < 0$ separately. For simplicity we denote by $\mathbb{E}_k$ the conditional expectation knowing the iterates $x_k$, $y_k$ and $z_{k-1}$ and write $\psi_k = \mathbb{E}_k\left[\hat{\psi}_k\right]$.

**Case $\mu \geq 0$.** Recall that $E_k^x$ is given by:

$$
E_k^x = \frac{\eta_k}{2}\mathbb{E}\left[\|x_k-x^\star\|^2\right] + (1-u)\mathbb{E}[\mathcal{L}(x_k)-\mathcal{L}^\star].
$$

For simplicity define $\epsilon_k = u\delta_k + (1-u)$, $e_k = (1-u)(\mathcal{L}(x_k)-\mathcal{L}^\star) + \frac{\eta_k}{2}\|x_k-x^\star\|^2$ and $e_k' = u\delta_k(\mathcal{L}(x_k)-\mathcal{L}^\star)$. It is then easy to see that $\mathbb{E}[e_k]$ is equal to the l.h.s of (10), i.e. $\mathbb{E}[e_k] = E_k^x$. We

will start by bounding the difference between two successive iterates of $e_k$:

$$
\begin{aligned}
e'_k + e_k - e_{k-1} \leq & u\delta_k(\mathcal{L}(x_k) - \mathcal{L}^\star) + (1-u)(\mathcal{L}(x_k) - \mathcal{L}(x_{k-1})) \\
& + \frac{\eta_k}{2}\|x_k - x^\star\|^2 - \frac{\eta_{k-1}}{2}\|x_{k-1} - x^\star\|^2 \\
\stackrel{(i)}{\leq} & u\delta_k(\mathcal{L}(x_k) - \mathcal{L}^\star) + (1-u)(\mathcal{L}(x_k) - \mathcal{L}(x_{k-1})) \\
& - \frac{\delta_k\eta_{k-1}}{2}\|x_{k-1} - x^\star\|^2 + \delta_k\left(\frac{\mu}{2}\|x_{k-1} - x^\star\|^2 - \nabla\mathcal{L}(x_{k-1})^\top(x_{k-1} - x^\star)\right) \\
& + \frac{\eta_k}{2}\left(\gamma_k^2\left\|\hat{\psi}_{k-1}\right\|^2 - 2\gamma_k\left(\hat{\psi}_{k-1} - \nabla\mathcal{L}(x_{k-1})\right)^\top(x_{k-1} - x^\star)\right) \\
\stackrel{(ii)}{\leq} & u\delta_k(\mathcal{L}(x_k) - \mathcal{L}^\star) + (1-u)(\mathcal{L}(x_k) - \mathcal{L}(x_{k-1})) \\
& - \frac{\delta_k\eta_{k-1}}{2}\|x_{k-1} - x^\star\|^2 - \delta_k(\mathcal{L}(x_{k-1}) - \mathcal{L}^\star) \\
& + \frac{\eta_k}{2}\left(\gamma_k^2\left\|\hat{\psi}_{k-1}\right\|^2 - 2\gamma_k\left(\hat{\psi}_{k-1} - \nabla\mathcal{L}(x_{k-1})\right)^\top(x_{k-1} - x^\star)\right) \\
\leq & (u\delta_k + (1-u))(\mathcal{L}(x_k) - \mathcal{L}(x_{k-1})) \\
& - \frac{\delta_k\eta_{k-1}}{2}\|x_{k-1} - x^\star\|^2 - \delta_k(1-u)(\mathcal{L}(x_{k-1}) - \mathcal{L}^\star) \\
& + \frac{\eta_k}{2}\left(\gamma_k^2\left\|\hat{\psi}_{k-1}\right\|^2 - 2\gamma_k\left(\hat{\psi}_{k-1} - \nabla\mathcal{L}(x_{k-1})\right)^\top(x_{k-1} - x^\star)\right) \\
\leq & \epsilon_k(\mathcal{L}(x_k) - \mathcal{L}(x_{k-1})) - \delta_k e_{k-1} \\
& + \frac{\eta_k}{2}\left(\gamma_k^2\left\|\hat{\psi}_{k-1}\right\|^2 - 2\gamma_k\left(\hat{\psi}_{k-1} - \nabla\mathcal{L}(x_{k-1})\right)^\top(x_{k-1} - x^\star)\right) \\
\stackrel{(iii)}{\leq} & -\delta_k e_{k-1} + \epsilon_k\nabla\mathcal{L}(x_{k-1})^\top(x_k - x_{k-1}) + \frac{\epsilon_k L}{2}\|x_k - x_{k-1}\|^2 \\
& + \frac{\eta_k}{2}\left(\gamma_k^2\left\|\hat{\psi}_{k-1}\right\|^2 - 2\gamma_k\left(\hat{\psi}_{k-1} - \nabla\mathcal{L}(x_{k-1})\right)^\top(x_{k-1} - x^\star)\right) \\
= & -\delta_k e_{k-1} - \gamma_k\epsilon_k\nabla\mathcal{L}(x_{k-1})^\top\hat{\psi}_{k-1} + \frac{\epsilon_k\gamma_k^2 L}{2}\left\|\hat{\psi}_{k-1}\right\|^2 \\
& + \frac{\eta_k}{2}\left(\gamma_k^2\left\|\hat{\psi}_{k-1}\right\|^2 - 2\gamma_k\left(\hat{\psi}_{k-1} - \nabla\mathcal{L}(x_{k-1})\right)^\top(x_{k-1} - x^\star)\right).
\end{aligned}
$$

(i) follows from the update expression $x_k = x_{k-1} - \gamma_k\hat{\psi}_{k-1}$, (ii) follows from the convexity of $\mathcal{L}$ and (iii) follows by $L$-smoothness of $\mathcal{L}$. Taking the expectation conditionally on the randomness at iteration $k-1$ and using that $\mathbb{E}_{k-1}\left[\hat{\psi}_{k-1}\right] = \psi_{k-1}$, we therefore get

$$
\begin{aligned}
\mathbb{E}_{k-1}[e'_k + e_k - e_{k-1}] \leq & -\delta_k e_{k-1} - \gamma_k\epsilon_k\nabla\mathcal{L}(x_{k-1})^\top\psi_{k-1} + \frac{\gamma_k}{2}(\delta_k + \epsilon_k)\mathbb{E}_{k-1}\left[\left\|\hat{\psi}_{k-1}\right\|^2\right] \\
& - \delta_k(\psi_{k-1} - \nabla\mathcal{L}(x_{k-1}))^\top(x_{k-1} - x^\star) \\
= & -\delta_k e_{k-1} + \gamma_k s_k\left(\mathbb{E}_{k-1}\left[\left\|\hat{\psi}_{k-1} - \psi_{k-1}\right\|^2\right] + \|\psi_{k-1} - \nabla\mathcal{L}(x_{k-1})\|^2\right) \\
& - \delta_k(\psi_{k-1} - \nabla\mathcal{L}(x_{k-1}))^\top(x_{k-1} - \gamma_k\nabla\mathcal{L}(x_{k-1}) - x^\star) \\
& - \frac{\gamma_k}{2}(\epsilon_k - \delta_k)\|\nabla\mathcal{L}(x_{k-1})\|^2 \\
\stackrel{(i)}{\leq} & -\delta_k e_{k-1} + \gamma_k s_k\left(\mathbb{E}_{k-1}\left[\left\|\hat{\psi}_{k-1} - \psi_{k-1}\right\|^2\right] + \|\psi_{k-1} - \nabla\mathcal{L}(x_{k-1})\|^2\right) \\
& - \delta_k(\psi_{k-1} - \nabla\mathcal{L}(x_{k-1}))^\top(x_{k-1} - \gamma_k\nabla\mathcal{L}(x_{k-1}) - x^\star)
\end{aligned}
$$

where (i) follows from $\delta_k \leq \epsilon_k$ since by construction $\delta_k \leq 1$. Taking the expectation w.r.t. all the randomness and applying Cauchy-Schwarz inequality to the last term yields the following inequality:

$$u\delta_k(\mathcal{L}(x_k) - \mathcal{L}^\star) + E_k^x \leq (1 - \delta_k)E_{k-1}^x + \gamma_k s_k \left( V_{k-1}^\psi + E_{k-1}^\psi \right) \tag{30}$$
$$+ \delta_k \left( E_{k-1}^\psi \right)^{\frac{1}{2}} \mathbb{E}\left[ \|x_{k-1} - \gamma_k \nabla \mathcal{L}(x_{k-1}) - x^\star\|^2 \right]^{\frac{1}{2}}.$$

Since $\mathcal{L}$ is convex, we have the inequality: $\|x_{k-1} - \gamma_k \nabla \mathcal{L}(x_{k-1}) - x^\star\|^2 \leq \|x_{k-1} - x^\star\|^2$. Hence, we can deduce that:

$$\delta_k \|x_{k-1} - \gamma_k \nabla \mathcal{L}(x_{k-1}) - x^\star\|^2 \leq \delta_k \|x_{k-1} - x^\star\|^2 \leq 2\gamma_k \eta_k \eta_{k-1}^{-1} E_{k-1}^x \leq 2\gamma_k E_{k-1}^x,$$

where we used that $\eta_k$ is non-increasing by construction. Combining the above inequality with (30) yields:

$$F_k + E_k^x \leq (1 - \delta_k)E_{k-1}^x + \gamma_k s_k \left( V_{k-1}^\psi + E_{k-1}^\psi \right) + \sqrt{2}\gamma_k^{\frac{1}{2}}\delta_k^{\frac{1}{2}} \left( E_{k-1}^\psi \right)^{\frac{1}{2}} \left( E_{k-1}^x \right)^{\frac{1}{2}}.$$

**Case $\mu < 0$.** Recall that for $\mu < 0$, we set $E_k^x = \frac{1}{L}\mathbb{E}\left[ \|\nabla \mathcal{L}(x_k)\|^2 \right]$. Using that $\mathcal{L}$ is $L$-smooth, we have that:

$$\mathcal{L}(x_k) - \mathcal{L}(x_{k-1}) \leq \nabla \mathcal{L}(x_{k-1})^\top (x_k - x_{k-1}) + \frac{L}{2}\|x_k - x_{k-1}\|^2$$
$$\leq -\gamma_k \nabla \mathcal{L}(x_{k-1})^\top \hat{\psi}_{k-1} + \frac{L\gamma_k^2}{2}\left\|\hat{\psi}_{k-1}\right\|^2$$
$$\leq -\gamma_k \|\nabla \mathcal{L}(x_{k-1})\|^2 - \gamma_k \nabla \mathcal{L}(x_{k-1})^\top \left( \hat{\psi}_{k-1} - \nabla \mathcal{L}(x_{k-1}) \right)$$
$$+ \frac{L\gamma_k^2}{2}\left( \left\|\hat{\psi}_{k-1} - \psi_{k-1}\right\|^2 + 2\left(\hat{\psi}_{k-1} - \psi_{k-1}\right)^\top \psi_{k-1} + \|\psi_{k-1}\|^2 \right).$$

Taking the expectation w.r.t. all randomness in the algorithm in the above inequality, we get:

$$\mathbb{E}[\mathcal{L}(x_k) - \mathcal{L}(x_{k-1})] \leq -\gamma_k \mathbb{E}\left[ \|\nabla \mathcal{L}(x_{k-1})\|^2 \right] - \gamma_k \mathbb{E}\left[ \nabla \mathcal{L}(x_{k-1})^\top (\psi_{k-1} - \nabla \mathcal{L}(x_{k-1})) \right]$$
$$+ \frac{L\gamma_k^2}{2}\left( \mathbb{E}\left[ \left( \left\|\hat{\psi}_{k-1} - \psi_{k-1}\right\|^2 \right) \right] + \mathbb{E}\left[ \|\psi_{k-1}\|^2 \right] \right)$$
$$= -\gamma_k(1 - \frac{L\gamma_k}{2})\mathbb{E}\left[ \|\nabla \mathcal{L}(x_{k-1})\|^2 \right] + \frac{L\gamma_k^2}{2}\left( V_{k-1}^\psi + E_{k-1}^\psi \right)$$
$$- \gamma_k(1 - L\gamma_k)\mathbb{E}\left[ \nabla \mathcal{L}(x_{k-1})^\top (\psi_{k-1} - \nabla \mathcal{L}(x_{k-1})) \right]$$
$$\leq -\frac{\gamma_k}{2}\mathbb{E}\left[ \|\nabla \mathcal{L}(x_{k-1})\|^2 \right] + \frac{L\gamma_k^2}{2}\left( V_{k-1}^\psi + E_{k-1}^\psi \right)$$
$$+ \gamma_k \mathbb{E}\left[ \|\nabla \mathcal{L}(x_{k-1})\|^2 \right]^{\frac{1}{2}} \left( E_{k-1}^\psi \right)^{\frac{1}{2}}.$$
$$= -\delta_k E_{k-1}^x + \frac{\delta_k \gamma_k}{2}\left( V_{k-1}^\psi + E_{k-1}^\psi \right) + \sqrt{2}\delta_k^{\frac{1}{2}}\gamma_k^{\frac{1}{2}} \left( E_{k-1}^x \right)^{\frac{1}{2}} \left( E_{k-1}^\psi \right)^{\frac{1}{2}}.$$

where we used that $1 - \frac{L\gamma_k}{2} \geq \frac{1}{2}$ and $0 \leq 1 - L\gamma_k \leq 1$ to get the last inequality. Using the definition of $F_k$ yields an inequality of the form:

$$F_k + E_k^x \leq (1 - \delta_k)E_{k-1}^x + \gamma_k s_k \left( V_{k-1}^\psi + E_{k-1}^\psi \right) + \sqrt{2}\gamma_k^{\frac{1}{2}}\delta_k^{\frac{1}{2}} \left( E_{k-1}^\psi \right)^{\frac{1}{2}} \left( E_{k-1}^x \right)^{\frac{1}{2}}.$$

Hence, in both cases $\mu \geq 0$ and $\mu < 0$ we get an inequality of the of the same form, but with different expressions for $F_k$ and $s_k$. We get the desired result using Young's inequality, to upper-bound the last term in the r.h.s. of the above inequality. More precisely, we use that for any $0 < \rho_k < 1$:

$$\sqrt{2}\gamma_k^{\frac{1}{2}}\delta_k^{\frac{1}{2}} \left( E_{k-1}^\psi \right)^{\frac{1}{2}} \left( E_{k-1}^x \right)^{\frac{1}{2}} \leq \frac{1}{2}\rho_k \delta_k E_{k-1}^x + \rho_k^{-1}\gamma_k E_{k-1}^\psi.$$

$\square$

### C.2 INNER-LEVEL ERROR BOUND

In this section we prove Proposition 5 which controls the evolutions of the warm-start errors $E_k^y$ and $E_k^z$. As a first step, in Proposition 12, we provide a result controlling the mean squared error between two successive iterates $x_{k-1}$, $x_k$ and $y_{k-1}$, $y_k$ which will be used in the proof of Proposition 5.

**Proposition 12** (Control of the increments of $x_k$ and $y_k$). *Consider $\zeta_k$, $\phi_k$ and $\tilde{R}_k^y$ as defined in Proposition 8 for some fixed $0 \leq v \leq 1$. Then, the following holds:*

$$\gamma_k^2 \mathbb{E}\left[\left\|\hat{\psi}_{k-1}\right\|^2\right] = \mathbb{E}\left[\|x_k - x_{k-1}\|^2\right] \leq \gamma_k^2\left(V_{k-1}^\psi + 2E_{k-1}^\psi + 2\zeta_k E_{k-1}^x\right)$$

$$\mathbb{E}\left[\|y_k - y_{k-1}\|^2\right] \leq 2\phi_k E_k^y + 2\tilde{R}_k^y$$

*Proof.* Proof of Proposition 12 We prove each inequality separately.

**Increments of $x_k$.** By the update equation, we have that $x_k = x_{k-1} - \gamma_k \hat{\psi}_{k-1}$, hence we only need to control $\mathbb{E}\left[\left\|\hat{\psi}_{k-1}\right\|^2\right]$. We have the following:

$$\mathbb{E}\left[\left\|\hat{\psi}_{k-1}\right\|^2\right] \leq \mathbb{E}\left[\left\|\hat{\psi}_{k-1} - \psi_{k-1}\right\|^2\right] + 2\mathbb{E}\left[\|\psi_{k-1} - \nabla\mathcal{L}(x_{k-1})\|^2\right] + 2\mathbb{E}\left[\|\nabla\mathcal{L}(x_{k-1})\|^2\right]$$

$$= V_{k-1}^\psi + 2E_{k-1}^\psi + 2\mathbb{E}\left[\|\nabla\mathcal{L}(x_{k-1})\|^2\right].$$

In the case ($\mu < 0$), we have $E_{k-1}^x = \frac{1}{2L}\mathbb{E}\left[\|\nabla\mathcal{L}(x_{k-1})\|^2\right]$, hence by setting $\zeta_k := 2L$, we get the desired inequality. In the convex case ($\mu \geq 0$), since $\mathcal{L}$ is $L$-smooth, we have that:

$$\|\nabla\mathcal{L}(x_{k-1})\|^2 \leq 2L(\mathcal{L}(x_{k-1}) - \mathcal{L}^\star) \leq 2L(1-u)^{-1}E_{k-1}^x,$$

provided that $u < 1$. We also have that $(\mathcal{L}(x_{k-1}) - \mathcal{L}^\star) \leq \frac{L}{2}\|x_{k-1} - x^\star\|^2 \leq L\eta_{k-1}^{-1}E_{k-1}^x$ which yields $\|\nabla\mathcal{L}(x_{k-1})\|^2 \leq 2L^2\eta_{k-1}^{-1}E_{k-1}^x$. Hence, we can set $\zeta_k = 2L\min\left((1-u)^{-1}, L\eta_{k-1}^{-1}\right)$.

**Increments of $y_k$.** Denoting by $\mathcal{D}_g^t$ a batch of samples at time iteration $t$ of algorithm $\mathcal{A}_k$ and using the update equation of $y^t$ we get the following inequality by application of the triangular inequality:

$$\mathbb{E}\left[\|y_k - y_{k-1}\|^2\right]^{\frac{1}{2}} \leq \alpha_k \sum_{t=0}^{T-1} \mathbb{E}\left[\|\partial_y\hat{g}(x_k, y^t, \mathcal{D}_g^t)\|^2\right]^{\frac{1}{2}} \leq \alpha_k \sum_{t=0}^{T-1}\left(\sigma_g^2 + L_g^2\mathbb{E}\left[\|y^t - y^\star(x_k)\|^2\right]\right)^{\frac{1}{2}}$$

$$\leq \alpha_k \sum_{t=0}^{T-1}\left(\sigma_g^2 + L_g^2 R_k + L_g^2\Lambda_{t,k}E_k^y\right)^{\frac{1}{2}} \leq \alpha_k T\left(\sigma_g^2\left(1 + 2L_g^2\alpha_k\mu_g^{-1}\right) + L_g^2 E_k^y\right)^{\frac{1}{2}}$$

where we applied Proposition 7 for every $0 < t \leq T-1$ to get the second line with $\Lambda_{t,k} := (1 - \alpha_k\mu_g)^t$. This directly implies the following bound:

$$\mathbb{E}\left[\|\tilde{y}_k - y_{k-1}\|^2\right] \leq 2\alpha_k^2 T^2\left(\sigma_g^2\left(1 + 2L_g^2\mu_g^{-1}\alpha_k\right) + L_g^2 E_k^y\right). \tag{31}$$

On the other hand, using a triangular inequality and applying Proposition 7, we also have that:

$$\mathbb{E}\left[\|y_k - y_{k-1}\|^2\right] \leq 2\mathbb{E}\left[\|y_k - y^\star(x_k)\|^2\right] + 2\mathbb{E}\left[\|y_{k-1} - y^\star(x_k)\|^2\right] \leq (4E_k^y + 2R_k^y). \tag{32}$$

The result follows by combining (31) and (32) using coefficients $1-v$ and $v$. □

### C.3 PROOF OF PROPOSITION 5

*Proof of Proposition 5 .* We will control each of $E_k^y$ and $E_k^z$ separately.

**Upper-bound on $E_k^y$.** Let $r_k$ be a non-increasing sequences between 0 and 1. The following holds:

$$
\begin{aligned}
E_k^y =& \mathbb{E}\Big[\|y_{k-1} - y^\star(x_{k-1})\|^2\Big] + \mathbb{E}\Big[\|y^\star(x_k - y^\star(x_{k-1})\|^2\Big] \\
& + 2\mathbb{E}\Big[(y_{k-1} - y^\star(x_{k-1}))^\top (y^\star(x_k - y^\star(x_{k-1})))\Big] \\
\overset{(i)}{\leq}& (1 + r_k)\mathbb{E}\Big[\|y_{k-1} - y^\star(x_{k-1})\|^2\Big] + \big(1 + r_k^{-1}\big)\mathbb{E}\Big[\|y^\star(x_k - y^\star(x_{k-1})\|^2\Big] \\
\overset{(ii)}{\leq}& (1 + r_k)\big(\Lambda_{k-1} E_{k-1}^y + R_{k-1}^y\big) + 2r_k^{-1}\mathbb{E}\Big[\|y^\star(x_k) - y^\star(x_{k-1})\|^2\Big] \\
\overset{(iii)}{\leq}& (1 + r_k)\big(\Lambda_{k-1} E_{k-1}^y + R_{k-1}^y\big) + 2L_y^2 r_k^{-1}\mathbb{E}\Big[\|x_k - x_{k-1}\|^2\Big] \\
\overset{(iv)}{\leq}& (1 + r_k)\big(\Lambda_{k-1} E_{k-1}^y + R_{k-1}^y\big) + 2L_y^2 r_k^{-1}\gamma_k^2 \mathbb{E}\Big[\big\|\hat{\psi}_{k-1}\big\|^2\Big]
\end{aligned}
\tag{33a}
$$

(i) follows by Young's inequality, (ii) uses Proposition 7 to bound the first term and that $(1 + r_k^{-1}) \leq 2r_k^{-1}$ for the second term, (iii) uses that $y^\star$ is $L_y$-Lipschitz by Proposition 6 and (iv) uses the update equation $x_k = x_{k-1} - \gamma_k \hat{\psi}_{k-1}$.

**Upper-bound on $E_k^z$.** Similarly, for a non-increasing sequence $0 < \theta_k \leq 1$, we have that:

$$
\begin{aligned}
E_k^z =& \mathbb{E}\Big[\|z_{k-1} - z^\star(x_{k-1}, y_{k-1})\|^2\Big] + \mathbb{E}\Big[\big\|z^\star(x_k, y_k) - z^\star(x_{k-1}, y_{k-1})^2\big\|\Big] \\
& + 2\mathbb{E}\Big[(z_{k-1} - z^\star(x_{k-1}, y_{k-1}))^\top (z^\star(x_k, y_k) - z^\star(x_{k-1}, y_{k-1}))\Big] \\
\overset{(i)}{\leq}& (1 + \theta_k)\mathbb{E}\Big[\|z_{k-1} - z^\star(x_{k-1}, y_{k-1})\|^2\Big] + (1 + \theta_k^{-1})\mathbb{E}\Big[\|z^\star(x_k, y_k) - z^\star(x_{k-1}, y_{k-1})\|^2\Big] \\
\overset{(ii)}{\leq}& (1 + \theta_k)\big(\Pi_{k-1} E_{k-1}^z + R_{k-1}^z\big) + 2\theta_k^{-1}\mathbb{E}\Big[\|z^\star(x_k, y_k) - z^\star(x_{k-1}, y_{k-1})\|^2\Big] \\
\overset{(iii)}{\leq}& (1 + \theta_k)\big(\Pi_{k-1} E_{k-1}^z + R_{k-1}^z\big) + 4L_z^2\theta_k^{-1}\Big(\mathbb{E}\Big[\|x_k - x_{k-1}\|^2 + \|y_k - y_{k-1}\|^2\Big]\Big) \\
\overset{(iv)}{\leq}& (1 + \theta_k)\big(\Pi_{k-1} E_{k-1}^z + R_{k-1}^z\big) + 4L_z^2\theta_k^{-1}\Big(\gamma_k^2\mathbb{E}\Big[\big\|\hat{\psi}_{k-1}\big\|^2\Big] + 2\phi_k E_k^y + 2\tilde{R}_k^y\Big)
\end{aligned}
\tag{34a}
$$

(i) follows by Young's inequality, (ii) uses Proposition 7 to bound the first term and that $(1 + \theta_k^{-1}) \leq 2\theta_k^{-1}$ for the second term, (iii) uses that $z^\star(x, y)$ is $L_z$-Lipschitz in $x$ and $y$ by Proposition 6 and, finally, (iv) uses the update equation $x_k = x_{k-1} - \gamma_k \hat{\psi}_{k-1}$ for the term $\mathbb{E}\Big[\|x_k - x_{k-1}\|^2\Big]$ and Proposition 12 to control the increments $\mathbb{E}\Big[\|y_k - y_{k-1}\|^2\Big]$.

In order to express the upper-bound on $E_k^z$ in terms of $E_{k-1}^y$ instead of $E_k^y$, we substitute $E_k^y$ in (34a) by its upper-bound in (33a) and use that $(1 + r_k) \leq 2$ to write:

$$
\begin{aligned}
E_k^z \leq& (1 + \theta_k)\big(\Pi_{k-1} E_{k-1}^z + R_{k-1}^z\big) + 4L_z^2\theta_k^{-1}\gamma_k^2\big(1 + 4L_y^2\phi_k r_k^{-1}\big)\mathbb{E}\Big[\big\|\hat{\psi}_{k-1}\big\|^2\Big] \\
& + 8L_z^2\theta_k^{-1}\Big(2\phi_k\big(\Lambda_{k-1} E_{k-1}^y + R_{k-1}^y\big) + \tilde{R}_k^y\Big)
\end{aligned}
\tag{35}
$$

We can then express (33a) and (35) jointly in matrix form as follows:

$$
\begin{pmatrix} E_k^y \\ E_k^z \end{pmatrix} \leq \boldsymbol{P_k} \begin{pmatrix} \Lambda_{k-1} E_{k-1}^y + R_{k-1}^y \\ \Pi_{k-1} E_{k-1}^z + R_{k-1}^z \end{pmatrix} + \gamma_k \mathbb{E}\Big[\big\|\hat{\psi}_{k-1}\big\|^2\Big]\boldsymbol{U_k} + \boldsymbol{V_k}
$$

where the $\boldsymbol{P_k}$ is a $2 \times 2$ matrix and $\boldsymbol{U_k}$ and $\boldsymbol{V_k}$ are 2-dimensional vectors given by (17). The desired result follows directly by substituting $\mathbb{E}\Big[\big\|\hat{\psi}_{k-1}\big\|^2\Big]$ by its upper-bound from Proposition 12 in the above inequality. $\qquad\square$

## C.4 GENERAL ERROR BOUND

*Proof of Proposition 9.* First note that, by assumption, we have that $\delta_k r_k^{-1} \leq \delta_{k-1} r_{k-1}^{-1}$ and $\delta_k \theta_k^{-1} \leq \delta_{k-1} \theta_{k-1}^{-1}$. Moreover, since $\alpha_k$ and $\beta_k$ are non-increasing, we also have that $\Lambda_{k-1} \leq \Lambda_k$ and $\Pi_{k-1} \leq \Pi_k$. This implies the following inequalities which will be used in the rest of the proof:

$$a_{k-1}^{-1} \Lambda_{k-1} \leq a_k^{-1} \Lambda_k, \qquad b_{k-1}^{-1} \Pi_{k-1} \leq b_k^{-1} \Pi_k. \tag{36}$$

Now, let $\rho_k$ be a non-increasing sequence with $0 < \rho_k < 1$. By Proposition 2, it follows that $E_k^x$ satisfies the inequality:

$$F_k + E_k^x \leq \left(1 - \left(1 - \frac{1}{2}\rho_k\right)\delta_k\right)E_{k-1}^x + \gamma_k s_k V_{k-1}^\psi + \gamma_k\left(s_k + \rho_k^{-1}\right)E_{k-1}^\psi \tag{37}$$

On the other hand, by Proposition 5 we know that $E_k^y$ and $E_k^z$ satisfy:

$$\begin{pmatrix} E_k^y \\ E_k^z \end{pmatrix} \leq \boldsymbol{P_k} \begin{pmatrix} \Lambda_{k-1} E_{k-1}^y + R_{k-1}^y \\ \Pi_{k-1} E_{k-1}^z + R_{k-1}^z \end{pmatrix} + \gamma_k \left(V_{k-1}^\psi + 2E_{k-1}^\psi + 2\zeta_k E_{k-1}^x\right)\boldsymbol{U_k} + \boldsymbol{V_k} \tag{38}$$

where the $\boldsymbol{P_k}$, $\boldsymbol{U_k}$ and $\boldsymbol{V_k}$ are defined in (17). For conciseness, define $\boldsymbol{S_k}$ and $\boldsymbol{E_k^I}$ to be:

$$\boldsymbol{S_k} := \begin{pmatrix} a_k & 0 \\ 0 & b_k \end{pmatrix}, \qquad \boldsymbol{E_k^I} = \boldsymbol{S_k} \begin{pmatrix} E_k^y \\ E_k^z \end{pmatrix}$$

By (36), we directly have that:

$$\boldsymbol{S_k P_k S_{k-1}^{-1}} \begin{pmatrix} \Lambda_{k-1} & 0 \\ 0 & \Pi_{k-1} \end{pmatrix} \leq \boldsymbol{S_k P_k S_k^{-1}} \begin{pmatrix} \Lambda_k & 0 \\ 0 & \Pi_k \end{pmatrix} := \tilde{\boldsymbol{P}}_{\boldsymbol{k}}, \tag{39}$$

where the inequality in (39) holds component-wise. Therefore, multiplying (38) by $\boldsymbol{S_k}$ and using (39) yields:

$$\boldsymbol{E_k^I} \leq \tilde{\boldsymbol{P}}_{\boldsymbol{k}} \boldsymbol{E_{k-1}^I} + \boldsymbol{S_k}\left(\boldsymbol{P_k} \begin{pmatrix} R_{k-1}^y \\ R_{k-1}^z \end{pmatrix} + \gamma_k\left(V_{k-1}^\psi + 2E_{k-1}^\psi + 2\zeta_k E_{k-1}^x\right)\boldsymbol{U_k} + \boldsymbol{V_k}\right) \tag{40}$$

Furthermore, by Proposition 4 we can bound $E_{k-1}^\psi$ and $V_{k-1}^\psi$ as follows:

$$E_{k-1}^\psi \leq 2L_\psi^2\left(\Lambda_k(a_k)^{-1}a_{k-1}E_{k-1}^y + \Pi_k(b_k)^{-1}b_{k-1}E_{k-1}^z\right) + 2L_\psi^2 R_{k-1}^y$$
$$V_{k-1}^\psi \leq w_x^2 + \sigma_x^2 \Pi_k(b_k)^{-1}b_{k-1}E_{k-1}^z,$$

where we used (36) a second time to replace $\Lambda_{k-1}(a_{k-1})^{-1}$ and $\Pi_{k-1}(b_{k-1})^{-1}$ by $\Lambda_k(a_k)^{-1}$ and $\Pi_k(b_k)^{-1}$. By summing both inequalities (37) and (40) and substituting all terms $E_{k-1}^\psi$ and $V_{k-1}^\psi$ by their upper-bounds we obtain an inequality of the form:

$$F_k + E_k^{tot} \leq A_k^x E_{k-1}^x + A_k^y a_{k-1} E_{k-1}^y + A_k^z b_{k-1} E_{k-1}^z + V_k^{tot}$$

where $A_k^x$, $A_k^y$, $A_k^z$ are the components of the vector $A_k$ defined in (18) and $V_k^{tot}$ is the variance term also defined in (18). The desired inequality follows by upper-bounding $A_k^x$, $A_k^y$, $A_k^z$ by their maximum value $\|A_k\|_\infty$. $\qquad\square$

## D CONTROLLING THE PRECISION OF THE INNER-LEVEL ALGORITHMS.

In this section, we prove Proposition 10. To achieve this, we first provide general conditions on $\Lambda_k$ and $\Pi_k$ for controlling the rate $\|A_k\|_\infty$ and which hold regardless of the choice of step-sizes. This is achieved in Proposition 13 of Appendix D.1. Then we prove Proposition 10 in Appendix D.2.

### D.1 CONTROLLING $\Pi_k$ AND $\Lambda_k$.

We introduce the following quantities:

$$D_k^{(1)} := \frac{1}{1-s} \left[ \log \left[ \frac{1 - (1 - \rho_k)\delta_k}{1 + 2r_k \left[ 1 + 2L_\psi^2 \gamma_k \delta_k^{-1} \left[ s_k + 1 + [2\rho_k]^{-1} \right] \right]} \right] \right] \tag{41a}$$

$$D_k^{(2)} := \frac{1}{s} \log \left( (16L_y^2)^{-1} \rho_k \zeta_k^{-1} \gamma_k^{-2} r_k^2 \right) \tag{41b}$$

$$D_k^{(3)} := -\frac{1}{s} \log \left( 4L_y^2 \delta_k \gamma_k r_k^{-2} \right) \tag{41c}$$

$$D_k^{(4)} := \frac{1}{1-s} \log \left[ \frac{1 - (1 - \rho_k)\delta_k}{1 + \theta_k \left[ 1 + 2\gamma_k \delta_k^{-1} \left( 2L_\psi^2 + \sigma_x^2 \right) \left[ s_k + 1 + (2\rho_k)^{-1} \right] \right]} \right] \tag{41d}$$

$$D_k^{(5)} := -\frac{1}{s} \log \left( 16L_z^2 \theta_k^{-2} \phi_k \right) \tag{41e}$$

$$D_k^{(6)} := -\frac{1}{s} \log \left( 2L_z^2 L_y^{-2} \theta_k^{-2} r_k^2 \left( 1 + 4L_y^2 r_k^{-1} \phi_k \right) \right) \tag{41f}$$

**Proposition 13.** *Let $\rho_k$ be a non-increasing sequence of positive numbers smaller than* 1. *Consider* $\Lambda_k$ *and* $\Pi_k$ *so that:*

$$\log \Lambda_k \le \min \left( D_k^{(1)}, D_k^{(2)}, D_k^{(3)} \right), \tag{42a}$$

$$\log \Pi_k \le \min \left( D_k^{(1)}, D_k^{(2)}, \log \Lambda_k + D_k^{(3)} \right). \tag{42b}$$

*Then, the following inequalities holds:*

$$\|A_k\|_\infty \le (1 - (1 - \rho_k)\delta_k), \qquad V_k^{tot} \le \tilde{V}_k^{tot}.$$

*where $A_k$ and $V_k^{tot}$ are defined in ([18](#)) of Proposition [9](#) and $\tilde{V}_k^{tot}$ is defined as:*

$$\tilde{V}_k^{tot} := \delta_k \left( 3r_k^{-1} + 2L_\psi^2 \gamma_k \delta_k^{-1} \left( 2 + (s_k + \rho_k^{-1}) \right) \right) R_{k-1}^y$$
$$+ \delta_k \Pi_k^s \left( 2\theta_k^{-1} R_{k-1}^z + 8L_z^2 \theta_k^{-2} \tilde{R}_k^y \right) + \gamma_k \left( s_k + 4L_y^2 \Lambda_k^s \delta_k \gamma_k r_k^{-2} \right) w_x^2$$

*Proof.* We first prove that $u_k^I \le u_k^+ \le 1$ and $p_k^+ \le 1$ under ([42a](#)) and ([42b](#)) with $u_k^+$, $p_k^+$ given by:

$$u_k^+ := 4L_y^2 \delta_k \gamma_k r_k^{-2} \Lambda_k^s, \qquad p_k^+ = 16L_z^2 \Pi_k^s \theta_k^{-2} \phi_k.$$

A direct calculation shows $u_k^+ \le 1$ whenever ([42a](#)) holds. Moreover, recall that $u_k^I = a_k U_k^{(1)} + b_k U_k^{(2)}$ with $U_k^{(1)}$ and $U_k^{(2)}$ being the components of the vector $\boldsymbol{U_k}$ defined in ([17](#)). Thus by direct substitution, we get the following expression for $u_k^I$:

$$u_k^I = 2L_y^2 \delta_k \gamma_k r_k^{-2} \Lambda_k^s \left( 1 + 2L_z^2 L_y^{-2} \theta_k^{-2} r_k^2 \left( 1 + 4L_y^2 r_k^{-1} \phi_k \right) \frac{\Pi_k^s}{\Lambda_k^s} \right).$$

Therefore, ([42b](#)) suffices to ensure that $u_k^I \le u_k^+$. Finally, ([42b](#)) implies directly that $p_k^+ \le 1$.

We will control each component $A_k^x$, $A_k^y$ and $A_k^z$ of the vector $A_k$ separately.

**Controlling $A_k^x$.** Recalling the expression of $A_k^x$, the first component of $A_k$ in ([18](#)), it holds that:

$$A_k^x = 1 - \left( 1 - \frac{1}{2}\rho_k \right)\delta_k + 2\zeta_k \lambda_k u_k^I \overset{(i)}{\le} 1 - \left( 1 - \frac{1}{2}\rho_k \right)\delta_k + 2\zeta_k \gamma_k u_k^+$$

$$\overset{(ii)}{\le} 1 - \left( 1 - \frac{1}{2}\rho_k \right)\delta_k + \frac{1}{2}\rho_k\delta_k = 1 - (1 - \rho_k)\delta_k.$$

(i) holds since $u_k^I \le u_k^+$ while (ii) follows from ([42a](#)) which ensures that $2\zeta_k \gamma_k u_k^+ \le \frac{1}{2}\rho_k\delta_k$.

**Controlling $A_k^y$.** Recall the expression of second component of $A_k^y$ in (18), we have:

$$A_k^y = \Lambda_k^{1-s}\left(1 + r_k\left(1 + 16L_z^2\Pi_k^s\theta_k^{-2}\phi_k + 2L_\psi^2\gamma_k\delta_k^{-1}\left(2u_k^I + (s_k + \rho_k^{-1})\right)\right)\right)$$

$$\overset{(i)}{\leq}\left(1 + 2r_k\left(1 + 2L_\psi^2\gamma_k\delta_k^{-1}\left(s_k + 1 + (2\rho_k)^{-1}\right)\right)\right)\Lambda_k^{1-s} \overset{(ii)}{\leq} 1 - (1 - \rho_k)\delta_k.$$

(i) holds since $p_k^+ := 16L_z^2\Pi_k^s\theta_k^{-2}\phi_k \leq 1$ and $u_k^I \leq 1$ while (ii) is a consequence of (42a).

**Controlling $A_k^z$.** Similarly, recalling the expression of the third component of $A_k$ we get that:

$$A_k^z = \Pi_k^{1-s}\left(1 + \theta_k\left(1 + \gamma_k\delta_k^{-1}\left(2L_\psi^2 + \sigma_x^2\right)\left(2u_k^I + (s_k + \rho_k^{-1})\right)\right)\right)$$

$$\overset{(i)}{\leq}\Pi_k^{1-s}\left(1 + \theta_k\left(1 + 2\gamma_k\delta_k^{-1}\left(2L_\psi^2 + \sigma_x^2\right)\left(1 + (s_k + (2\rho_k)^{-1})\right)\right)\right) \overset{(ii)}{\leq} 1 - (1 - \rho_k)\delta_k,$$

where (i) uses that $u_k^I \leq 1$ and (ii) follows from (42b).

**Controlling $V_k^{tot}$.** Recalling the expression of $V_k^{tot}$ from (18), we have that:

$$V_k^{tot} := \delta_k\left(\Lambda_k^s(1 + r_k^{-1}) + 16L_z^2\phi_k\theta_k^{-2}\Pi_k^s + 2L_\psi^2\gamma_k\delta_k^{-1}\left(2u_k^I + (s_k + \rho_k^{-1})\right)\right)R_{k-1}^y$$

$$+ \delta_k\left(\left(1 + \theta_k^{-1}\right)\Pi_k^s R_{k-1}^z + 8L_z^2\theta_k^{-2}\Pi_k^s\tilde{R}_k^y\right) + \gamma_k\left(s_k + u_k^I\right)w_x^2$$

$$\leq \delta_k\left(3r_k^{-1} + 2L_\psi^2\gamma_k\delta_k^{-1}\left(2 + (s_k + \rho_k^{-1})\right)\right)R_{k-1}^y$$

$$+ \delta_k\left(2\theta_k^{-1}\Pi_k^s R_{k-1}^z + 8L_z^2\theta_k^{-2}\Pi_k^s\tilde{R}_k^y\right) + \gamma_k\left(s_k + u_k^+\right)w_x^2 = \tilde{V}_k^{tot}.$$

where we use $16L_z^2\phi_k\theta_k^{-2}\Pi_k^s \leq 1$ and $u_k^I \leq 1$ for the first line and $u_k^I \leq u_k^+$ for the last line.  □

### D.2 CONTROLLING THE NUMBER OF INNER-LEVEL ITERATIONS

We provide now a proof of Proposition 10 which is a consequence of Proposition 13.

*Proof of Proposition 10.* We first provide conditions on the number of iterations $T$ and $N$ of algorithms $\mathcal{A}_k$ and $\mathcal{B}_k$ to control the rate $\|A_k\|_\infty$ and then provide an upper-bound on $V_k^{tot}$.

**Conditions on $T$ and $N$.** We consider the setting with constant step-size $\gamma_k = \gamma$, $\alpha_k = \alpha$ and $\beta_k = \beta$ and choose $r_k = \theta_k = 1$ and $\delta_k = \delta_0$ for some $0 < \delta_0 < 1$. We also take $v = 1$ so that $\phi_k = 2$ and $\tilde{R}_k^y = R_k^y$. By direct substitution of the parameters $r_k$, $\theta_k$, $\phi_k$, $\gamma_k$, $\delta_k$, $\rho_k$ and $\zeta_k$, in the expressions of $D_k^{(1)}$, $D_k^{(2)}$, $D_k^{(3)}$, $D_k^{(4)}$, $D_k^{(5)}$ and $D_k^{(6)}$ defined in (41a) to (41f), we verify that:

$$-C_1 \leq D_k^{(1)}, \qquad -C_1' \leq D_k^{(4)}, \qquad -C_2 \leq \min\left(D_k^{(2)}, D_k^{(3)}\right), \qquad -C_2' \leq \min\left(D_k^{(5)}, D_k^{(6)}\right).$$

Hence, we can ensure the conditions of Proposition 13 hold by choosing $T$ and $N$ so that:

$$\log\Lambda_k \leq -\max\left(1, C_1, C_2\right), \qquad \log\Pi_k \leq -\log\Lambda_k - \max\left(1, C_1', C_2'\right).$$

This is achieved by for the following choice:

$$T = \lfloor\alpha^{-1}\mu_g^{-1}\max\left(C_1, C_2, C_3\right)\rfloor + 1, \tag{43}$$

$$N = \lfloor 2\beta^{-1}\mu_g^{-1}\left(\max\left(C_1, C_2, C_3\right) + \max\left(C_1', C_2', C_3'\right)\right)\rfloor + 1$$

Hence, for such choice, we are guaranteed by Proposition 13 that $\|A_k\|_\infty \leq 1 - (1 - \rho_k)\delta_k$.

**Bound on the variance $V_k^{tot}$.** By choosing $T$ and $N$ as in (43), we know that $\Lambda_k$ and $\Pi_k$ satisfy (42), so that the variance term $V_k^{tot}$ is upper-bounded by $\tilde{V}_k^{tot}$. Moreover, by direct substitution of the sequences appearing in the expression of $\tilde{V}_k^{tot}$ by their values, we get:

$$V_k^{tot} \leq \tilde{V}_k^{tot} = \delta_k\left(\Lambda_k^s(1 + r_k^{-1}) + 16L_z^2\phi_k\theta_k^{-2}\Pi_k^s + 2L_\psi^2\eta_k^{-1}\left(2u_k^I + (s_k + \rho_k^{-1})\right)\right)R_{k-1}^y$$

$$+ \delta_k\left(\left(1 + \theta_k^{-1}\right)\Pi_k^s R_{k-1}^z + 8L_z^2\theta_k^{-2}\Pi_k^s\tilde{R}_k^y\right) + \gamma_k\left(s_k + u_k^I\right)w_x^2$$

$$\leq \delta_0\left(\eta_0^{-1}\frac{1-u}{2} + \left(\frac{1+u}{2}\gamma + 4L_y^2\Lambda_k^s\gamma^2\right)\right)w_x^2$$

$$+ \delta_0\left(2\Lambda_k^s + 32L_z^2\Pi_k^s + 10L_\psi^2\eta_0^{-1}\right)R_{k-1}^y + \delta_0\left(2\Pi_k^s R_{k-1}^z + 8L_z^2\Pi_k^s\tilde{R}_k^y\right)$$

Furthermore, by definition of $R_{k-1}^y$ and $R_{k-1}^z$, we have that:

$$R_{k-1}^y \le 2\mu_g^{-1} L_g^{-1} \sigma_g^2, \qquad R_{k-1}^z \le \mu_g^{-3}\Big(B^2 L_g^{-1}\sigma_{g_{yy}}^2 + 3\mu_g \sigma_f^2\Big).$$

Moreover, recall that $\tilde{R}_k^y = R_k^y$ since we chose $v=1$. Thus $\tilde{R}_k^y \le 2\mu_g^{-1}\alpha\sigma_g^2$. This implies that:

$$
\begin{aligned}
V_k^{tot}\delta_0^{-1}\gamma^{-1} \le & \left(\delta_0^{-1}\frac{1-u}{2} + \big(1+4L_y^2\Lambda_k^s\gamma\big)\right)w_x^2 + 2\Pi_k^s\gamma^{-1}\mu_g^{-3}\Big(B^2\beta\sigma_{g_{yy}}^2 + 3\mu_g\sigma_f^2\Big) \quad (44)\\
& + \big(4\Lambda_k^s + 80L_z^2\Pi_k^s + 20L_\psi^2\eta_0^{-1}\big)\mu_g^{-1}\alpha\gamma^{-1}\sigma_g^2
\end{aligned}
$$

By choosing $T$ and $N$ as in (43), the following conditions hold:

$$\Lambda_k^s \le \frac{L}{4L_y^2}, \qquad \Lambda_k^s \le 5L_\psi^2\eta_0^{-1}, \qquad \Pi_k^s \le \gamma\Big(\sigma_{g_{xy}}^2 + (L_g')^2\Big), \qquad \Pi_k^s \le \frac{1}{4L_z^2}L_\psi^2\eta_0^{-1}.$$

By applying these inequalities in (44), we get:

$$
\begin{aligned}
V_k^{tot}\delta_0^{-1}\gamma^{-1} \le & \left(\delta_0^{-1}\frac{1-u}{2} + 2\right)w_x^2 + 2\mu_g^{-3}\Big(\sigma_{g_{xy}}^2 + (L_g')^2\Big)\Big(B^2\beta\sigma_{g_{yy}}^2 + 3\mu_g\sigma_f^2\Big)\\
& + 60L_\psi^2\eta_0^{-1}\mu_g^{-1}L_g^{-1}\gamma^{-1}\sigma_g^2\\
\le & \left(\delta_0^{-1}\frac{1-u}{2} + 3\right)w_x^2 + \frac{60L_\psi^2}{\delta_0\mu_g L_g}\sigma_g^2 = \mathcal{W}^2
\end{aligned}
$$

where we used that $2\mu_g^{-3}\Big(\sigma_{g_{xy}}^2 + (L_g')^2\Big)\Big(B^2\beta\sigma_{g_{yy}}^2 + 3\mu_g\sigma_f^2\Big) \le w_x^2$ by definition of $w_x^2$ in (16a). Therefore, we have shown that $V_k^{tot} \le \gamma\delta_0\mathcal{W}^2$, with $\mathcal{W}^2$ given by (20). $\qquad\square$

# E    STOCHASTIC LINEAR DYNAMICAL SYSTEM WITH CORRELATED NOISE

Let $A$ be a positive definite matrix in $\mathbb{R}^d \times \mathbb{R}^d$ satisfying $0 < \mu_g \ge |\sigma_i(A)| \le L_g$ and $b$ a vector in $\mathbb{R}^d$. We denote by $z^\star = -A^{-1}b$. Consider $A_m$ be a sequence of i.i.d. positive symmetric matrices in $\mathbb{R}^d \times \mathbb{R}^d$ such that $\mathbb{E}[A_m] = A$, and $\hat{b}$ a random vector in $\mathbb{R}^d$ such that $\mathbb{E}\big[\hat{b}\big] = b$, with $A_m$ and $\hat{b}$ being mutually independent. Define $\Sigma_A = \mathbb{E}\big[(A_n - A)^\top(A_n - A)\big]$ and denote by $\sigma_A$ and $L_A$ the largest singular values of $\Sigma_A$ and $A^{-1}\Sigma_A A^{-1}$. Let $\beta$ be such that $\beta \le \frac{1}{L_g}$. Finally let $\sigma_c^2$ be an upper-bound on $\mathbb{E}\Big[\big\|\hat{b} - b\big\|^2\Big]$. Let $z$ and $z'$ be two vectors in $\mathbb{R}^d$ and define the iterates $z^n$ and $\bar{z}^n$ such that $z^0 = z$ and $\bar{z}^0 = z'$ and using the recursion:

$$z^n = (I - \beta A_n)z^{n-1} - \beta\hat{b}, \bar{z}^n = (I - \beta A)\bar{z}^{n-1} - \beta b.$$

Hence, from the definition of $z^n$ and $\bar{z}^n$ we directly have that:

$$z^n = \prod_{t=1}^n (I - \beta A_n)z - \sum_{t=1}^n \prod_{j=t+1}^n \beta(I - \beta A_j)\hat{b}, \quad \bar{z}^n = \prod_{t=1}^n(I - \beta A)z' - \sum_{t=1}^n \prod_{j=t+1}^n \beta(I - \beta A)b.$$

The next proposition computes the bias $\mathbb{E}\Big[z^n - \bar{z}^n\big|\hat{b}\Big]$.

**Proposition 14.** *The following identities hold:*

$$
\begin{aligned}
\mathbb{E}\Big[z^n - \bar{z}^n\big|\hat{b}\Big] =& (I - \beta A)^n(z - z') - \beta\left(\sum_{t=1}^n(I - \beta A)^{n-t}\right)\Big(\hat{b} - b\Big)\\
\mathbb{E}[z^n - \bar{z}^n] =& (I - \beta A)^n(z - z')
\end{aligned}
$$

*Proof.* The proof is a consequence of $A_n$ and $\hat{b}$ being i.i.d. and unbiased estimates of $A$ and $b$. $\qquad\square$

The next proposition controls the mean squared errors $\mathbb{E}\left[\|z^n - z^\star\|^2\right]$ and $\mathbb{E}\left[\|z^n - \bar{z}^n\|^2\right]$.

**Proposition 15.** *Define* $n^\star(n, \beta) = \min(n, \frac{1}{\beta \mu_g})$. *Let* $\beta$ *such that:*

$$\beta \leq \frac{1}{2L_g} \min\left(1, \frac{2L_g}{\mu_g\left(1 + \mu_g^{-2}\sigma_A^2\right)}\right)$$

*Then, the following inequalities holds:*

$$\mathbb{E}\left[\|z^n - z^\star\|^2\right] \leq (1 - \beta\mu_g)^n \|z - z^\star\|^2 + \beta^2\left(\sigma_A^2\|z^\star\|^2 + 3\left(n \wedge \frac{1}{\beta\mu_g}\right)\sigma_c^2\right)\left(n \wedge \frac{1}{\beta\mu_g}\right),$$

$$\|\bar{z}^n - z^\star\|^2 \leq (1 - \beta\mu_g)^n \|z' - z^\star\|^2.$$

*Moreover, if* $z' = z$, *then we have:*

$$\mathbb{E}\left[\|z^n - \bar{z}^n\|^2\right] \leq 4\mu_g^{-2}\sigma_A^2(1 - \frac{\beta\mu_g}{2})^n\|z - z^\star\|^2 \tag{45}$$
$$+ 2\beta^2\left(\sigma_A^2\|z^\star\|^2 + 3\left(n \wedge \frac{1}{\beta\mu_g}\right)\sigma_c^2\right)\left(n \wedge \frac{1}{\beta\mu_g}\right).$$

*Proof.* It is straightforward to see that:

$$\|\bar{z}^n - z^\star\|^2 \leq (1 - \beta\mu_g)\|z' - z^\star\|^2.$$

Now, let's control $\mathbb{E}\left[\|z^n - \bar{z}^n\|^2\right]$. The following identity holds by definition of $z^n$ and $\bar{z}^n$:

$$\mathbb{E}\left[\|z^n - \bar{z}^n\|^2\right] = \mathbb{E}\left[\left(z^{n-1} - \bar{z}^{n-1}\right)^\top\left((I - \beta A)^2 + \beta^2\Sigma_A\right)\left(z^{n-1} - \bar{z}^{n-1}\right)\right]$$
$$- 2\beta\mathbb{E}\left[\left(z^{n-1} - \bar{z}^{n-1}\right)^\top(I - \beta A)(\hat{b} - b)\right]$$
$$+ \beta^2\left(\mathbb{E}\left[\left\|\hat{b} - b\right\|^2\right] + \mathbb{E}\left[\left(\bar{z}^{n-1}\right)^\top\Sigma_A\bar{z}^{n-1}\right]\right)$$
$$= \mathbb{E}\left[\left(z^{n-1} - \bar{z}^{n-1}\right)^\top\left((I - \beta A)^2 + \beta^2\Sigma_A\right)\left(z^{n-1} - \bar{z}^{n-1}\right)\right]$$
$$+ \beta^2\left(\left(\bar{z}^{n-1}\right)^\top\Sigma_A\bar{z}^{n-1} + \mathbb{E}\left[\left(\hat{b} - b\right)^\top(I + 2(I - \beta A)D_n)\left(\hat{b} - b\right)\right]\right)$$

where $\Sigma_A = \mathbb{E}\left[(A_n - A)^\top(A_n - A)\right]$ and $D_n = \sum_{t=0}^{n-1}(I - \beta A)^t$. By simple calculation we can upper-bound the last term by:

$$\beta^2\mathbb{E}\left[\left(\hat{b} - b\right)^\top(I + 2(I - \beta A)D_n)\left(\hat{b} - b\right)\right] \leq 3\left(n \wedge \frac{1}{\beta\mu_g}\right)\beta^2\mathbb{E}\left[\left\|\hat{b} - b\right\|^2\right].$$

Moreover, provided that $\beta \leq \frac{1}{L_g(1 + L_A)}$, where $L_A$ is the highest eigenvalue of $A^{-1}\Sigma_A A^{-1}$, then we have the following:

$$\mathbb{E}\left[\|z^n - \bar{z}^n\|^2\right] \leq (1 - \beta\mu_g)\mathbb{E}\left[\|z^{n-1} - \bar{z}^{n-1}\|^2\right]$$
$$+ \beta^2\left(\left(\bar{z}^{n-1}\right)^\top\Sigma_A\bar{z}^{n-1} + 3\left(n \wedge \frac{1}{\beta\mu_g}\right)\mathbb{E}\left[\left\|\hat{b} - b\right\|^2\right]\right)$$
$$\leq (1 - \beta\mu_g)\mathbb{E}\left[\|z^{n-1} - \bar{z}^{n-1}\|^2\right] + \beta^2\left(\sigma_A^2\|\bar{z}^{n-1}\|^2 + 3\left(n \wedge \frac{1}{\beta\mu_g}\right)\sigma_c^2\right),$$

Unrolling the recursion, it follows that:

$$\mathbb{E}\left[\|z^n - \bar{z}^n\|^2\right] \leq (1 - \beta\mu_g)^n\|z - z'\|^2 + \beta^2\sum_{t=1}^{n}(1 - \beta\mu_g)^{n-t}\left(\sigma_A^2\|\bar{z}^{n-1}\|^2 + 3\left(n \wedge \frac{1}{\beta\mu_g}\right)\sigma_c^2\right) \tag{46}$$

In particular, if $z' = z^\star$, then $\bar{z}^n = z^\star$ and we get:

$$\mathbb{E}\left[\|z^n - z^\star\|^2\right] \leq (1 - \beta\mu_g)^n \|z - z^\star\|^2 + \beta^2 \left(\sigma_A^2 \|z^\star\|^2 + 3\left(n \wedge \frac{1}{\beta\mu_g}\right)\sigma_c^2\right)\left(n \wedge \frac{1}{\beta\mu_g}\right).$$

To get the last inequality, we simply choose $z' = z$ and recall that:

$$\left\|\bar{z}^{t-1}\right\| \leq 2\left((1 - \beta\mu_g)^{t-1}\|z' - z^\star\| + \|z^\star\|\right).$$

Using the above in in (46) yields:

$$\mathbb{E}\left[\|z^n - \bar{z}^n\|^2\right] \leq 2n\beta^2\sigma_A^2(1 - \beta\mu_g)^{n-1}\|z - z^\star\|^2 + 2\beta^2\left(\sigma_A^2\|z^\star\|^2 + 3\left(n \wedge \frac{1}{\beta\mu_g}\right)\sigma_c^2\right)\left(n \wedge \frac{1}{\beta\mu_g}\right).$$

Moreover, by Lemma 3 we know that $n\beta^2\mu_g^2(1 - \beta\mu_g)^{n-1} \leq (1 - \frac{\beta\mu_g}{2})^{n-1}$ and since $\beta\mu_g \leq 1$, we have that $(1 - \frac{\beta\mu_g}{2})^{-1} \leq 2$ so that $(1 - \frac{\beta\mu_g}{2})^{n-1} \leq 2(1 - \frac{\beta\mu_g}{2})^n$. Hence, we can write:

$$\mathbb{E}\left[\|z^n - \bar{z}^n\|^2\right] \leq 4\mu_g^{-2}\sigma_A^2(1 - \frac{\beta\mu_g}{2})^n \|z - z^\star\|^2 + 2\beta^2\left(\sigma_A^2\|z^\star\|^2 + 3\left(n \wedge \frac{1}{\beta\mu_g}\right)\sigma_c^2\right)\left(n \wedge \frac{1}{\beta\mu_g}\right).$$

$\square$

**Lemma 2.** *Let $A$ and $\Sigma_A$ be symmetric positive matrix in $\mathbb{R}^d \times \mathbb{R}^d$ with $\sigma_A^2$ its largest singular value of $\Sigma_A$ and $0 < \mu_g \leq \sigma_i(A) \leq L_g$. Let $\beta$ be a positive number such that:*

$$\beta \leq \frac{1}{2L_g}\min\left(1, \frac{2L_g}{\mu_g(1 + \mu_g^{-2}\sigma_A^2)}\right)$$

*Then the following holds:*

$$\left\|(I - \beta A)^2 + \beta^2\Sigma_A\right\|_{op} \leq 1 - \beta\mu_g.$$

*Proof.* First note that $\beta \leq \frac{1}{L_g}$, so that $I - \beta A$ is positive. Now, we observe that $\left\|(I - \beta A)^2 + \beta^2\Sigma_A\right\|_{op} \leq (1 - \beta\mu_g)^2 + \beta^2\sigma_A^2$ which holds since $I - \beta A$ is positive. And since $\beta \leq \frac{\mu_g}{\mu_g^2 + \sigma_A^2}$, we further have $(1 - \beta\mu_g)^2 + \beta^2\sigma_A^2 \leq 1 - \beta\mu_g$, which yields the desired result. $\square$

**Lemma 3.** *Let $0 \leq b < 1$ and $n \geq 1$, then the following inequality holds:*

$$nb^2(1 - b)^{n-1} \leq (1 - \frac{b}{2})^{n-1}.$$

*Proof.* We consider the function $h(n, b)$ defined by:

$$h(n, b) := (n - 1)\log\left(\frac{1 - \frac{b}{2}}{1 - b}\right) - \log(nb^2).$$

We need to show that $h(n, b)$ is non-negative for any $n \geq 1$ and $0 \leq b < 1$. For this purpose, we fix $b$ and consider the variations of $h(n, b)$ in $n$:

$$\partial_n h(n, b) = \log\left(\frac{1 - \frac{b}{2}}{1 - b}\right) - \frac{1}{n}.$$

$\partial_n h(n, b)$ is non-negative for $n \geq n^\star = \log\left(\frac{1 - \frac{b}{2}}{1 - b}\right)^{-1}$ and non-positive for all $n \leq n^\star$. Hence, $h(n, b)$ achieves its minimum value in $n^\star$ over the $(0, +\infty)$. We distinguish two case depending on whether $n^\star$ is greater of smaller than 1.

**Case $n^\star \leq 1$.** In this case $n \mapsto h(n, b)$ is increasing on the interval $[1, +\infty)$ since $\partial_n h(n, b) \geq 0$ for $n \geq n^\star$. Hence, $h(n, b) \geq h(1, b)$ for all $n \geq 1$. Moreover, since $h(1, b) = -\log(b^2) \geq 0$ the result follows directly.

**Case** $n^\star > 1$. In this case we still have $h(n, b) \geq h(n^\star, b)$ for all $n \geq 1$, since $n^\star$ achieves the minimum value of $h$. Thus we only need to show that $h(n^\star, b) \geq 0$. Using the expression of $n^\star$, we have:

$$h(n^\star, b) = 1 - \log\left(\frac{1 - \frac{b}{2}}{1 - b}\right) - \log\left(n^\star b^2\right)$$

$$= 1 - \frac{1}{n^\star} - \log\left(n^\star b^2\right)$$

Since $n^\star > 1$, the first term $1 - \frac{1}{n^\star}$ is non-negative, thus we only need to show that $n^\star b^2 \leq 1$ so that the last term is also non-negative. It is easy to see that $n^\star b^2 \leq 1$ is equivalent to having $\tilde{h}(b) \geq 0$, where we define the function $\tilde{h}(b)$ as:

$$\tilde{h}(b) = \log\frac{1 - \frac{b}{2}}{1 - b} - b^2.$$

We can analyze the variations of $\tilde{b}$ be computing its derivative which is given by:

$$\partial_b \tilde{h}(b) = \frac{1}{(1 - b)(2 - b)} - 2b.$$

Hence, we have the following equivalence:

$$\partial_b \tilde{h}(b) \geq 0 \iff 2b(1 - b)(2 - b) \leq 1$$

This is always true for $0 \leq b < 1$ since $b(1 - b) \leq \frac{1}{4}$ so that $2b(1 - b)(2 - b) \leq \frac{2 - b}{2} \leq 1$. Thus we have shown that $\tilde{h}$ is increasing over $[0, 1)$ so that $\tilde{h}(b) \geq \tilde{h}(0) = 0$. As discussed above, this is equivalent to having $n^\star b^2 \leq 1$, so that $h(n^\star, b) \geq 0$ which concludes the proof.

$\square$

# F EXPERIMENTS

## F.1 DETAILS OF THE SYNTHETIC EXAMPLE

We choose the functions $f$ and $g$ to be of the form: $f(x, y) := \frac{1}{2}x^\top A_f x + y^\top C_f$ and $g(x, y) := \frac{1}{2}y^\top A_g y + y^\top B_g x$ where $A_f$ and $A_g$ are symmetric definite positive matrices of size $d_x \times d_x$ and $d_y \times d_y$, $B_g$ is a $d_y \times d_x$ matrix and $C_f$ is a $d_y$ vector with $d_x = 2000$ and $d_y = 1000$.

We generate the parameters of the problem so that the smoothness constants $L$ and $L_g$ are fixed to 1, $\kappa_\mathcal{L} = 10$ and $\kappa_g$ taking values in $\{10^i, i \in \{0, .., 7\}\}$. We then solve each problem using different methods and perform a grid-search on the number of iterations $T$ and $M$ of algorithms $\mathcal{A}_k$ and $\mathcal{B}_k$.

We fix the step-sizes to $\gamma_k = 1/L$ and $\alpha_k = \beta_k = 1/L_g$ and perform a grid-search on the number of iterations $T$ and $M$ of algorithms $\mathcal{A}_k$ and $\mathcal{B}_k$ from $\{10^i, i \in 0, 1, 2, 3\}$. For AID methods without warm-start in $\mathcal{B}_k$, we consider an additional setting where $M$ increases logarithmically with $k$, as suggested in Ji et al. (2021), with $M = \lfloor 10^3 \log(k) \rfloor$. Similarly, for (ITD) and (Reverse), we additionally use an increasing $T$ of the same form.

## F.2 EXPERIMENTAL DETAILS FOR LOGISTIC REGRESSION

The inner-level and outer-level cost functions for such task take the following form:

$$f(x, y) = \frac{1}{|\mathcal{D}_{val}|}\sum_{\xi \in \mathcal{D}_{val}} L(y, \xi), \qquad g(x, y) = \frac{1}{|\mathcal{D}_{tr}|}\sum_{\xi \in \mathcal{D}_{tr}} L(y, \xi) + \frac{1}{pd}\sum_i \exp(x_i)\|y_{.,i}\|^2$$

For the *default* setting, we use the well-chosen parameters reported in Grazzi et al. (2020); Ji et al. (2021) where $\alpha_k = \gamma_k = 100$, $\beta_k = 0.5$, and $T = N = 10$. For the grid-search setting, we select the best performing parameters $T$, $M$ and $\beta_k$ from a grid $\{10, 20\} \times \{5, 10\} \times \{0.5, 10\}$, while the batch-size (chosen to be the same for all steps of the algorithms) varies from $10 * \{0.1, 1, 2, 4\}$. We also compared with VRBO (Yang et al., 2021) using the implementation available online and noticed instabilities for large values of $T$ and $N$, as reported by the authors, but also a drop in performance compared to stocBiO for smaller $T$ and $N$ due to inexact estimates of the gradient.

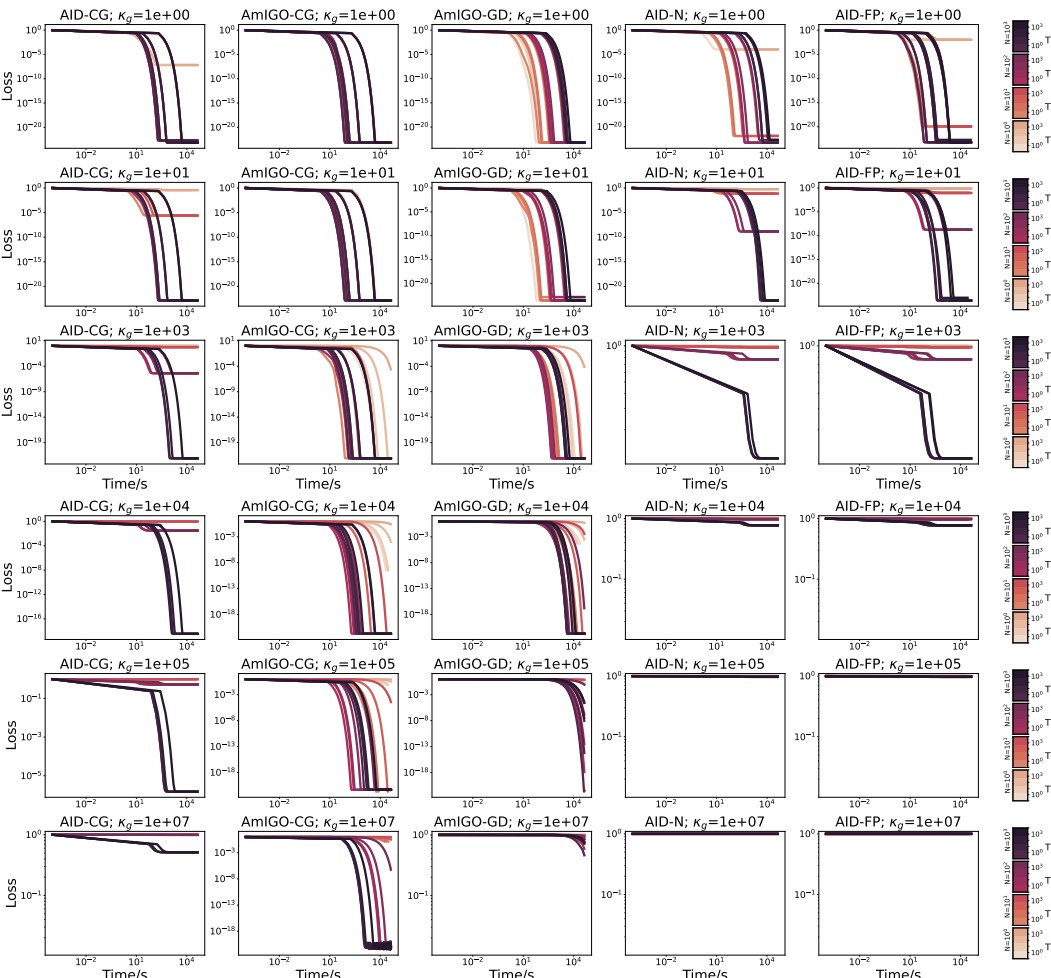

Figure 2: Evolution of the relative error vs. time in seconds for different AID based methods on the synthetic example. Each column corresponds to a method (AID-CG, AmIGO-CG, AmIGO-GD, AID-N, AID-FP) and each row corresponds to a choice of the conditioning number $\kappa_g$. For each method we consider $T$ and $N$ from a grid $\{1, 10, 10^2, 10^3\} \times \{1, 10, 10^2, 10^3\}$. Lightest colors corresponds to smaller values of $N$ while nuances within each color correspond to increasing values of $T$.

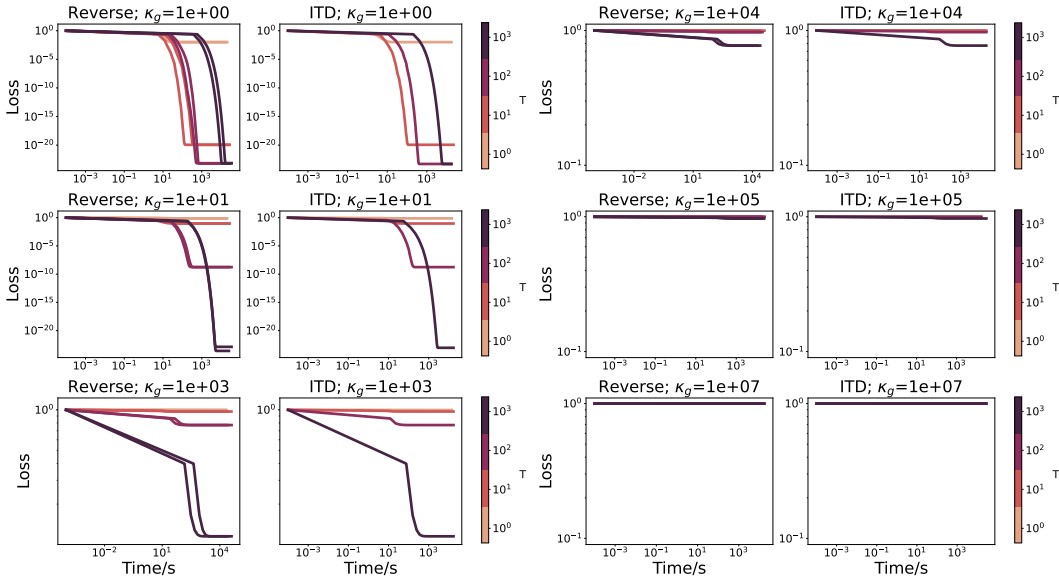

Figure 3: Evolution of the relative error vs. time in seconds for different ITD based methods on the synthetic example. From the left to the right, the first two columns correspond to Reverse and ITD method small conditioning numbers $\kappa_g \in \{1, 10, 10^3\}$, last two column are for higher conditioning numbers $\kappa_g \in \{10^4, 10^5, 10^7\}$. For each method we consider $T \in \{1, 10, 10^2, 10^3\}$. Lightest colors correspond to smaller values of $T$.

## F.3 DATASET DISTILLATION

Dataset distillation (Wang et al., 2018; Lorraine et al., 2020) consists in learning a small synthetic dataset such that a model trained on this dataset achieves a small error on the training set. Specifically, we consider a classification problem of $C$ classes using a linear model and a training dataset $\mathcal{D}_{tr}$ where each training point $\xi \in \mathcal{D}_{tr}$ is a $d$-dimensional vector with a class $c_\xi \in \{1, ..., C\}$. The linear model is represented by a matrix $y \in \mathbb{R}^{c \times d}$ multiplying a data point $y\xi$ and providing the logits of each class. The dataset distillation can be cas as a bilevel problem of the form:

$$\min_{x \in \mathbb{R}^{c \times d}, \lambda \in \mathbb{R}^d} \frac{1}{|\mathcal{D}_{tr}|} \sum_{\xi \in \mathcal{D}_{tr}} CE(y^\star(x, \lambda)\xi, c_\xi),$$

$$y^\star(x, \lambda) \in \arg\min_{y \in \mathbb{R}^{c \times d}} \frac{1}{C} \sum_{c=1}^{C} CE(yx_c, c) + \frac{1}{Cd} \sum_{i=1}^{d} \exp(\lambda_i)\|y_{.,i}\|^2.$$

where $\lambda \in \mathbb{R}^d$ is a vector of hyper-parameter for regularizing the inner-level problem which we found beneficial to add.

**Experimental setup.** We perform the distillation task on MNIST dataset. We set the step-sizes $\alpha_k = \beta_k = 0.1$ and $T = N = 10$. We perform a grid-search on the outer-level step-size $\gamma_k \in \{0.01, 0.001, 0.0001\}$ and run the algorithms for $k = 10000$ iterations.

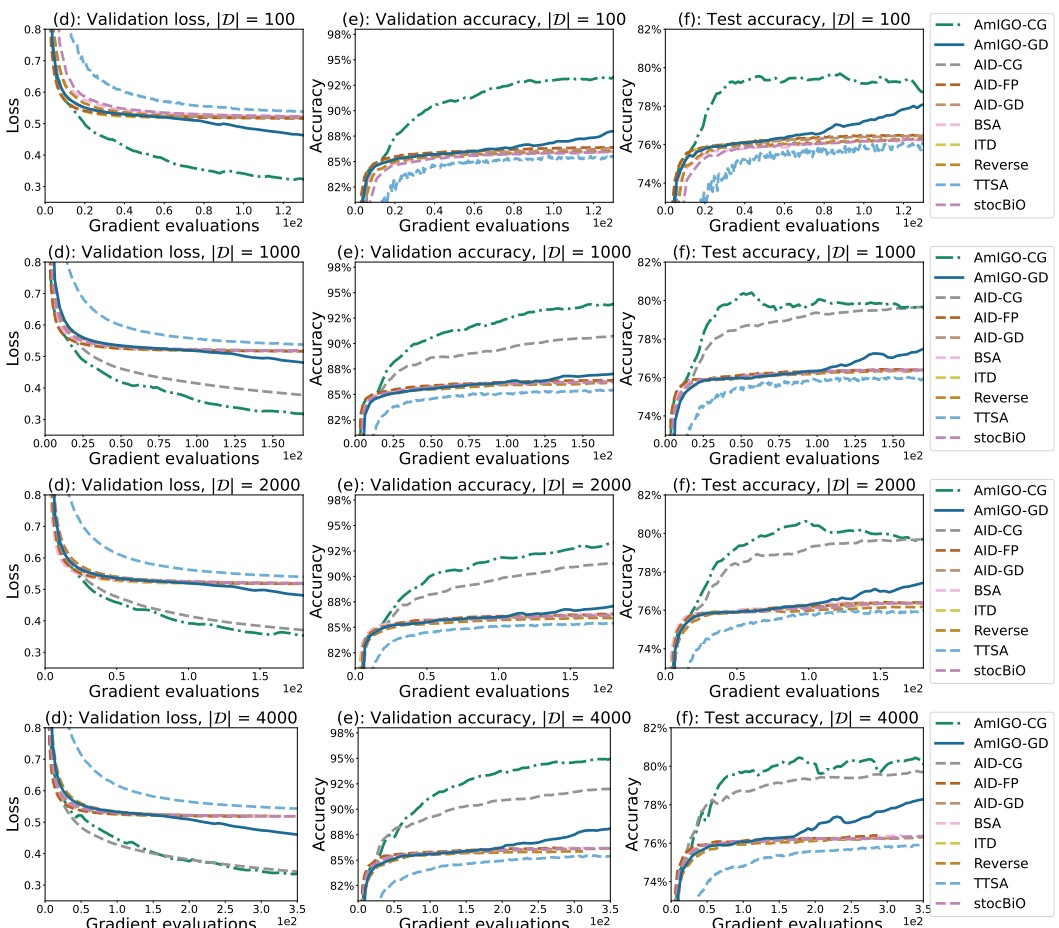

Figure 4: Evolution of the validation loss (left column), validation accuracy (middle column) and test accuracy (right column) in time (s) for different methods on the logistic regression task. Each row correspond to different choices for the size of the batch $|\mathcal{D}| \in \{100, 1000, 2000, 4000\}$ chosen to be the same for all gradient, Hessian and Jacobian-vector products evaluations. Time is reported in seconds.

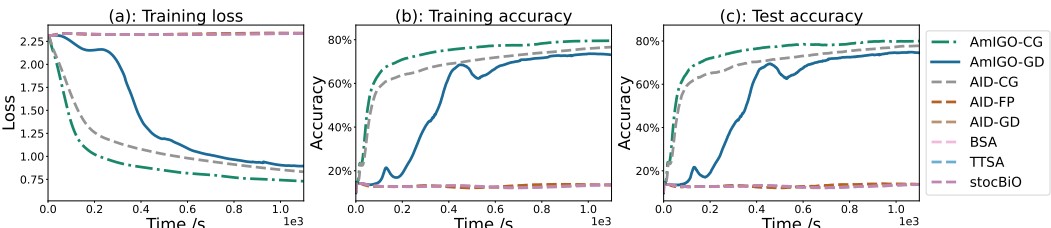

Figure 5: Performance of various bi-level algorithms on the dataset distillation task on MNIST dataset.

