# OpenReview forum: "Amortized Implicit Differentiation for Stochastic Bilevel Optimization"
_ICLR.cc/2022/Conference — ICLR 2022 Poster_

### Official Review · Reviewer_QCfa · 2021-10-25

**Correctness:** 2
**Technical Novelty And Significance:** 2
**Empirical Novelty And Significance:** 2
**Recommendation:** 3
**Confidence:** 3

**Main Review:**

This paper is not well-motivated, and the theoretical and experimental results are not convincing enough.

1. This paper focuses on a class of problems: the inner objective is strongly convex (the strongly convex constant can be negative). But the authors do not explain why such a problem is essential and why it deserves study.  This paper follows [Bilevel Optimization: Convergence Analysis and Enhanced Design] in many places. The authors need to explain more details on the difference between that paper.

2. The comparison between existing works (Table 1) is unfair. For example, the algorithm has no inner loop in [ A two-timescale framework for bilevel optimization: Complexity analysis and application to actor-critic]. But in your method, the inner loops mean that your iteration complexity multiplies a large constant.

3. Even the number of the inner loop is small in the deterministic case. The theoretical results shown in Table 1 mean that the proposed algorithm seems worse than existing algorithms in the strongly convex case. While for the non-convex case, the authors indeed consider the weakly convex function rather than general non-convex cases. The comparison is not convincing.

4. The number of the inner loop $N=O(\kappa_g^3\sigma_{g_{yy}}^2)$, which seems very large. I doubt how to use the algorithm in practice, especially when  $\kappa_g$ is not small.

4. The experiments are weak. The authors only consider quadratic or linear ones.

**Summary Of The Paper:**

This paper proposes amortized implicit differentiation for the bilevel optimization when the inner  objective is strongly
convex.

**Summary Of The Review:**

Due to previous work  [Bilevel Optimization: Convergence Analysis and Enhanced Design], I deem that the novelty of this paper is limited. I vote to reject.

---

> ### Author Response · Authors · 2021-11-22
> **Author response:  Novelty**
>
> Thank you for your comments. We provide the following clarifications about our work which we hope address your concerns and convince you to reconsider your decision. We are open to discussion during the open period if you have any further questions or comments.
>
> ## Novelty compared to Ji et. al. 2021
>
> Our understanding of your main reason for rejection is the novelty compared to Ji et. al 2020. However, we believe our work provides significant contributions which are complementary with the work of Ji et. al 2021 for the following reasons:
>
> 1. Their analysis of the warm-start is **limited to the deterministic non-convex setting**.
> 2. The stochastic version of their algorithm does not use warm-start when solving the linear system.
> 3.There is no hint in the paper about how to extend their analysis of warm-start to important settings such as stochastic non-convex or strongly convex. In fact, in the stochastic setting, warm-start is not proposed for solving the linear system neither does the convergence analysis thus yielding worse complexities than the ones we obtain (see Table 1, Non-convex stochastic setting and corollary 4). As a consequence, many important questions are left open:
>
> - Does warm-start improve the complexity in the challenging stochastic non-convex setting over not using it? If so, what is the complexity?
> - Can warm-start improve the complexity of strongly convex bi-level problems? If so, what is the complexity?
> - Can all these questions be answered in a unified way?
>
> Analyzing the impact of warm-start in bi-level optimization is a challenging problem and was little studied (see discussion section in [1]). Our main contribution is to rigorously show that warm-start can effectively yield faster convergence rates in all these important cases. Therefore, our work bridges this gap in the literature and provides a rigorous justification for warm-start by showing that it yields smaller complexities (and thus faster rates) than previously known ones in the context of bilevel optimization.
>
> [1] Grazzi et al. 2020 “On the iteration complexity of hypergradient computation”.
>
> ## Motivation for the class of problems:
> As we discuss in the introduction, the analysis of general bilevel problems is challenging. When the inner problem is non-convex, the problem can be ill defined, in general, due to multiple possible solutions of the inner-problem. Strongly convex inner problems ensure the bi-level problem is well defined and with a well-defined gradient.  Our paper is neither the only nor the first one to consider such a setting which we discussed in the introduction and section 3.
> There are many situations where the inner-problem is strongly convex,  as we discussed in the paper (Section 3.1 just after the statement of Assumption 1 on strong convexity), (see also [1] and references therein). In particular, this includes hyper-optimization for kernel methods, where by construction, the inner-level problem is strongly convex (see Grazzi et al 2020, 2021, Franceschi 2018).
>
> [1]: Blondel, Mathieu and Berthet, Quentin and Cuturi, Marco and Frostig, Roy and Hoyer, Stephan and Llinares-López, Felipe and Pedregosa, Fabian and Vert, Jean-Philippe, “Efficient and Modular Implicit Differentiation”.

---

> > ### Author Response · Authors · 2021-11-22
> > **Author response: Comparaison of the complexities / Experiments**
> >
> > ## Comparaison of the complexities:
> > ### How to fairly compare complexities?
> > We clarify that our comparisons are based on **counting the total number of derivatives** made by the algorithm for **both inner and outer loop** to reach a given error $\epsilon$. It is fair because it allows a comparison between algorithms regardless of their implementation detail (**with or without inner-loop**). In the revised version, we also provided in Table 1 the dependence on the conditioning numbers which are also in favor of our method. In particular for TTSA for which we derived the dependence on condition numbers in Appendix A.5.1 and report those in Table 1.
> >
> > ###  Smaller complexities are better
> > We clarify that your comment “Even the number of the inner loop is small in the deterministic case” is a positive point: A method that needs fewer derivatives evaluations to reach an error $\epsilon$ is faster and has a **smaller/better complexity** as discussed in the beginning of Section 4.2.
> >
> > ### Strongly convex case: “The theoretical results shown in Table 1 mean that the proposed algorithm seems worse than existing algorithms in the strongly convex case.”
> > We respectfully disagree, the complexities are improved: In the deterministic setting, we need a fewer number of derivative evaluations (only $O(\log(1/\epsilon))$) to achieve an error of order $\epsilon$ compared to other methods (Ghadimi and Wang +  Ji and Liang.) which require a more derivatives ($O(\log(1/\epsilon)^2)$). The same conclusions hold for the stochastic setting.
> >
> > ### Non-convex case  “While for the non-convex case, the authors indeed consider the weakly convex function rather than general non-convex cases. The comparison is not convincing.”
> > We respectfully disagree, we treat a rather general case where the loss is $L$-smooth and non-convex (a fortiori weakly convex). This setting  is considered in all the other works we compared with.  Hence the comparison with the prior works is relevant since we consider a similar problem setting.  In the context of machine learning, we are not aware of any other work treating bi-level problems without the smoothness assumption .
> >
> > ### Dependence of the number of iterations $N$ on $\kappa_g$
> > In the first submission, we obtained a dependence of $N=O(\kappa_g(1 + \kappa_g^2 \sigma^2))$. Such dependence appears in standard analysis of SGD (Thm 4.6 Bottou 2016 et al. “Optimization Methods for Large-Scale Machine Learning”) when the variance of the noise is unbounded (which is our case when solving the stochastic quadratic problem). In the revised version, we improved this dependence to $N = O(\kappa_g)$ provided the batch-size  $|D_{g_{yy}}|$ is of order$\kappa_g$. That being said, our choice of $N$ is still independent of the error $\epsilon$ thanks to warm-start. Without warm-start, all other methods require $N= O(\kappa_g \log\frac{1}{\epsilon})$ which is a worse dependence.
> > Finally, the experimental results in Figure 4 of the Appendix F.1 show that our method consistently outperforms other methods for various choices of the batch-size ranging from {100, 1000,2000,4000}.
> >
> >
> > ## Experiments:
> > In the revised version, we considered three types of experiments: a synthetic one with a quadratic problem, an experiment on hyper-optimization for logistic regression on 20NewsGroup dataset and a dataset distillation task on MNIST. None of these experiments involve a linear loss. The last two yield a non-convex loss $\mathcal{L}$ due to the non-linear dependence of $y^{\star}(x)$ on $x$. All experiments support our analysis which is the main contribution of our paper.

---

> ### Author Response · Authors · 2021-12-01
> **We hope to hear your feedback**
>
> Dear reviewer, we believe we have made substantial effort to answer your comments. We hope to hear your feedback and we would be grateful if you would update your score accordingly.

---

> ### Comment · Area_Chair_ZWfo · 2021-12-03
> **Reply wanted**
>
> Dear reviewer,
>
> The authors replied to all your comments and have uploaded a revised PDF. It would be great if you could reply to them and update your score accordingly.
>
> Best,
> the AC

---

### Official Review · Reviewer_Xg4d · 2021-10-31

**Correctness:** 4
**Technical Novelty And Significance:** 3
**Empirical Novelty And Significance:** 2
**Recommendation:** 6
**Confidence:** 3

**Main Review:**

This is an interesting paper that gives nice theoretical links between the proposed algorithm, SPS, and also the stochastic estimate sequence framework. Here are some questions for the authors.

Q1. In table 1, is it possible to include $\kappa_l$ and $\kappa_g$ dependence in stochastic settings as well?

Q2. After Corollary 1, the authors point out that $\kappa_g$ dependence can be further improved by using acceleration or variance reduction. Is it possible to have a more detailed comparison with [Ji and Liang 2021]?

Q3. It looks like the choice for ${\cal A}_k$ and ${\cal B}_k$ is critical. More discussions should be provided on this point.

Q4. In numerical tests, it looks like in real-world problems CG based algorithms (Amigo-CG and AID-CG) perform the best. Have the authors test it on other datasets? Some additional results to support nonconvex settings are also beneficial.

**Summary Of The Paper:**

This work focuses on bilevel optimization. The key innovation here is the warm start, which enables improved complexity bounds (under some settings). The analysis nicely builds on singularly perturbed systems (SPS). Numerical tests are also provided on both synthetic and real problems, where the merit of warm start is demonstrated.

**Summary Of The Review:**

In general this paper is theoretically interesting. This paper could benefit from more discussions on ${\cal A}_k$ and ${\cal B}_k$, as well as additional experiments.

---

> ### Author Response · Authors · 2021-11-22
> **Author response**
>
> Thank you for the encouraging comments and insightful questions. We are glad to see that you find the analysis in this paper theoretically interesting. In the revised version, we have made a considerable effort to answer all the questions you asked. We hope those new discussions are satisfactory and we are open to discussion during the open period if you have any further questions or comments. We summarize these new discussions below.
>
> ## Dependence on the conditioning in Table 1
> We have now included those in the table. In the non-convex setting, the dominant dependence on $\kappa_g$ is similar to that of (Ghadimi et. al 2018) and (Ji et al. 2021): $\kappa_g^9$ and improves over TTSA $\kappa_g^{16}$.
> In the strongly convex setting, we get $\kappa_{L}^2\kappa_g^3$ which improves over $\kappa_{\mathcal{L}}^4$ in Ghadimi et al. 2018 and $  \kappa_{\mathcal{L}}^{0.5}\kappa_g^{8.5}+ \kappa_{\mathcal{L}}^{3.5}$ in Hong et al. 2020
> and over $\kappa_{\mathcal{L}}^4$ in Ghadimi et al. 2018 when $\kappa_g$ is moderate. We re-derived the explicit dependence on the conditioning for TTSA method in proposition 11 of Appendix A.4.1 based on the main results stated in their paper and their optimal choice of parameters. For MRBO/VRBO, we reported $Poly(\kappa_g)$ since the dependence was not explicitly provided in (Yang et al. 2021).
>
> ## Comparison with Ji and Liang 2021:
> We added a discussion in A.4.2 on the results of Ji and Liang 2021 which obtain a complexity of $O(\kappa_{\mathcal{L}}^{\frac{1}{2}} \kappa_g^{\frac{1}{2}} \log \frac{1}{\epsilon}^2 )$ and compare it to the complexity of AmIGO  $O(\kappa_{\mathcal{L}} \kappa_g \log \frac{1}{\epsilon} )$ in terms of $\kappa_{\mathcal{L}}$ and $\kappa_{g}$:
>
> 1. **Improved dependence on conditioning in Ji and Liang 2021**  due to an acceleration scheme in both inner-level and outer-level problems.
> 2. **Improved dependence  on $\epsilon$ for AmIGO** due to the use of warm-start when solving the linear system in AID which requires only $N= O(\kappa_g)$ iterations while  Ji and Liang 2021 require $N=O(\kappa_g^{\frac{1}{2}} \log (\frac{1}{\epsilon}))$  as their algorithm does not exploit warm-start for the linear system.
> 3. We further clarify that the complexities reported in Ji and Liang 2021 were expressed in terms of $\mu$ and $\mu_g$ instead of $\kappa_{\mathcal{L}}$ and $\kappa_g$. The transition from one notation to another follows by noting that $\kappa_{\mathcal{L}} = \frac{L}{\mu}$ with $L= O(\mu_g^{-2})$ under their assumption that the hessian $\partial_{yy}g(x,y)$ is constant in $y$ which yields a complexity of $O(\mu^{-\frac{1}{2}}\mu_g^{-\frac{3}{2}}(\log\frac{1}{\epsilon})^2 )$. In the general case $L=O(\mu_g^{-3})$ thus yielding a worse dependence on $\mu_g$. Hence, the expression in terms of the conditioning allows a more stable comparison with other results regardless of the assumption made on the hessian.
> 4. In section A.5.1, we discuss how using acceleration in Algorithm A_k and B_k yields an improved dependence on $\kappa_g$ for AmIGO, while retaining the optimal dependence on $\epsilon$: $O(\kappa_{\mathcal{L}} \kappa_g^{\frac{1}{2}} \log \frac{1}{\epsilon} )$.
>
> ## Choice of Algorithms $\mathcal{A}_k$ and $\mathcal{B}_k$
> We discuss in Appendix A.5 the choice of  these algorithms explaining that it has an impact on the total complexity of the algorithm. In particular, we discuss two general classes of algorithms which can improve the complexity:
>
> - **Acceleration:** In Appendix A.5.1, we discuss how using acceleration in $\mathcal{A}_k$ and   $\mathcal{B}_k$ yields a complexity of  $O({\kappa}\kappa_g^{\frac{1}{2}}\log(\frac{1}{\epsilon}) )$
> within our theoretical framework. This is in line with the empirical observations in Figure 1 which shows that Conjugate gradient for $\mathcal{B}_k$ (which enjoys accelerated convergence rates) yields
> faster algorithms. The improvement is even greater as the conditioning number $\kappa_g$ is large as shown in the experiments.
> - **Variance reduction:** In Appendix A.5.2, we discuss how variance reduction algorithms in $\mathcal{A}_k$ can improve the dependence on the conditioning numbers  in the stochastic setting.
>
> ## Additional experiments:
>
> We run additional experiments on a dataset distillation task for MNIST following Grazzi et al 2020 (Figure 5 of Appendix F.2). This results in a bilevel problem where the loss is non-convex. We compare to the other methods and obtain observations that are consistent with the rest of the experiments: warm-start is beneficial for solving the bilevel problem. Note that while this observation is not surprising, the corresponding analysis (which is the main contribution of our paper) is challenging.

---

> > ### Comment · Reviewer_Xg4d · 2021-12-01
> > **Thanks for your response**
> >
> > My concerns are mostly addressed, although mnist is a dataset that is arguably small. Anyway, I believe this paper is more on the theory side and hence I will keep my score.

---

### Official Review · Reviewer_2dfc · 2021-11-01

**Correctness:** 3
**Technical Novelty And Significance:** 2
**Empirical Novelty And Significance:** 2
**Recommendation:** 6
**Confidence:** 3

**Main Review:**

Strengths
Sections 1 to 3 of the paper is well written and provides a clear overview of the problem and algorithm. The authors also propose a unified analysis which encompasses many of the existing results in the literature.

Weaknesses
I find Section 4 difficult to read.  The downside to this unified analysis seems to be that the results are difficult to parse (making it difficult to see how to apply these results beyond the known settings).  Here are some of the issues with the exposition:
- Section 4.1 and 4.2 list properties of the inner and outer problem without providing any clear and concise statements on the conclusions. It would be better to first explain the main result (Theorem 1) before explaining the proof technique.
- Theorem 1 makes use of notation from Proposition 1 without explicitly mentioning this.
- There are a list of corollaries about the case of W = 0 and W>0, but W is not really described and it is difficult to see what this quantity corresponds to -- e.g. what are the situations where W = 0?
- I'm not sure what the message of Proposition 4 is, it makes use of matrices and vectors P_k, U_k, V_k without saying anything about what these are, so I am not sure what to make of this proposition.



**Summary Of The Paper:**

The paper studies the problem of stochastic bilevel optimisation, proving convergence rates for an amortised algorithm which makes use of unbiased estimates of the inner and outer function. The main contributions are theoretical, where they prove sublinear convergence rates.

**Summary Of The Review:**

In general, I feel that a 'unified' analysis should be elegant and more instructive than existing ad hoc analysis, but this does not seem to be the case here. There is a lot of notation introduced without proper explanation of what they represent, so it is difficult to properly read and check the proofs of this paper in the limited reviewing time of ICLR.

---

> ### Author Response · Authors · 2021-11-22
> **Author response**
>
> Thank you for the constructive comments to help improve the clarity of the paper. We are glad to see that you find the exposition in Section 1 to 3 well written and with a good overview of the problem and algorithm. In the revised version of the paper, we took into account all your comments regarding Section 4 and we refer to the second section of this response for a summary of the modifications made to Section 4. In the first section of the response we clarify how **the contributions of the present paper go beyond a unified analysis of known results**.
>
> We hope that the improvements and clarifications made to the revised version and this response addresses all your concerns and convince you to raise the score of the review. We have made a considerable effort to take into account the interesting suggestions that you proposed. In any case, we are open to discuss further during the open period if you have any further questions or comments.
>
>
> ## Theoretical contributions:
> **The contributions of this work proves new convergence results that improve over previously known ones** in the context of bilevel optimization. In particular, we obtain the following improved rates:
> 1. Deterministic strongly convex setting: We prove \bf{linear convergence rates} corresponding to a complexity of $O(\log\frac{1}{2})$. To our knowledge,  this is the first time such rates are obtained in the context of bi-level optimization. Prior work obtained slower rates with a complexity of $O(\log(\frac{1}{\epsilon})^2)$.
> 2. Stochastic strongly convex setting: We obtain the improved rate of $O(\frac{1}{\epsilon}\log(\frac{1}{\epsilon}))$, while the best known rate for such setting was $O(\frac{1}{\epsilon^{-3/2}}\log(1/\epsilon)).$
> 3. Stochastic non-convex setting: We obtain improved rate of $O(\frac{1}{\epsilon^2})$ which improves over the best previously known rate of  $O(\frac{1}{\epsilon^2}\log(\frac{1}{\epsilon}))$ under the same assumptions.
>
> Our main contribution is to rigorously show that such faster rates are possible using  a warm-start strategy. Analyzing the impact of warm-start in bi-level optimization is a challenging problem and was poorly studied (see discussion section in [1]). Therefore, our work bridges this gap in the literature and provides a rigorous justification for warm-start by showing that it yields faster convergence rates than previously known rates in the context of bilevel optimization.
>
> [1] Grazzi et al. 2020 “On the iteration complexity of hypergradient computation”.
>
> ## Modifications in section 4:
> ### Improved structure for section 4:
> We first state Theorem 1 before explaining the proof technique as you suggested. More precisely:
> - **Section 4.1** first describes the general approach and \emph{states the main theorem (Theorem 1)}. Now the statement of Theorem 1 \emph{does not use any notation from Proposition 1 anymore} , only from the main text.
> - **Section 4.2** discusses the main corollaries of Theorem 1 and \emph{how they improve over previously known results}.
> - **Section 4.3** provides a sketch of the proof discussing how the errors of both inner and outer level are controlled using propositions 2, 3, 4 and 5. For each proposition **we provide a concise statement explaining why such a result is needed**.
>
> ### Clarifying the main message from Proposition 5 (previously Proposition 4):
> In the revised version, we discuss the result of Proposition 5 as follow:
> The main purpose of Proposition 5 is to characterize how the inner-level errors evolve in time, which is important since the error in the estimation of the gradient is bounded by the inner-level errors by proposition 4.  The matrices and vectors arise from discretization errors and depend on the step-sizes and constants of the problem. Their exact expression is not important for a high level interpretation of the result. We thus deferred their exact expression to the appendix due to space constraints.
>
> ### Meaning of $W^2$ (previously $W$)
> We redefined $W^2$ so that it can be interpreted as an effective variance of the problem:
> “The **effective variance** $\mathcal{W}^2$ accounts for  interactions between both levels in the presence of noise and becomes proportional to the outer-level variance $\sigma_f^2$ when the inner-level problem is solved exactly. In the deterministic setting, all variances vanish so that $\mathcal{W}^2{=}0$. Hence, we characterize such a setting by $\mathcal{W}^2{=}0$ and the stochastic one by $\mathcal{W}^2{>}0$.”

---

> > ### Comment · Reviewer_2dfc · 2021-11-29
> > **response to authors**
> >
> > Thanks for the response and clarifications. Overall, I find the paper well written, and the theoretical bounds are a useful contribution to the literature. I have raised my score to a '6', but I think that this paper is better suited to a journal to allow for proper reviewing of the proofs.

---

> > > ### Author Response · Authors · 2021-11-29
> > > **Thank you for your response**
> > >
> > > Thank you for your response and for raising your score. We are glad to hear that you find the paper well written and the contributions useful.
> > > This paper provides indeed a detailed proof analyzing warm-start in several cases of interest. Yet, we believe the take home message from the paper is both simple and of high practical use: “warm start provably speeds up bi-level optimization methods” and is therefore well suited for publication at ICLR which has a history of long theory papers with a practical importance such as the references [1,2,3] provided in the response to reviewer 6zLQ.

---

### Official Review · Reviewer_6zLQ · 2021-11-01

**Correctness:** 3
**Technical Novelty And Significance:** 3
**Empirical Novelty And Significance:** 2
**Recommendation:** 8
**Confidence:** 2

**Main Review:**

Their general proof strategy is to use previous results on singularly perturbed systems with different time scales and a lyapunov-esque analysis. Their rates/complexities improve significantly on previous works (except in the case of nonconvex upper level/strongly convex lower level stochastic methods, for which a previous work provides a better rate under a slightly stricter assumption) and their arguments encompass both the nonconvex upper level and strongly convex upper level cases at the same time (also the deterministic and stochastic cases at the same time), giving a unified analysis.

The assumptions, claims, and arguments in the paper are rigorously stated and seemingly rigorously proved. However, the majority of the proofs are relegated to the supplementary material, which isn't required to be checked during the review. When I did check some of the proofs, I found an error in one of them that could affect the validity of other arguments. In particular, in the proof of prop 11 in the supplementary material, there is an error in the update equation (should be k-1 in psi subscript) which affects (perhaps only superficially) the definition of zeta_k and perhaps of some other things. In section 4, after “Outer-level problem”, it is written "and u\in [0,1]" Surely this should be u\in\{0,1\}? Is u ever taken to be in (0,1)? I am inclined to believe these can be easily fixed but it’s not completely clear due to the overall density of the paper.

Remarks about trivial/superficial details (I didn't use these for my evaluation but you might be happy to correct them anyways):
* Immediately after proposition 1 it is written "... provides an expression for L in terms of partial derivatives of f and g evaluated at (x,y*(x))." Should this be “an expression for \nabla L”?
* In Appendix A.2 the first line states "In order to prove Appendix A.2 ..." Should this be "In order to prove Theorem 1"?
* In the proof of proposition 6 in the appendix it's written "This allows to deduce that ... any by application" Should this be "and by application"?
* In B.3 it is written "the second inequality" but I can only find one inequality?
* There are too many commas in the "Controlling the iterates z^n of B_k" section where it’s written “the following choices for A_n, A, \hat{b}, b:”

**Summary Of The Paper:**

The paper presents a quite rigorous analysis of approximate implicit differentiation with warm starts applied to strongly convex upper level/strongly convex lower level and nonconvex upper level/strongly convex lower level bilevel optimization algorithms in a very general yet also very practical framework. They allow for stochastic errors in the algorithms solving the upper and lower level problems, making their work practical and applicable to real problems in machine learning (hyperparameter optimization), while analyzing in a way that is agnostic towards which algorithms are specifically used for the lower and upper level problems.

**Summary Of The Review:**

* Pros: rigorous arguments, clearly stated assumptions, problem framework is applicable to practice, very good rates in all cases (strongly convex/nonconvex, stochastic/deterministic) except one case (in which the previous related work uses a slight stricter assumptions), comprehensive bibliography/related work section, compelling experiments on both synthetic and "real world" problems with comparisons to many other methods of interest in the literature, generally well written and clear given the subject and level of rigor employed.
* Cons: extremely dense and notation heavy for a conference paper (otherwise well written), I was unable to check several proofs in detail; in some sense this work is much better suited for a journal where it can be reviewed properly.

---

> ### Author Response · Authors · 2021-11-22
> **Author response**
>
> Thank you for the encouraging and constructive remarks. We are happy to see that you found our paper well written, the analysis rigorous, and the framework to be general and yet practical.
>
> In an effort to treat several cases of interest (those in table 1), we provided a detailed proof that analyzes all these cases at once. Yet, we believe the take home message from the paper is both simple and of high practical use:  “warm start provably speeds up bi-level optimization methods” and is therefore well suited for publication at ICLR which has a history of long theory papers with a practical importance such as [1,2,3].
>
> In the revised version of the paper, we fixed the few mistakes and typos that you pointed out and which have no incidence on the validity of the results as we discuss below. We hope we addressed all of your concerns and we will do our best to answer any further questions or comments you would have during the discussion period.
>
> ## Proof of Proposition 11
> Thank you for pointing us to the mistake in the proof of Prop 11, which we have fixed in the revised version. The statement of this proposition remains correct, which does not affect the rest of the paper. More precisely, as the reviewer noticed, this implies replacing the subscript $k$ by $k-1$ in the definition of $\zeta_k$. Hence, $\zeta_k$ becomes $ \zeta_k =  2L\ min((1-u)^{-1},L\eta_{k-1}^{-1}) $ instead of $\zeta_k =  2L \min((1-u)^{-1},L\eta_{k}^{-1}) $. This does not affect the rest of the proofs since, the only place where the expression of zeta is used is in Proposition 10, where we choose $\eta_{k}$  to be constant with $\eta_k= \eta_{k-1}=\mu$ in the strongly convex case and $\eta_k=\eta_{k-1}=L$ in non-convex case.
>
>
> ## Improving the notations and reducing the denseness
>
> ### An improved structure for section 4.
> We improved the structure of section 4 to reduce the denseness due to technical notations and also in accordance with the suggestions of reviewer R3. More precisely,
> - **Section 4.1** first describes the general approach and states the main theorem 1.
> - **Section 4.2** discusses the main corollaries while
> - **Section 4.3** provides a sketch of the proof discussing how the errors of both inner and outer level are controlled using propositions 2, 3, 4 and 5.
>
> We hope that by doing so, a reader interested mainly in the convergence results can have a clear idea by reading section 4.1 and 4.2, and then get an idea of the main approach in section 4.3. We also hope that the proof outline that we provide in Appendix A.1 allows the reader to quickly get the gist of the approach used in the proof.
>
> ### Choice of u and other typos:
> Thank you for bringing these to our attention. In the revised version, we have restricted the parameter u to {0,1} instead of [0,1] as these are the only values used in the paper. We also fixed all the listed typos.
>
> ## References
> [1] Gorbunov, Eduard and Bibi, Adel and Sener, Ozan and Bergou, El Houcine and Richtárik, Peter, ‘A stochastic derivative free optimization method with momentum’, ICLR 2020.
>
> [2] Chaoyue Liu and Mikhail Belkin, Accelerating SGD with momentum for over-parameterized learning, ICLR 2020.
>
> [3] Rahul Kidambi, Praneeth Netrapalli, Prateek Jain, Sham M. Kakade, “On the insufficiency of existing momentum schemes for Stochastic Optimization”

---

> > ### Comment · Reviewer_6zLQ · 2021-11-29
> > **Reponse to the authors**
> >
> > Regarding the proof of Proposition 11, if these sequences are ultimately constant then what is the utility of writing the proofs/theorems with general sequences? Otherwise I appreciate the response and maintain my initial scores.

---

> > > ### Author Response · Authors · 2021-11-29
> > > **Thank you for your response**
> > >
> > > Thank you for your feedback, we are glad to hear you appreciated the response.
> > > To answer your question, we agree that those results could be stated using constant sequences in our context.
> > > We chose however to use the general sequences to highlight that the intermediate results in Proposition 2, 3, 4 and 5 are general and could still be used directly to analyse other bi-level algorithms including those where the step-size decreases. Hence, only Proposition 10 (which was previously Proposition 11) would need to be adapted. We will add a discussion on this matter in the conclusion to clarify that those general results could be used for future work.

---

### Official Review · Reviewer_ikrt · 2021-11-04

**Correctness:** 4
**Technical Novelty And Significance:** 3
**Empirical Novelty And Significance:** 2
**Recommendation:** 6
**Confidence:** 3

**Main Review:**

The paper studies the problem of bi-level optimization using implicit differentiation. In particular, the paper focuses on methods that approximately solve the inner problem and the implicit differentiation. Warm-start strategy is used for solving the inner problem and the implicit differentiation, hence the name “amortized”, although I felt the word “amortized” is a bit less accurate than “warm-up” given that the method still does iterative updates with a warm-up strategy.

The convergence of the proposed method is analyzed by viewing the iterates of the optimization as a dynamical system. It is different front the previous literature in the way that it analyzes approximate inner optimization and implicit differentiation with warm-up strategy and with faster convergence in the stochastic setting. But I didn’t check the proofs since I am not very familiar with the techniques used in the literature.

The experimental results show that the proposed method AmIGO is better than its counterpart without warm-start.
In terms of weakness, I would hope to see how AmIGO compares to AID-BiO proposed in [1] with warm-start only for the inner-level problem or for solving the linear system in AID.

[1] Kaiyi Ji, Junjie Yang, and Yingbin Liang. Bilevel optimization: Convergence analysis and enhanced design. International Conference on Machine Learning, pages 4882–4892. PMLR, 2021.


**Summary Of The Paper:**

The paper studies the problem of bi-level optimization. In particular, it considers algorithms based on inexact implicit differentiation, where the inner problem and the implicit differentiation are not solved exactly. Warm-up is used when solving the inner problem and implicit differentiation. The convergence of the proposed method is analyzed by viewing the iterates of the proposed method as a dynamical system using the idea of singularly perturbed systems.

**Summary Of The Review:**

The paper proposes a method that uses warm-up for both solving the inner problem and the linear system arising from AID. It also proposes an improved convergence analysis of the warm-start strategy with inspiration from the singularly perturbed systems. The proposed method shows faster convergence than its counterpart without warm-up. But I would like to see comparing it with warm-up only for the inner problem or for solving the linear system in AID.

---

> ### Author Response · Authors · 2021-11-22
> **Author response**
>
> Thank you for the encouraging and constructive remarks. In the revised submission, we added the comparaisons that you suggested, which we summarize below.
>
> ### Comparaison with AID-BiO  + warm-start only for the linear system in AID
> We added this comparaison and refer to it by AID-CG-WS in Figure 1 for the quadratic example. We used the same protocol as for the other methods (similar grid search, setting). We observe that not using warm-start for algorithm $A_k$ (inner-problem) achieves a lower performance compared to using warm-start for both $A_k$ and $B_k$ as done by AmIGO.
>
> ### Comparaison with AID-BiO + warm-start only for the inner-problem
> We clarify in the paper that this corresponds to the method AID-CG  which uses warm-start for algorithm $A_k$ and not for algorithm $B_k$. The figure shows that AID-CG under-performs AmIGO as it does not exploit warm start for solving the linear system, as soon as the linear systems are poorly conditioned or high-dimensional, or both.
>
> These two ablations show that warm-start is beneficial for both the inner-level problem and for solving the linear system in AID. Note that while this observation is not surprising, the corresponding analysis (which is the main contribution of our paper) is challenging.
>
> In conclusion, these empirical observations along with the rest of the experiments support the theory that we provide, which shows faster convergence rate compared to methods that do not exploit warm-start in both deterministic and stochastic settings, when the loss is either strongly convex or non-convex.
>
> We hope we addressed all your comments and we are happy to discuss further during the open period - please let us know if you have any further questions or comments, and we will do our best to answer.

---

### Author Response · Authors · 2021-11-22
**General response**

We thank the reviewers for their insightful comments and for encouraging us to strengthen the discussions and presentation of the paper. We have updated a revised version of the paper which includes the suggestions of the reviewers. We hope this new version addresses their concerns and we will do our best to answer any further questions or comments that may arise during the open period. We now summarize the corresponding modifications:
1. We implemented all suggestions of R3 to clarify the exposition of Section 4 and
improved the structure of Section 4 as follows:
     - Section 4.1 describes the general approach and states the main theorem 1.
     - Section 4.2 states the corollaries and discusses them.
     - Section 4.3 provides a sketch of the proofs a states Propositions 2,3,4 and 5 which represents the main steps of the proofs.
2. We corrected the few mistakes and typos that R2 pointed out, which do not affect the validity of the proof as we further discuss in the response to R2.
3. We included the dependence on the conditioning numbers in Table 1 as suggested by R4, with a re-derivation of those for TTSA (Hong et al. 2020 ) in Appendix A.4.1 for the comparison suggested by R5.
4. We added a detailed comparison with the accelerated bi-level algorithm proposed  by Ji and Liang 2021 in Appendix A.4.2 as suggested by R4.
5. In Appendix A.5, we added a detailed discussion on the choice of the inner-level algorithms $\mathcal{A}_k$ and $\mathcal{B}_k$ in particular accelerated algorithms (Appendix A.5.1) and variance reduction (Appendix A.5.2).
6. We added the experimental comparisons suggested by R1 with AID-BiO in Table 1.
7. We added an experiment on dataset distillation in section 5.3 which yields results that are consistent with the analysis in the paper.

---

### Decision · Program_Chairs · 2022-01-20

**Decision:**

Accept (Poster)

**Comment:**

The paper presents a quite rigorous analysis of approximate implicit differentiation with warm starts applied to strongly convex upper level/strongly convex lower level and nonconvex upper level/strongly convex lower level bilevel optimization algorithms in a very general yet also very practical framework. They allow for stochastic errors in the algorithms solving the upper and lower level problems, making their work practical and applicable to real problems in machine learning (hyperparameter optimization), while analyzing in a way that is agnostic towards which algorithms are specifically used for the lower and upper level problems.

Three out of four reviewers were rather positive of the paper (scores: 6, 6, 8). One reviewer was very negative (score: 3). To my knowledge, the authors have convincingly answered all the points raised by the reviewer. Unfortunately, the reviewer did not follow up.

Similarly to reviewers, I found sections 1-3 to be extremely well-written and to give a nice overview of the field. Section 4 had slight clarity issues (dense notation) that were addressed in the revision. Reviewer 6zLQ partially proof-read proofs.

Overall, I recommend acceptance as a poster, as this paper is advancing stochastic implicit differentiation and should be of interest to many at the ICLR conference.